# Understanding Catastrophic Forgetting In LoRA via Mean-Field Attention Dynamics

**Hugo Koubbi** [* 1]   **Louis Hernandez** [* 2]   **Matthieu Boussard** [2]

## Abstract

Low-Rank Adaptation (LoRA) is the dominant parameter-efficient fine-tuning method due to its favorable compute-performance trade-off, yet it suffers from catastrophic forgetting. We study forgetting through a tractable *mean-field self-attention* toy model, where tokens evolve as an interacting particle system and LoRA acts as a low-rank perturbation. Using tools from partial differential equations and dynamical systems, we characterize regimes suggesting a phase transition between forgetting and non-forgetting behavior. We show that one phase transition appears with respect to the norm of the perturbation, and the other with respect to the depth of the Transformers. We further bound the time-to-deviation in terms of the perturbation size and spectral quantities, and corroborate the predicted trends with experiments and exploratory analyses on real models under LoRA fine-tuning.

## 1. Introduction

Since their introduction (Vaswani et al., 2017), transformer architectures have been scaled to large language models (LLMs) with unprecedented capabilities. However, even for open-source LLMs such as Touvron et al. (2023); Jiang et al. (2023); Yang et al. (2025), fine-tuning remains a practical bottleneck under limited hardware, because of both memory footprint and computational cost.

To mitigate this issue, parameter-efficient fine-tuning methods (Mangrulkar et al., 2022), such as Low-rank Adaptation (LoRA) (Hu et al., 2022), have been proposed. LoRA reduces the number of trainable parameters by learning low-rank updates to the attention projection matrices, while keeping the pre-trained backbone fixed. In practice, it preserves much of the pre-trained model's performance, despite significantly lowering the computational and memory costs of fine-tuning.

However, a practical challenge is to acquire new capabilities without degrading previously learned ones, leading to a phenomenon known as *catastrophic forgetting* (Li & Hoiem, 2017). The complexity of full-scale LLMs makes them intractable to analyze mathematically; we study a simplified toy model aimed at understanding forgetting from a theoretical perspective. Following the recent *mean-field Transformer* viewpoint (see Rigollet (2025) for a survey), we model the forward pass as the evolution of an interacting particle system of tokens. From this point, we use changes in the emergent representation geometry as a proxy for forgetting.

**Mean-field modeling.** The Neural ODE framework (Haber & Ruthotto, 2017; Weinan, 2017; Chen et al., 2018) views depth as (discrete) time and analyzes continuum(-depth) limits of residual networks. In particular, a ResNet (He et al., 2016) can be seen as a forward Euler discretization of

$$\dot{x}(t) = f_\theta(x(t)), \tag{1}$$

where $\theta$ denotes learned parameters and $f_\theta : \mathbb{R}^d \to \mathbb{R}^d$ is a vector field acting on representations.

A rapidly growing line of work recasts the forward pass of a deep (encoder) Transformer as the evolution of a cloud of interacting *tokens*: depth plays the role of time, token embeddings are viewed as particles (often constrained to $\mathbb{S}^{d-1}$ after normalization), and the network induces a flow map $\mu_0 \mapsto \mu_T$ (with $T > 0$) on the space of probability measures by evolving an interacting particle system together with its mean-field limit (Geshkovski et al., 2023; 2024; 2025; Chen et al., 2026; 2025; Bruno et al., 2025; 2026; Karagodin et al., 2024; 2026; Alcalde et al., 2025; Cowsik et al., 2025; Koubbi et al., 2026; Fedorov et al., 2026; Agazzi et al., 2026). In particular, Sander et al. (2022) propose a continuous-depth idealization of self-attention in which

---
[*]Equal contribution [1]Université Paris Dauphine, France [2]Craft AI, 34 Rue Guersant, Paris, France. Correspondence to: Hugo Koubbi <hugo.koubbi@dauphine.psl.eu>.

*Proceedings of the 43rd International Conference on Machine Learning*, Seoul, South Korea. PMLR 306, 2026. Copyright 2026 by the author(s).

token embeddings $(x_i(t))_{i=1}^n \subset \mathbb{R}^d$ evolve according to

$$\dot{x}_i(t) = \sum_{j=1}^n \frac{\exp\left(\langle Q(t)x_i(t), K(t)x_j(t)\rangle\right)}{\sum_{k=1}^n \exp\left(\langle Q(t)x_i(t), K(t)x_k(t)\rangle\right)} V(t)x_j(t),$$
(2)

where $(x_1(0), \dots, x_n(0)) \in \mathbb{R}^{d \times n}$ are the initial tokens and $(Q(t), K(t), V(t))_{t \geq 0} \in (\mathbb{R}^{d \times d})^3$ are the given attention matrices typically called (Query, Key, Value). Denote the state $(x_1(T), \dots, x_n(T))$, produced by Eq.(2), as the *Transformer representation* at depth (time) $T$. In the rest of the paper, we work under the following assumption.

**Assumption 1.1.** The matrices $(Q(t), K(t), V(t))_{t \geq 0}$ are constant in time, i.e.,

$$K(t) = K, \quad Q(t) = Q, \quad V(t) = V. \tag{3}$$

This particle-system viewpoint has already yielded a detailed mathematical picture of the *asymptotic geometry* of representations, including clustering and representation collapse.

Following the literature on mean-field transformers (see Section 1), we study a tractable model: we assume a **single learned attention head that is tied (identical) across layers**. Under this tied-weights assumption, the forward pass reduces to an interacting particle system that can drive token embeddings toward a *clustering* regime[1].

From this point onward, we interpret changes in the emergent representation geometry as a *proxy* for forgetting. In our toy model, the long-time cluster configuration summarizes the representations produced by the pre-trained dynamics, and the model's predictions are functions of these representations. LoRA modifies the forward dynamics, which can move the clusters and thus change the representations fed to the output layer; we interpret such deviations in cluster configurations as a proxy for forgetting. Empirically, this geometric drift correlates with degraded base-task performance (e.g., higher base perplexity), which we report in several experiments.

Mathematically, we quantify these deviations using the Wasserstein distance between the empirical measures of the tokens, or by qualitatively comparing the two limiting clusters. We correlate these proxies with the degradation of base-task perplexity (next-token prediction on a fixed dataset).

**LoRA modeling.** At layer $\ell$, we consider LoRA-modified attention matrices $(\widetilde{Q}^\ell, \widetilde{K}^\ell, \widetilde{V}^\ell)$ of the form, for $M \in \{Q, K, V\}$;

$$\widetilde{M}^\ell = M + \Delta M^\ell, \tag{4}$$

---

[1] We empirically observe a *collapse of the representation* in Llama 2 (Touvron et al., 2023), see Section E.5, motivating the relevance of the toy model.

with low-rank updates for $M \in \{V, K, Q\}$ given by

$$\Delta M^\ell = (M_A^\ell)^\top M_B^\ell,$$

where $(V_A^\ell, V_B^\ell, K_A^\ell, K_B^\ell, Q_A^\ell, Q_B^\ell) \in (\mathbb{R}^{r \times d})^6$ and $r \ll d$ is the LoRA rank.

We investigate two stylized regimes for the LoRA factors across depth:

- **Deterministic (tied) adapters:** for all $\ell \in \{1, \dots, L\}$, $M^\ell = M$ (and similarly for the other factors). for $M \in \{Q, K, V\}$.

- **Random adapters:** for all $\ell \in \{1, \dots, L\}$ $(\Delta M^\ell)_{\ell=1}^L \overset{\text{i.i.d.}}{\sim} \rho$ (and similarly for the other factors), where $\rho$ is a Gaussian distribution on matrices for $M \in \{Q, K, V\}$.

Deterministic model can be viewed as a worst-case scenario for forgetting, while the random-adapter model serves as a proxy for an "average-case" effect and admits sharp predictions via homogenization-type arguments (see Section D). Our goal is to understand how LoRA updates

1. affect the forward-pass dynamics (and the induced representation map $(x_1, \dots, x_n) \mapsto (x_1(L), \dots, x_n(L))$),

2. and characterize a transition between *representation stability* and *representation degradation or collapse* induced by these updates.

Throughout, we view the backbone $(Q, K, V)$ as already trained and focus on the effect of LoRA at inference-time, rather than modeling the optimization dynamics itself.

**Contributions.** Our contributions are:

- **A tractable forgetting model.** We propose a mean-field self-attention toy model with tied weights, where LoRA acts as a low-rank perturbation, and we quantify forgetting via representation-geometry drift (cluster displacement or Wasserstein proxy) that empirically correlates with base-task degradation.

- **General perturbation stability.** We prove a quantitative stability bound in Wasserstein distance for the mean-field dynamics under perturbations of $(Q, K, V)$ (Proposition 3.1).

- **Long-time stability and spectral role.** In the Post-LayerNorm setting, we identify a *spectral condition* on $(Q, K, V)$ under which the limiting cluster direction is stable, yielding an explicit bound on the induced drift (Proposition 3.3).

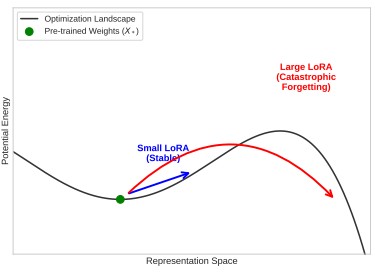

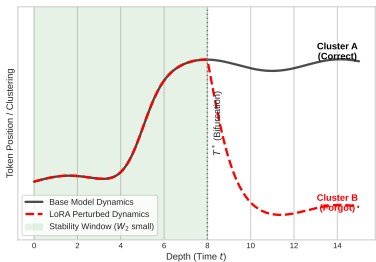

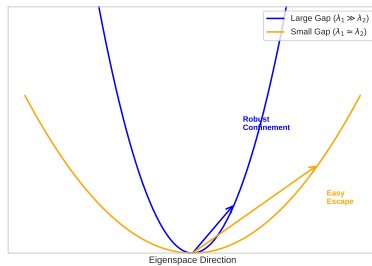

*(a)* **Norm-based Bifurcation (Theorem 4.2):** Visualizes the threshold phenomenon where the dynamics remain trapped in the pre-trained basin (stable) only if the LoRA perturbation norm is sufficiently small.

*(b)* **Depth-based Bifurcation (Theorem 4.6):** Depicts the "stability window" where perturbed tokens track the base model until a critical depth $T^*$, after which they diverge toward a new clustering geometry.

*(c)* **Spectral Control (Proposition 3.3):** Illustrates how the eigenvalue gap gap governs the curvature of the potential well; a larger gap enforces tighter confinement of the representation.

*Figure 1.* Schematic representation of the theoretical results.

- **Phase transitions and experiments.** We characterize (i) a norm-controlled transition for random adapters via a time-to-deviation estimate (Theorem 4.2) and (ii) a depth-controlled transition (Theorem 4.6), and we empirically verify our results with synthetic (Figures 5) and LLM-side evidence (Figures 2, 3, 4).

## Related Work

**LoRA fine-tuning** The increase in the number of LLM parameters makes fine-tuning increasingly costly. One way of reducing the cost is to reduce the number of parameters to be trained. The algorithm LoRA (Hu et al., 2022) developed for NLP applications proposes adapting pre-trained foundation model such as Llama (Touvron et al., 2023), Mistral (Jiang et al., 2023), by freezing all its weights and training attention matrices $\tilde{Q}, \tilde{K}$ and $\tilde{V}$ of low rank (see (4)). Since the introduction of LoRA fine-tuning (Hu et al., 2022), many variants have emerged (Dettmers et al., 2023; Wang et al., 2023), and we refer to surveys (Yang et al., 2024; Mao et al., 2025) for more details.

**Catastrophic forgetting** Catastrophic forgetting refers to performance regressions on previously acquired capabilities after adapting a pretrained model to new data. Recent empirical studies report substantial forgetting under instruction tuning and continual fine-tuning, and analyze it across knowledge, reasoning, and domain generalization benchmarks (Luo et al., 2025; Kotha et al., 2024; Li et al., 2024; Huang et al., 2024; Jiang et al., 2025). Within parameter-efficient fine-tuning, LoRA often reduces forgetting compared to full fine-tuning, but does not eliminate it (Biderman et al., 2024). Several recent works propose orthogonalization or projection constraints to reduce interference between the update subspace and dominant pretrained directions (Xiong & Xie, 2025; Wang et al., 2023), which aligns with the spectral-stability mechanisms highlighted by our

Proposition 3.3.

**Mean-field Transformers.** Self-attention acts on sets $\{x_1, \ldots, x_n\}$ in a permutation-equivariant manner, making measure-valued descriptions natural. A set of tokens can be represented as an empirical measure $\mu = \frac{1}{n} \sum_{i=1}^{n} \delta_{x_i}$, and attention can be viewed as a map on measures (De Bie et al., 2019; Vuckovic et al., 2020; Zweig & Bruna, 2021; Sander et al., 2022).

The clustering effect mathematically proved by the line of work on mean-field Transformers, (see (Rigollet, 2025) for a survey) is linked to the *signal propagation*, *rank-collapse* phenomena literature; see Dong et al. (2021); Feng et al. (2022); Noci et al. (2022); Joudaki et al. (2023); Zhao et al. (2023); Zhai et al. (2023); Noci et al. (2024); Bao et al. (2024). Empirically, the signal propagation is linked to the trainability of the neural networks (Cowsik et al., 2025). Some of the scaling laws derived by (Cowsik et al., 2025) have since been used in training large language models such as OLMO2 7B & 13B, (OLMo et al., 2024)).

## 2. Setup: Mean-Field Self-Attention

**Notation** We write $\langle \cdot | \cdot \rangle$ and $\|\cdot\|$ for the Euclidean inner product and norm on $\mathbb{R}^d$. The unit sphere is denoted by $\mathbb{S}^{d-1}$, and

$$\mathsf{P}_x := \mathrm{Id} - xx^\top$$

denotes the orthogonal projection onto $x^\perp$. For a probability measure $\mu$, we write $\mathrm{supp}(\mu)$ for its support. For a configuration $X = (x_1, \ldots, x_n)$, we denote its empirical measure by

$$\mu_X := \frac{1}{n} \sum_{i=1}^{n} \delta_{x_i}. \tag{5}$$

Untilded quantities refer to the base model, while tilded quantities refer to the LoRA-perturbed model. Thus

$$A := K^\top Q, \qquad \tilde{A} := \widetilde{K}^\top \widetilde{Q},$$

so that $\langle Qx, Ky \rangle = \langle Ax, y \rangle$. When $V = V^\top \succeq 0$, we denote its eigenvalues by

$$\lambda_1 > \lambda_2 \geq \cdots \geq \lambda_d$$

and choose an orthonormal eigenbasis $(u_1, \ldots, u_d)$. We write

$$\text{gap} := \lambda_1 - \lambda_2$$

for the spectral gap.

**Self-Attention dynamics.  Non-normalized dynamics.**
Let $(K, Q, V) \in (\mathbb{R}^{d \times d})^3$ be a triple of attention matrices. Consider an initialization of tokens $(x_1, \ldots, x_n) \in (\mathbb{R}^d)^n$. *Without a normalization layer*, tokens evolve through

$$\dot{x}_i(t) = \sum_{j=1}^n \frac{\exp\left(\langle Qx_i(t), Kx_j(t) \rangle\right)}{\sum_{k=1}^n \exp\left(\langle Qx_i(t), Kx_k(t) \rangle\right)} V x_j(t),$$

$$x_i(0) = x_i. \tag{6}$$

Equation (6) corresponds to the forward pass of tokens through layers of trained Transformers without non-linearities or normalization. Previous work has shown that the dynamics (6) can diverge under general assumptions (Geshkovski et al., 2023). We can normalize (6) by considering the variables $z_i(t) := e^{-tV} x_i(t)$, which satisfy the dynamics

$$\dot{z}_i(t) = \sum_{j=1}^n \frac{\exp\left(\langle Qe^{tV}z_i(t), Ke^{tV}z_j(t) \rangle\right)}{\sum_{k=1}^n \exp\left(\langle Qe^{tV}z_i(t), Ke^{tV}z_k(t) \rangle\right)} V(z_j - z_i). \tag{7}$$

**Post-Layer normalization dynamics.** Another way to prevent divergence is to add a normalization layer. In the case of *Post-LayerNorm*, the tokens evolve on the sphere; given an initialization $(x_1, \ldots, x_n) \in (\mathbb{S}^{d-1})^n$, they follow

$$\dot{x}_i(t) = \mathsf{P}_{x_i(t)} \sum_{j=1}^n \frac{\exp\left(\langle Ax_i(t), x_j(t) \rangle\right)}{\sum_{k=1}^n \exp\left(\langle Ax_i(t), x_k(t) \rangle\right)} V x_j(t),$$

$$x_i(0) = x_i. \tag{8}$$

The models given by Eq. (8) and Eq. (7) are known in the literature as *interacting particle systems*, and are reminiscent of the extensive literature on the synchronization of such systems (Kuramoto, 1975; Krause, 2000; Lu et al., 2019; Tadmor, 2023). In the rest of the paper, we use the terms particle and token interchangeably.

Writing $X(t) = (x_1(t), \ldots, x_n(t)) \in (\mathbb{S}^{d-1})^n$, we sometimes use the compact notation

$$\dot{x}_i(t) = \mathsf{P}_{x_i(t)} B_{\boldsymbol{V}, \boldsymbol{A}}[\mu_{X(t)}](x_i(t)), \qquad x_i(0) = x_i, \tag{9}$$

$$B_{\boldsymbol{V}, \boldsymbol{A}}[\mu](x) = \frac{1}{Z_{\boldsymbol{A}}[\mu](x)} \int e^{\beta \langle \boldsymbol{A}x, y \rangle} \boldsymbol{V} y \, \mu(\mathrm{d}y), \tag{10}$$

with normalizing constant

$$Z_{\boldsymbol{A}}[\mu](x) = \int e^{\beta \langle \boldsymbol{A}x, y \rangle} \mu(\mathrm{d}y).$$

In the special case $V = A$, the projected flow (8) can be interpreted as a (Riemannian) gradient flow of the interaction energy

$$\mathrm{E}_A(x_1, \ldots, x_n) := \sum_{i=1}^n \sum_{j=1}^n \exp\left(\langle Ax_i, x_j \rangle\right), \tag{11}$$

with respect to a suitable metric; see Geshkovski et al. (2025); Burger et al. (2025) for precise statements.

By the Stable manifold theorem, (ascending) gradient flows generically converge towards a local maximizer. This observation is used by Geshkovski et al. (2025); Criscitiello et al. (2024); Polyanskiy et al. (2025) to prove convergence towards the unique local maximizer $x_1 = \ldots = x_n$ when $V = I_d$. It is conjectured that the local maximizer of $\mathrm{E}_A$ is supported in the span of the eigenvectors associated with the largest eigenvalues, as suggested by Burger et al. (2025); Abella et al. (2025).

The dynamics given by Eq. (7) and Eq. (8) exhibit asymptotic clustering phenomena as proved in Geshkovski et al. (2023; 2025) under specific conditions. Additionally, we conducted experiments to identify a clustering phenomenon in Llama 2 (see Section E.5).

**Continuity equation (mean-field viewpoint).** Because self-attention is permutation-equivariant, it is natural to describe the evolution at the level of measures. Let $(\mu_t)_{t \geq 0}$ be defined as in Eq. (5). Define the mean-field attention vector field

$$\mathcal{X}[\mu](x) := \frac{\displaystyle\int_{\mathbb{R}^d} \exp\left(\langle Qx, Ky \rangle\right) Vy \, \mu(\mathrm{d}y)}{\displaystyle\int_{\mathbb{R}^d} \exp\left(\langle Qx, Ky \rangle\right) \mu(\mathrm{d}y)}. \tag{12}$$

Formally, $\mu_t$ solves the continuity equation

$$\partial_t \mu_t + \nabla \cdot (\mathcal{X}[\mu_t]\mu_t) = 0, \qquad \mu_{t=0} = \mu_0. \tag{13}$$

We refer to Section B.1 for well-posedness and additional details. The particle description is the *Lagrangian* viewpoint, while (13) is the corresponding *Eulerian* (measure-valued) formulation. From now on, we leverage both viewpoints as they offer complementary tools.

## 3. Stability Under Perturbations

This section examines how token evolution, starting from identical initial conditions, is influenced by variations in the attention matrix parameters.

### 3.1. Finite-Time Wasserstein Stability

We first present a general stability result in the Wasserstein metric.

**Proposition 3.1.** *Let $R > 0$, and let $\mu_0 \in \mathcal{P}_c(\mathbb{R}^d)$ be a probability measure with support in $B(0, R)$. Consider the vector fields $\chi$ and $\widetilde{\chi}$ defined in Eq. (12) with attention weights $(Q, K, V)$ and $(\widetilde{Q}, \widetilde{K}, \widetilde{V})$, respectively. Let $(\mu_t)_{t \geq 0}$ and $(\nu_t)_{t \geq 0}$ be the solutions to Eq. (13) associated with $\chi$ and $\widetilde{\chi}$. Then, for all $t \geq 0$,*

$$W_2(\mu_t, \nu_t)^2 \leq L_t(\Delta A, \Delta V) \cdot \exp\left(2C_t e^{3D_t}\right), \quad (14)$$

*where the constants are defined as:*

$$L_t = 6\left(\|\Delta V\|_{\text{op}}^2 \vee \|\Delta A\|_{\text{op}}\|V\|_{\text{op}}\right),$$
$$D_t = 2\|A\|_{\text{op}}\|V\|_{\text{op}}M_0^2 e^{2\|V\|_{\text{op}}t}t,$$
$$R_t = M_0 e^{\max\{\|V\|_{\text{op}}, \|\widetilde{V}\|_{\text{op}}\}t},$$
$$C_t = 4\|V\|_{\text{op}}^2\|A\|_{\text{op}}^2 R_t^2\left(1 + e^{2R_t\|A\|_{\text{op}}}\right)^2 t.$$

Note that if $\|\Delta V\|_{\text{op}} \leq \varepsilon$ and $\|\Delta A\|_{\text{op}} \leq \varepsilon$, then for a constant $c > 0$ depending on $(A, V)$, Eq. (14) yields

$$W_2(\mu_t, \nu_t) \leq c\,\varepsilon\,e^{c\,e^{ct}}.$$

This result highlights the robustness of the model against small perturbations in parameters over short time scales. A similar estimate holds for the solution of Eq. (8). However, the bound (14) grows doubly exponentially with time $t$.

### 3.2. Long-Time Stability via Spectral Structure

To obtain tighter stability guarantees that persist over long times, we must account for the geometry of the low-rank perturbations. We focus here on the *Post-LayerNorm* setting; since tokens are normalized, the Wasserstein distance is uniformly bounded. We seek to characterize perturbations that preserve the representation geometry in the infinite-time limit.

Let $(V, Q, K)$ be a triple of matrices such that $A = K^\top Q = V \succeq 0$. The analysis in Burger et al. (2025) suggests that the local maxima of the energy are supported in the span of the eigenvectors associated with the largest eigenvalues. Our main insight is a precise characterization of the stability of these equilibria under LoRA dynamics.

To facilitate our analysis, we adopt the following assumption regarding the spectral[2] properties of $V$.

**Assumption 3.2.** *Let $V \in \mathbb{R}^{d \times d}$ be a positive semi-definite matrix ($V \succeq 0$) with eigenvalues $(\lambda_i)_{i=1}^d$ satisfying $\lambda_1 > \lambda_2 > \ldots > \lambda_d$. Furthermore, assume the initial tokens*

---

[2]We examined the eigenvalue distributions of the attention matrices in BERT and Llama 2 (see Appendix E.4).

$(x_1, \ldots, x_n)$ *satisfy $\langle x_i, u_1 \rangle \geq \gamma > 0$, where $u_1$ is the eigenvector associated with $\lambda_1$.*

This assumption is standard in the consensus literature; see e.g., Abella et al. (2025). In the following proposition, we analyze the LoRA setting by modeling the update as $\Delta V = \Delta A$. Choose $\tilde{u}_1$ be the leading normalized largest eigenvector of $\Delta V$ such that $\langle u_1, \tilde{u}_1 \rangle \geq 0$.

**Proposition 3.3.** *Let $V \in \mathbb{R}^{d \times d}$ and $(x_1, \ldots, x_n)$ satisfy Assumption 3.2. Let $(X(t))_{t \geq 0} \subset (\mathbb{S}^{d-1})^n$ denote the Post-LayerNorm dynamics (8) driven by $V$, and let $(\widetilde{X}(t))_{t \geq 0} \subset (\mathbb{S}^{d-1})^n$ be the corresponding LoRA dynamics. Suppose that $\Delta V \in \text{Sym}(d)$. Define $a := u_1^\top \Delta V u_1$, $b := P_\perp \Delta V u_1$ and $E := P_\perp \Delta V P_\perp$. If*

$$\text{gap} + a > 2\|b\| + \|E\|_{\text{op}}. \quad (15)$$

*Then,*

$$X(t) \underset{t \to \infty}{\to} (u_1, \ldots, u_1) \quad \text{and} \quad \widetilde{X}(t) \underset{t \to \infty}{\to} (\widetilde{u}_1, \ldots, \widetilde{u}_1).$$

*Furthermore, we have*

$$\|u_1 - \widetilde{u}_1\| \lesssim \frac{2\|b\| + \|E\|_{\text{op}}}{\text{gap} + a}. \quad (16)$$

This result suggests that the eigengap $\text{gap}$ plays a key role in stability. The LoRA update space can be decomposed into stable and unstable directions: the component along $\text{span}(u_1)$ (determined by $a$) can reinforce or degrade stability, while orthogonal components (captured by $E$) act as perturbations bounded by the gap. We empirically verify this gap in pre-trained models; see Figure 7.

*Remark* 3.4. We can refine the above result by decomposing the matrix $\Delta V$ into block matrices. We introduce $E : u_1^\perp \to u_1^\perp$ is the restriction of $\Delta V$ to $u_1^\perp$, and $\Lambda$ is the restriction of $V$ to $u_1^\perp$. Assume that $E$ and $\Lambda$ commutes, then for large $t$ we have

$$\|X(t) - \widetilde{X}(t)\|^2 \simeq \sum_{j:\ e_j \neq 0}\left(\frac{\alpha_j}{\lambda_1 - \lambda_j - e_j(r)}\right)^2, \quad (17)$$

where $\alpha_j := \langle \Delta V u_1, u_j \rangle$. where $(e_j(r))_{j=1}^d$ are the eigenvalues of $E$. This result highlights that higher-rank LoRA updates can induce larger forgetting when they align with eigenspaces corresponding to smaller spectral gaps. The proof is analogous to the one presented, and relies on the introduced decomposition.

Our result motivates constraining LoRA updates to avoid directions already utilized by the pre-trained weights—for instance, by projecting the LoRA perturbation onto the orthogonal complement of the learned subspace. This idea has been explored in recent orthogonalized variants of LoRA (Xiong & Xie, 2025; Wang et al., 2023) and is supported by the stability mechanism in Eq. (15), (16) and (17).

# 4. Phase Transitions in Representation Drift

In this section, we characterize the bifurcation between the pre-trained behavior and the perturbed dynamics. We investigate the impact of the LoRA norm in Section 4.1 and of the depth in Section 4.2.

## 4.1. Phase Transition With Respect to LoRA Norm

### 4.1.1. THEORETICAL RESULT

In this subsection, we consider LoRA weights being random adapters (i.e. i.i.d drawn from $\rho \in \mathcal{P}(\mathbb{R}^{d \times d} \times \mathbb{R}^{d \times d})$). For the sake of clarity, we assume $V = A$, and we only consider LoRA perturbations for the Values matrix $\Delta V$ i.e.

$$\widetilde{A}^\ell := A, \quad \widetilde{V}^\ell := V + \Delta V^\ell.$$

Recall that the original dynamics associated to $V$ is converging (under Assumption 3.2) to a cluster $X_* = (u_1, \ldots, u_1)$ where $u_1$ is defined in 3.2. We now inspect how the random LoRA updates are changing the dynamic.

Let $L$ be the depth and let $\ell \in \{0, \ldots, L-1\}$ denote the layer index. Following the formalism of Koubbi et al. (2026), considering the Post-LayerNorm setting, the (normalized) token iterate for a transformers of depth $L$ is

$$\begin{aligned} x_i^{\ell+1} &= \mathrm{N}\left(x_i^\ell + \frac{1}{L} B_{\widetilde{V}^\ell, \widetilde{A}^\ell}[\mu_{X^\ell}](x_i^\ell)\right), \quad (18) \\ x_i^0 &= u_1 + \delta_i \in \mathbb{S}^{d-1}, \end{aligned}$$

where $B$ is defined in (10) and N denotes the normalization on the sphere i.e. $\mathrm{N}(x) = \frac{x}{\|x\|}$.

According to computations in Section 2.2 in Koubbi et al. (2026), since the increments are centered and independent, their typical scale after $L$ iterates is of order $\frac{\sqrt{L\mathrm{Var}(\Delta V)}}{L}$, so in order to the LoRA weights affect the dynamics, we need to have $\|\Delta V^\ell\| \simeq \sqrt{L}$. Otherwise, the transformers iterates are similar to the original dynamics.

**Assumption 4.1.** Let $(\Delta V^\ell)_{\ell \in \mathbb{N}}$ be i.i.d random variables drawn from a common distribution $\rho \in \mathcal{P}(\mathbb{R}^{d \times d})$ such that

$$(\Delta V^\ell)_{\ell=1}^L \overset{i.i.d}{\sim} \eta_L \sum_{a=1}^r s_a u_a^\ell (v_a^\ell)^\top, \quad (19)$$

where $(u_a^\ell, v_a^\ell) \overset{i.i.d}{\sim} \mathcal{N}(0, \frac{I_d}{d})$, $s_a > 0$ and $\eta_L > 0$.

The next theorem proves that under this noise scaling, the LoRA dynamics is confined in a small neighborhood of $X^*$ up to a critical value of the noise. We state here an informal version of the theorem proved in appendix.

**Theorem 4.2.** *Under Assumptions 3.2, and 4.1, and $\|\delta\| \ll 1$*

- *If $\eta_L \ll \sqrt{L}$, then*

$$\mathbb{E}[\|x_i^\ell - u_1\|^2] \lesssim O(1/L). \quad (20)$$

- *If $\eta_L = \sqrt{L}$, then for $\ell$ large enough, and $\kappa$ small enough*

$$\sum_{i=1}^n \mathbb{E}\|x_i^\ell - u_1\|^2 \simeq n \sum_{i=2}^d \frac{\kappa}{2(\lambda_1 - \lambda_i) + d\kappa} + O(1/L), \quad (21)$$

*where $\kappa := \frac{1}{d^2} \sum_{a=1}^r s_a^2$ is the magnitude of the noise.*

Equations (20) and (21) reveal a scaling transition governed by the size of the random LoRA perturbation:

- If $\eta_L \ll \sqrt{L}$, the cumulative random effect of the LoRA updates vanishes in the large-depth limit, and the dynamics remain trapped near $X_\star$.

- If $\eta_L \simeq \sqrt{L}$, the accumulated perturbation has a non-trivial diffusive effect. In this regime, the dynamics are confined, in mean square, to a neighborhood of $X_\star$ whose size is controlled by

$$\kappa \sum_{i=1}^d \frac{1}{2(\lambda_1 - \lambda_i) + d\kappa}.$$

This expression highlights the stabilizing role of the spectral gaps: directions with smaller gaps contribute more strongly to the deviation.

- If $\eta_L \gg \sqrt{L}$, the perturbative homogenization argument no longer applies. We expect the random LoRA fluctuations to dominate the pretrained drift, potentially leading to synchronization around a strongly random moving direction, but this regime is outside the scope of the theorem.

Thus, the term *phase transition* should be understood as a transition between a perturbative regime, where random LoRA updates average out across depth, and a diffusive regime, where their accumulated effect survives in the continuum-depth limit. The proof further shows that, in the critical scaling $\eta_L \simeq \sqrt{L}$, the tokens synchronize around a common random direction; see Figure 8 for an illustration. This synchronization mechanism is not specific to LoRA and is related in spirit to recent stochastic-synchronization phenomena studied in (Agazzi et al., 2026; Engel & Shalova, 2026).

The apparent rank-independence in the random-adapter experiment is consistent with the rotational invariance of the Gaussian model in Assumption 4.1, provided the perturbations are compared at fixed Frobenius norm, or equivalently fixed noise magnitude $\kappa$. In that case, the rank mainly

changes how the energy of the perturbation is distributed across random directions, but not its average orientation relative to the pretrained spectral structure. By contrast, learned LoRA updates are not isotropic (see Figure 4). A natural way to model rank-dependent forgetting would be to replace the isotropic Gaussian assumption by an anisotropic covariance, allowing the LoRA updates to preferentially align with unstable or weakly stable eigendirections of the pretrained dynamics.

### 4.1.2. EMPIRICAL EVIDENCE

We first test our results regarding the evolution of the per­turbed loss with **random adapters**. In Figure 2, we observe a phase transition for the perturbed loss with respect to the perturbation norm. This is consistent with Theorem 4.2. Notice that forgetting is independent of the rank in this observation, likely due to the rotational invariance of the Gaussian distribution.

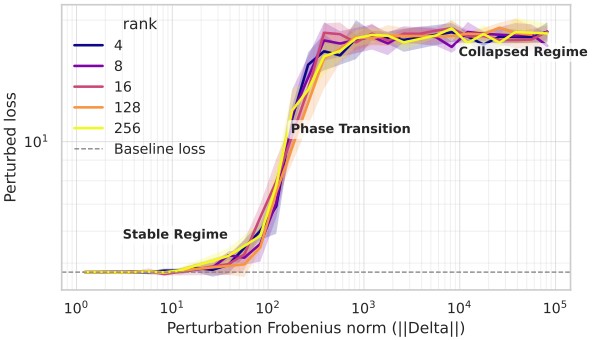

*Figure 2.* Evolution of the loss when adding Gaussian matrices of rank $r \in \{4, 8, 16, 128, 256\}$ with respect to the Frobenius norm of the perturbation. The perturbed loss is measured by next-token prediction on a base dataset distinct from the fine-tuning dataset. The base model is Qwen 3 0.6B (Yang et al., 2025).

**Extrapolation to training**  Our theorems rely on perturba­tive methods and might initially seem irrelevant for under­standing forgetting during training. However, empirically (see Figure 3), we observe that the norm remains a crucial factor. Note that the rank does play a role here, a phe­nomenon our toy model does not fully capture, except via the spectral alignment intuition in Proposition 3.3.

**Geometric Alignment with Stable Directions**  To under­stand *why* forgetting occurs, we analyzed the geometry of the learned LoRA updates. To connect Proposition 3.3 with trained Transformers, we measure whether learned LoRA updates align with the dominant spectral directions of the pretrained value matrices. Since the value projections $V^\ell$ of a real Transformer are not necessarily symmetric, we define $u_1^\ell$ as the top right singular vector of $V^\ell$, equivalently the leading eigenvector of $(V^\ell)^\top V^\ell$. This vector plays the role

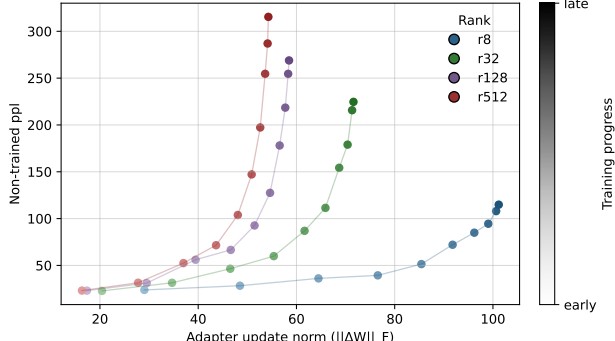

*Figure 3.* Evolution of evaluation perplexity during LoRA training for different ranks $r \in \{4, 8, 16, 128, 256\}$. The perturbed loss is next-token prediction on a base dataset distinct from the fine-tuning dataset.

of the stable direction $u_1$ in the symmetric toy model.

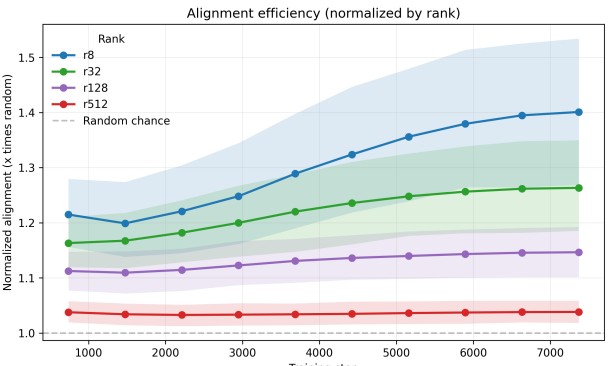

*Figure 4.* **Alignment with the Stable Direction** ($u_1$). Evolution of normalized subspace alignment during training. Values $> 1.0$ indicate that the LoRA update targets the base model's principal feature ($u_1$) more than a random vector would. Low-rank adapters ($r = 8, 32$) show a tendency to interfere with this stable direction.

For each layer $\ell$, we consider the learned LoRA update

$$\Delta V^\ell = (V_A^\ell)^\top V_B^\ell.$$

We then measure the alignment between $u_1^\ell$ and the input subspace of the update, namely the row space of $\Delta V^\ell$:

$$\text{Align}_\ell := \frac{d}{r} \left\| \Pi_{\text{Row}(\Delta V^\ell)} u_1^\ell \right\|^2 .$$

The normalization is chosen so that $\text{Align}_\ell = 1$ in expecta­tion for a uniformly random $r$-dimensional subspace of $\mathbb{R}^d$. Values larger than 1 therefore indicate that the LoRA update is more aligned with the dominant pretrained direction than a random update of the same rank.

As shown in Figure 4, alignment scores are consistently higher than the random baseline ($y = 1.0$). Notably, lower-rank adapters ($r = 8, 32$) exhibit a strong "locking on"

effect, where alignment with $u_1$ increases during training. This confirms that LoRA updates are not isotropic; they selectively interfere with the dominant features of the pre-trained model. This interference drives the representation away from its initial cluster, precipitating the observed forgetting. (See Appendix E.3 for details).

## 4.2. Phase Transition With Respect to Depth Scaling

We now investigate bifurcations with respect to the depth of the neural network. We work under the following clustering hypothesis.

**Assumption 4.3** (Clustering hypothesis). Let $(Q, K, V)$ be a triple of attention matrices. For the initial configuration under consideration, assume that there exists a finite set

$$\mathcal{C} = \{c_1, \dots, c_k\} \subset \mathbb{R}^d$$

such that, for every $i \in \{1, \dots, n\}$, there exists $j \in \{1, \dots, k\}$ satisfying $z_i(t) \xrightarrow[t \to +\infty]{} c_j$.

Under this assumption, the original dynamics clusters for sufficiently large times. In particular, when $V = I_d$ and $A := Q^\top K \succeq 0$, the triple $(Q, K, V)$ satisfies Assumption 4.3; see Theorem 3.1 of Geshkovski et al. (2023).

**Definition 4.4.** Let $\mathcal{C} = \{c_1, \dots, c_k\}$ be the limiting cluster set associated with the initial configuration. For $\delta > 0$, define

$$S_\delta(t) := \{i \in \{1, \dots, n\} : \mathrm{d}(z_i(t), \mathcal{C}) \leq \delta\}.$$

For the modified dynamics, we similarly define $\widetilde{S}_\delta(t)$. Since the pre-trained dynamics clusters, for every $\delta > 0$ we set

$$T_\delta := \inf \{T \geq 0 : \forall t \geq T, \quad S_\delta(t) = \{1, \dots, n\}\}.$$

For each limiting cluster $c_a \in \mathcal{C}$, we denote by

$$I_a := \left\{ i \in \{1, \dots, n\} : z_i(t) \xrightarrow[t \to +\infty]{} c_a \right\}$$

the set of original tokens converging to $c_a$. We say that $c_a$ is occupied if $I_a \neq \varnothing$.

The following assumption identifies a direction along which the modified value matrix creates an instability.

**Assumption 4.5.** Let $\widetilde{V}$ be diagonalizable with real spectrum. Assume that the leading eigenvalue is simple and dominant:

$$\lambda_1 > |\lambda_2| \geq \cdots \geq |\lambda_d|, \qquad \mathrm{gap} := \lambda_1 - |\lambda_2| > 0.$$

Set $\ell := \varphi_1^*$. We assume the following conditions.

1. The leading attention coefficient is positive:

$$c_{11} := \langle Q\varphi_1, K\varphi_1 \rangle > 0.$$

2. There exists a unique occupied cluster maximizing the projection along $\ell$. Namely, there exists an occupied cluster $c_+$ such that

$$\ell(c_+) > \ell(c) \qquad \text{for every occupied cluster } c \neq c_+.$$

We define the maximality gap

$$D_+ := \ell(c_+) - \max_{\substack{c \neq c_+ \\ c \text{ occupied}}} \ell(c) > 0.$$

3. There exists an occupied non-maximal cluster $c_-$ such that $\ell(c_-) > 0$, $D_- := \ell(c_+) - \ell(c_-) > 0$. We set $m_* := \ell(c_-) > 0$.

4. There exist indices $i_- \in I_-$ and $j_+ \in I_+$ such that $\|\widetilde{z}_{i_-}(T_\delta) - c_-\| \leq 2\delta$, and $\|\widetilde{z}_{j_+}(T_\delta) - c_+\| \leq 2\delta$.

5. There exists $M > 0$ such that, as long as $\widetilde{S}_{2\delta}(t) = \{1, \dots, n\}$, one has

$$|\varphi_p^*(\widetilde{z}_i(t))| \leq M \qquad \forall p \in \{1, \dots, d\} \forall i \in \{1, \dots, n\}.$$

Define $C_{\mathrm{sub}} := M^2 \sum_{(p,q) \neq (1,1)} |\langle Q\varphi_p, K\varphi_q \rangle|$, and set

$$a_* := \frac{c_{11} m_* D_+}{4}, \qquad p_* := \frac{D_-}{2(D_- + 4M)}.$$

We define the first exit time from the original clustered regime by

$$T^*(\delta) := \inf \left\{ t \geq T_\delta : \widetilde{S}_{2\delta}(t) \neq \{1, \dots, n\} \right\}.$$

By convention, $T^*(\delta) = +\infty$ if the set above is empty.

**Theorem 4.6.** Let $(Q, K, V)$ satisfy Assumption 4.3.

1. **Stability for small depths.** For every $\delta > 0$, there exists $\eta(\delta) > 0$ such that, if $\|\widetilde{V} - V\|_{\mathrm{op}} \leq \eta(\delta)$, then

$$\forall t \in [T_\delta, 10T_\delta], \qquad \widetilde{S}_{2\delta}(t) = \{1, \dots, n\}.$$

2. **Bifurcation at larger depths.** Assume in addition that Assumption 4.5 holds. Then

$$T^*(\delta) \leq \max\{T_\delta, T_{\mathrm{dom}}(\delta)\} + \frac{16\|\ell\|\delta}{\lambda_1 D_-},$$

with

$$T_{\mathrm{dom}}(\delta) := \max \left\{ \frac{1}{\mathrm{gap}} \log \left( \frac{4C_{\mathrm{sub}}}{a_*} \right), \right.$$

$$\left. \frac{1}{2\lambda_1} \log \left( \frac{2}{a_*} \log \left( \frac{n}{p_*} \right) \right) \right\}.$$

An illustration of the result can be found in Figure 5. The proof can be found in Appendix D.3. It is clear from our findings that assigning a single time frame $T_\delta$ for cluster formation across different token initializations is not feasible.

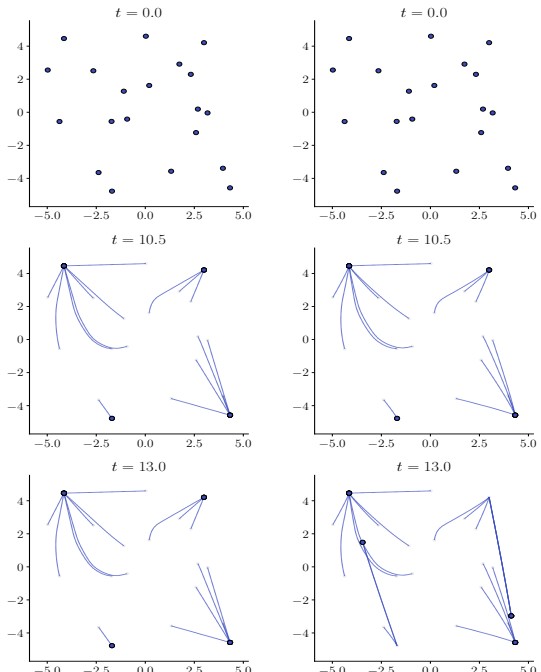

Figure 5. Illustration of Theorem 4.6 with $d = 2$ and $n = 20$. We have chosen an initialization of the tokens. On the first column, we display the dynamics of the tokens with $Q = K = V = I_2$ and in the second column, the one with $Q = K = I_2$ and $\tilde{V} = I_2 - \varepsilon \mathbf{e}_2 \mathbf{e}_2^T$ with $\varepsilon = 0.01$.

### 4.2.1. EXPERIMENTAL OBSERVATIONS

**Empirical bifurcation depth.** We now test the depth-controlled bifurcation predicted by Theorem 4.6 on a trained LoRA model. We use the Qwen 3 0.6B LoRA checkpoints from Figure 3. For each fine-tuning checkpoint $s$ and each layer $\ell$, we compare the hidden representations of the LoRA-adapted model with those of the frozen pretrained model on 512 held-out samples. Let $H_\ell^{\text{base}}$ denote the hidden states of the base model at layer $\ell$, and let $H_\ell^{\text{LoRA}}(s)$ denote the corresponding hidden states after $s$ LoRA fine-tuning steps. We define the relative Frobenius representation drift

$$\Delta_\ell(s) := \frac{\left\| H_\ell^{\text{LoRA}}(s) - H_\ell^{\text{base}} \right\|_F}{\left\| H_\ell^{\text{base}} \right\|_F}. \quad (22)$$

For a tolerance parameter $\tau > 0$, we then define the empirical bifurcation depth as

$$b_\tau(s) := \inf \left\{ \ell \in \{1, \dots, L\} \colon \Delta_\ell(s) > \tau \right\}, \quad (23)$$

with the convention that $b_\tau(s) = L$ if no layer exceeds the tolerance.

Figure 6 reports $b_\tau(s)$ as a function of the fine-tuning step $s$. The results are consistent with the stability-window picture of Theorem 4.6: as LoRA fine-tuning progresses, the first layer at which the adapted model significantly deviates from the pretrained model moves earlier in depth. For

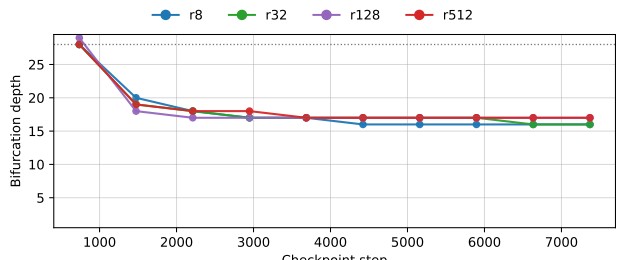

Figure 6. **Empirical bifurcation depth during LoRA fine-tuning.** For each checkpoint step $s$, we compute the relative $\ell_2$ representation drift $\Delta_\ell(s)$ between the LoRA-adapted model and the frozen pretrained model at each layer $\ell$. The empirical bifurcation depth $b_\tau(s)$ is defined as the first layer where $\Delta_\ell(s) > \tau$, using threshold $\tau = 0.2$. Curves correspond to LoRA ranks $r \in \{8, 32, 128, 512\}$. As fine-tuning progresses, $b_\tau(s)$ decreases, showing that the stability window of the pretrained representation shrinks over training.

$\tau = 0.2$, the average empirical bifurcation depth decreases from approximately layer 28 at the beginning of training to approximately layer 16 at the end of training. This indicates that the stability window of the pretrained representation progressively shrinks as the LoRA perturbation accumulates.

## 5. Discussion and Acknowledgements

The first theorem isolates the role of the perturbation norm, while the second explains how depth can amplify initially small perturbations until a bifurcation time. In particular, when the low-rank updates are random, forgetting is essentially insensitive to the rank: this is predicted by Theorem 4.2. By contrast, the training experiments in Figure 3 indicate that this rank-independence does not extrapolate to learned updates. A plausible interpretation is that optimization is not isotropic: LoRA updates are biased towards unstable directions, amplifying forgetting even when the Frobenius norm is controlled.

We thank an anonymous reviewer for asking us to derive the result obtained in Remark 3.4 and Figure 6. This work was performed using HPC resources from GENCI-IDRIS (Grant 20XX-AD011016341).

## Impact Statement

This manuscript details research aimed at furthering the field of Machine Learning. While there are broader societal impacts tied to our work, we do not believe it necessary to highlight specific immediate consequences in this document.

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

## A. Generalities on Optimal Transport

Optimal Transport is a theory that endows probability measures, and general measures with natural and meaningful distances called Wasserstein distances. We refer to the books (Villani et al., 2009; Santambrogio, 2015) for a general view of Optimal Transport from a theoretical point of view and to (Peyré & Cuturi, 2019; Chewi et al., 2025) for the computational aspects and application to data science.

**Definition A.1.** Coupling Let $(\mathcal{X}, d)$ a Polish metric space. Let $\mu, \nu$ two probability measures on $\mathcal{X}$ we say that $\pi \in \mathcal{P}(\mathcal{X} \times \mathcal{X})$ is a coupling between $\mu$ and $\nu$ if and only if the marginals of $\pi$ are $\mu$ and $\nu$. We denote $\Pi(\mu, \nu)$ the set of couplings between $\mu$ and $\nu$.

**Definition A.2.** (Wasserstein distances) Let $(\mathcal{X}, d)$ a Polish metric space, and let $p \in [1, \infty($. For any two probability measures $\mu, \nu$ on $\mathcal{X}$, the Wasserstein distance of order $p$ between $\mu$ and $\nu$ is defined by

$$W_p(\mu, \nu) = \left( \inf_{\pi \in \Pi(\mu, \nu)} \int_{\mathcal{X}} d(x, y)^p d\pi(x, y) \right)^{\frac{1}{p}}, \tag{24}$$

where the set $\Pi(\mu, \nu)$ is the set of couplings between $\mu, \nu$.

We often call $W_1$ the Kantorovich-Rubinstein distance. This distance has nice properties and satisfies a duality property.

**Proposition A.3.** *(Kantorovich-Rubinstein duality) Let $(\mathcal{X}, d)$ a Polish metric space, and let $p \in [1, \infty[$. For any two probability measures $\mu, \nu$ on $\mathcal{X}$,*

$$W_1(\mu, \nu) = \sup_{\|\phi\|_{Lip \leq 1}} \left\{ \int_{\mathcal{X}} \phi d\mu - \int_{\mathcal{X}} \phi d\nu \right\}. \tag{25}$$

**Lemma A.4.** *Let $p \geq 1$. Consider a measurable function $f : \mathcal{X} \to \mathcal{X}$, and two probability measures $\alpha, \beta \in \mathcal{P}(\mathcal{X})$. Then, it holds*

$$W_p(f_\sharp \alpha, f_\sharp \beta) \leq Lip(f) W_p(\alpha, \beta). \tag{26}$$

*Proof.* We remark that a coupling $\gamma \in \Pi(\alpha, \beta)$ induces a coupling $\gamma' \in \Pi(f_\sharp \alpha, f_\sharp \beta)$ defined as

$$\gamma'(A \times B) = \gamma \left( f^{-1}(A) \times f^{-1}(B) \right). \tag{27}$$

We deduce the following inequalities

$$
\begin{aligned}
W_p(f_\sharp \alpha, f_\sharp \beta)^p &= \inf_{\gamma' \in \Gamma(f_\sharp \alpha, f_\sharp \beta)} \int \|x - y\|^p d\gamma'(x, y) \\
&\leq \inf_{\gamma \in \Gamma(\alpha, \beta)} \int \|f(x) - f(y)\|^p d\gamma(x, y) \\
&\leq \mathrm{Lip}(f)^p \inf_{\gamma \in \Gamma(\alpha, \beta)} \int \|x - y\|^p d\gamma(x, y) \\
&\leq \mathrm{Lip}(f)^p W_p(\alpha, \beta)^p.
\end{aligned}
$$

It finishes the proof of the lemma. $\qquad \square$

**Lemma A.5.** *Let $p \geq 1$. Consider two measurable functions $\varphi, \psi \colon \mathcal{X} \to \mathcal{X}$, and a probability measure $\mu \in \mathcal{P}_p(\mathcal{X})$ such that $\varphi_\sharp \mu \in \mathcal{P}_p(\mathcal{X})$ and $\psi_\sharp \mu \in \mathcal{P}_p(\mathcal{X})$. Then, it holds*

$$W_p(\varphi_\sharp \mu, \psi_\sharp \mu) \leq \|\varphi - \psi\|_{L^p(\mu)}.$$

*Proof.* Recall that

$$W_p(\varphi_\sharp \mu, \psi_\sharp \mu)^p = \inf_{\pi' \in \Pi(\varphi_\sharp \mu, \psi_\sharp \mu)} \int \|x - y\|^p d\pi'(x, y).$$

Now consider the following coupling between $\varphi_\sharp \mu$ and $\psi_\sharp \mu$, defined by the relation

$$\pi'(B \times C) := \mu(\varphi^{-1}(B) \cap \psi^{-1}(C))$$

for every Borel sets $B, C \subset \mathcal{X}$. In other words, we set $\mathrm{d}\pi'(y, z) := \int_{\varphi^{-1}(y) \cap \psi^{-1}(z)} \mathrm{d}\mu$, and $\mathrm{d}\pi'(y, z) = 0$ if $\varphi^{-1}(y) \cap \psi^{-1}(z) = \emptyset$. With this definition of $\pi'$, we have

$$W_p(\varphi_\sharp \mu, \psi_\sharp \mu)^p \leq \int \|x - y\|^p \mathrm{d}\pi'(x, y)$$
$$= \int \|\varphi(x) - \psi(x)\|^p \mathrm{d}\mu(x).$$

$\square$

A simple consequence of the Jensen inequality implies the growth of Wasserstein distances.

**Proposition A.6.** *Let $(\mathcal{X}, d)$ a Polish metric space, and let $1 \leq p < q < +\infty$. For any two probability measures $\mu, \nu$ on $\mathcal{X}$,*

$$W_p(\mu, \nu) \leq W_q(\mu, \nu). \tag{28}$$

# B. General results

We denote by $\mathcal{P}_c(\mathbb{R}^d)$ the set of compactly supported probability measures on $\mathbb{R}^d$, and by $\mathcal{P}_2(\mathbb{R}^d)$ the set of probability measures $\mu$ on $\mathbb{R}^d$ having finite second moment: $\int_{\mathbb{R}^d} \|x\|^2 \mathrm{d}\mu(x) < +\infty$. Let $C^0(\mathbb{R}; \mathcal{P}_c(\mathbb{R}^d))$ denote the Banach space of continuous curves $\mathbb{R} \ni t \mapsto \mu(t) \in \mathcal{P}_c(\mathbb{R}^d)$. Here $\mathcal{P}_c(\mathbb{R}^d)$ is endowed with the weak topology, which coincides with the topology induced by the Wasserstein distance $W_p$ for any $p \in [1, +\infty)$.

## B.1. General definition

Let $\Omega \subset \mathbb{R}^d$ be a convex open set. Let $T \in \mathbb{R}_+$, and let a $v : [0, T] \times \Omega \to \mathbb{R}^d$ be a bounded vector field. We say that $v$ is Lipschitz continuous in space (uniformly) if it exists $L > 0$ such that

$$|v(t, x) - v(t, y)| \leq L|x - y| \quad \forall t \leq T, (x, y) \in \Omega. \tag{29}$$

We remind the reader of the Cauchy-Lipchitz theorem

**Theorem B.1.** *Given $T > 0$, let $v : [0, T] \times \mathbb{R}^d$ be a bounded vector field which is Lipschitz in space. For any $x \in \mathbb{R}^d$, there exists a unique trajectory $t \mapsto X_t(x)$ solving the Cauchy problem*

$$\begin{cases} \frac{\mathrm{d}}{\mathrm{d}t} X_t(x) &= v(X_t(x), t) \\ X_0(x) &= x. \end{cases} \tag{30}$$

*in the integral sense. The map $X : [0, T) \times \mathbb{R}^d \to \mathbb{R}^d$ is the flow of $v$ starting at time $0$.*

We can apply this theorem to the flow associated to the velocity field $x \mapsto \mathcal{X}[\mu](x)$, which gives the flow associated to the *attention-kernel* velocity-field, resulting that the empirical measure $\mu_t$ of the tokens is the pushforward of the flow of the initial empirical measure

$$\mu_t = X_{t\sharp}(\mu_0). \tag{31}$$

Besides, this measure $\mu_t$ is a solution to the continuity equation

$$\frac{\mathrm{d}}{\mathrm{d}t} \int_{\mathbb{R}^d} g(x) dX_{t\sharp}(\mu_0) = \frac{\mathrm{d}}{\mathrm{d}t} \int_{\mathbb{R}^d} g(X_t(x)) d\mu_0(x)$$
$$= \int_{\mathbb{R}^d} \frac{\mathrm{d}}{\mathrm{d}t} g(X_t(x)) d\mu_0(x)$$
$$= \int_{\mathbb{R}^d} \langle \nabla g(X_t(x)), \dot{X}_t(x) \rangle d\mu_0(x)$$
$$= \int_{\mathbb{R}^d} \langle \nabla g(X_t(x)), \mathcal{X}[\mu_t](X_t(x)) \rangle d\mu_0(x)$$
$$= \int_{\mathbb{R}^d} \langle \nabla g(x), \mathcal{X}[\mu_t](x) \rangle d\mu_t(x).$$

We have proved that the pushforward measure of the flow associated to the ODE is indeed a solution of the continuity equation. However, the notion of continuity equation is more general, even if in some cases it can be an equivalence (see the Proposition (B.5) below). The point of view associated with ODEs is called Lagrangian, whereas the point of view associated with measures is called Eulerian. We can often switch from one formulation to another, as each has its advantages.

As seen below, for compactness purposes regarding solutions to the continuity equation, we consider an additional property on the support of such curves, summarized by the following definition.

**Definition B.2** (Equi-compactly supported curves). The set $C^0_{\text{comp}}(\mathbb{R}; \mathcal{P}_c(\mathbb{R}^d))$ consists of all elements $\mu \in C^0(\mathbb{R}; \mathcal{P}_c(\mathbb{R}^d))$ such that for any $t_0, t_1 \in \mathbb{R}$, there exists a compact subset $\mathcal{K} \subset \mathbb{R}^d$ such that $\text{supp}(\mu(t)) \subset \mathcal{K}$ for any $t \in [t_0, t_1]$.

We will make use of the following notion of solution. We want to define the continuity equation

$$\begin{cases} \partial_t \mu + \text{div}(\mathcal{X}[\mu]\mu) = 0 & \text{in } (0, +\infty) \times \mathbb{R}^d \\ \mu_{|t=0} = \mu_0 & \text{in } \mathbb{R}^d, \end{cases} \tag{32}$$

when $\mathcal{X}[\mu]$ is the *attention kernel*

$$\mathcal{X}[\mu](x) := \frac{\displaystyle\int_{\mathbb{R}^d} e^{\langle Qx, Ky \rangle} Vy \mathrm{d}\mu(y)}{\displaystyle\int_{\mathbb{R}^d} e^{\langle Qx, Ky \rangle} \mathrm{d}\mu(y)}. \tag{33}$$

**Definition B.3.** Fix $\mu_0 \in \mathcal{P}_c(\mathbb{R}^d)$. We say that $t \mapsto \mu(t) =: \mu_t$ is a solution to the Cauchy problem (32) if $\mu \in C^0_{\text{comp}}(\mathbb{R}, \mathcal{P}_c(\mathbb{R}^d))$, the function

$$\mathbb{R} \ni t \mapsto \int_{\mathbb{R}^d} g(x) \mathrm{d}\mu_t(x)$$

is absolutely continuous for every $g \in C^\infty_c(\mathbb{R}^d)$, and

$$\int_{\mathbb{R}^d} g(x) \mathrm{d}\mu_t(x) = \int_{\mathbb{R}^d} g(x) \mathrm{d}\mu_0(x) + \int_0^t \int_{\mathbb{R}^d} \langle \nabla g(x), \mathcal{X}[\mu_t](x) \rangle \, \mathrm{d}\mu_s(x) \mathrm{d}s.$$

We will remind some preliminaries concerning continuity equation, and about the construction of its solution as a the pushforward measure of a certain vector field. All these results are borrowed from Ambrosio et al. (2005).

**Lemma B.4.** *(Ambrosio et al., 2005)(Lemma 8.1.4) Let $\mathcal{X}[\mu]$ be a Borel vector field such that for every compact set $B \subset \mathbb{R}^d$, such that for every compact set $B \subset \mathbb{R}^d$, we have*

$$\int_0^T \left( \sup_B |\mathcal{X}[\mu_t]| + Lip(\mathcal{X}[\mu_t], B) dt \right) < +\infty. \tag{34}$$

*Then for every $x \in \mathbb{R}^d$, and $s \in [0, T]$ the ODE*

$$X_s(x, s) = x, \quad \frac{\mathrm{d}}{\mathrm{d}t} X_t(x, s) = \mathcal{X}[\mu_t](X_t(x, s)). \tag{35}$$

*admits a unique maximal solution in an interval $I(x, s)$. Besides if we have*

$$S \stackrel{\bullet}{=} \int_0^T \left( \sup_{\mathbb{R}^d} |\mathcal{X}[\mu_t]| + Lip(\mathcal{X}[\mu_t], \mathbb{R}^d) dt \right) < +\infty. \tag{36}$$

*Then, the flow map satisfies*

$$\sup_{t, s \in [0, T]} Lip\left(X_t(\cdot, s), \mathbb{R}^d\right) \leq e^S. \tag{37}$$

**Proposition B.5** ( (Ambrosio et al., 2005)[Proposition 8.1.8). *] Let $v_t$ be a Borel velocity field satisfying (34), and . Let $\mu_0 \in \mathcal{P}_c(\mathbb{R}^d)$, and let $X_t$ be the maximal solution of the ODE (35). Then, $t \mapsto \mu_t = (X_t)_\sharp \mu_0$ is a continuous solution of (B.3) and this solution is the unique globally defined solution. Moreover, if*

$$\int_0^T \int_{\mathbb{R}^d} |v_t(x)|^p d\mu_t(x) dt < +\infty \quad \text{for some } p > 1. \tag{38}$$

*Then, the velocity field $v_t$ is the time derivative of $X_t$ in the $L^p$ sense i.e.*

$$\lim_{h \to 0} \int_0^{T-h} \int_{\mathbb{R}^d} \left| \frac{X_{t+h}(x) - X_t(x)}{h} - v_t(X_t(x)) \right| d\mu_0(x) dt = 0 \tag{39}$$

**Lemma B.6.** *Consider the Kernel attention defined by Eq.*(12).*The ODE solution defined by* (35) *admits a unique maximal solution, and $\mu_t = (X_t)_\sharp \mu_0$ is the unique solution to* (B.3).

### B.2. Control of particles

In this subsection, we state some results concerning some global results about the rescaled particles behavior. These results are taken from Geshkovski et al. (2023). These results allow some global control over particles, and highlight some monotonicity properties of token trajectories which are crucial in the following.

**Lemma B.7.** *Suppose $k \in [d]$ is such that $\lambda_k \geq 0$. Then $t \mapsto \max_{j \in [n]} \varphi_k^*(z_j(t))$ and $t \mapsto \max_{j \in [N]} \varphi_k^*(z_j(t))$ is a non-increasing and bounded function, and $t \mapsto \min_{j \in [n]} \varphi_k^*(z_j(t))$ and $t \mapsto \min_{j \in [N]} \varphi_k^*(z_j(t))$ is a non-decreasing and bounded function. In particular, $t \mapsto \varphi_k^*(z_i(t))$ is uniformly bounded as a function on $[0, +\infty)$ for any $i \in [n]$.*

*Proof.* For the sake of completeness, we reproduce the proof presented in Geshkovski et al. (2023). We focus on proving the results for the real part, and the results for the imaginary part follow analogously. For any $k \in [d]$ and any $t \geq 0$, set

$$\alpha_k(t) = \min_{j \in [n]} \varphi_k^*(z_j(t)), \qquad \beta_k(t) = \max_{j \in [n]} \varphi_k^*(z_j(t)).$$

Let $i \in [n]$ be an index such that $\alpha_k(t) = \varphi_k^*(z_i(t))$. Then we have

$$\frac{\mathrm{d}}{\mathrm{d}t} \varphi_k^*(z_i(t)) = \sum_{j=1}^n P_{ij}(t) \varphi_k^* \left( V(z_j(t) - z_i(t)) \right)$$

$$= \lambda_k \sum_{j=1}^n P_{ij}(t)(\varphi_k^*(z_j(t)) - \varphi_k^*(z_i(t))) \geq 0$$

where the last inequality stems from the fact that $\lambda_k \geq 0$ and the choice of index $i$. This proves that $\alpha_k(\cdot)$ is non-decreasing, as desired. Arguing similarly, one finds that $\beta_k(\cdot)$ is non-increasing. As a consequence, $\alpha_k(0) \leq \alpha_k(t) \leq \beta_k(t) \leq \beta_k(0)$ for any $t \geq 0$, which shows that $\alpha_k(\cdot)$ and $\beta_k(\cdot)$ are bounded. $\qquad \square$

**Corollary B.8.** *If $V$ only has real non-negative eigenvalues, then $z_i(\cdot) \in L^\infty([0, +\infty))$. all trajectories $t \mapsto z_i(t)$ are uniformly bounded in time.*

## C. Proofs of results in Section 3

### C.1. Bound on attention kernel with respect to attention matrices perturbations

We remind the results from concerning the attention kernels, and its properties regarding Lipschitz constants, boundness, and the norm of its gradients.

**Lemma C.1.** *(Lemma 6.5 from Geshkovski et al. (2023)) For any $R > 0$ there exists a constant $C_1(R) > 0$ such that for any $\mu, \nu \in \mathcal{P}_c(\mathbb{R}^d)$ with support in $B(0, R)$,*

$$\|\mathcal{X}[\mu]\|_{L^\infty(\mathbb{R}^d; \mathbb{R}^d)} \leq \|V\|_{\mathrm{op}} R, \tag{40}$$

$$\|\nabla_x \mathcal{X}[\mu]\|_{L^\infty(\mathbb{R}^d; \mathbb{R}^{d \times d})} \leq 2\|Q^\top K\|_{\mathrm{op}} \|V\|_{\mathrm{op}} R^2 \tag{41}$$

$$\|\mathcal{X}[\mu](\cdot) - \mathcal{X}[\nu](\cdot)\|_{L^\infty(B(0,R); \mathbb{R}^d)} \leq C_2(R) W_2(\mu, \nu). \tag{42}$$

*Besides, we can give a bound on the constant $C_2(R)$ which will be useful for the proofs*

$$C_2(R) \leq 2R^2 \|V\|_{\mathrm{op}} \|A\|_{\mathrm{op}} \left( 1 + e^{2R^2 \|A\|_{\mathrm{op}}} \right). \tag{43}$$

These inequalities allow us to define a unique solution to the continuity equation. This construction relies heavily on the fact that the Attention kernel are Lipschitz functions in the space of measures. We will need finest estimates of (41),

**Lemma C.2.** *Let* $(\tilde{V}, V)$ *a couple of attention matrices,* $\tilde{Q} = Q$ *and* $\tilde{K} = K$. *We denote* $\mathcal{X}(resp.\tilde{\mathcal{X}})$ *the* attention kernel *associated to the pre-trained matrices (resp. LoRA matrices). For any* $R > 0$ *there exists a constant* $C_1(R) > 0$ *such that for any* $\mu, \nu \in \mathcal{P}_c(\mathbb{R}^d)$ *with support in* $B(0, R)$, *then, we have the following bound*

$$\|\mathcal{X}[\mu] - \tilde{\mathcal{X}}[\nu]\|_{L^\infty(\mathbb{R}^d, \mathbb{R}^d)} \leq \|V - \tilde{V}\|_{\mathrm{op}} R + C_2(R) W_2(\mu, \nu) \tag{44}$$

*Proof.* Let $x > 0$, we have

$$
\begin{aligned}
\|\mathcal{X}[\mu](x) - \tilde{\mathcal{X}}[\nu](x)\| &\leq \|\mathcal{X}[\nu](x) - \tilde{\mathcal{X}}[\nu](x)\| + \|\mathcal{X}[\mu](x) - \mathcal{X}[\nu](x)\| \\
&\leq \left\| \int_{\mathbb{R}^d} \frac{e^{\langle Qx, Ky \rangle}}{\int_{\mathbb{R}^d} e^{\langle Qx, Ky \rangle} d\nu(y)} (V - \tilde{V}) y d\nu(y) \right\| + C_2(R) W_2(\mu, \nu) \\
&\leq \|V - \tilde{V}\|_{\mathrm{op}} \times \int_{\mathbb{R}^d} \frac{e^{\langle Qx, Ky \rangle}}{\int_{\mathbb{R}^d} e^{\langle Qx, Ky \rangle} d\nu(y)} \|y\| d\nu(y) + C_2(R) W_2(\mu, \nu) \\
&\leq \|V - \tilde{V}\|_{\mathrm{op}} R + C_2(R) W_2(\mu, \nu)
\end{aligned}
$$

where in the second line we have used the inequality (42) to bound the right term of the expression, at the third line we use the fact that $\nu$ is supported in $B(0, R)$ to bound the left term by $R$. $\qquad\square$

We can give a statement consisting by bounding differences between the attention kernels with respect to $Q, K$ variations.

**Lemma C.3.** *Let* $(Q, K)$ *and* $(\tilde{Q}, \tilde{K})$ *two couple of attention matrices, and let* $V$ *a Value matrix. We denote* $\mathcal{X}(resp.\tilde{\mathcal{X}})$ *the* attention kernel *associated to the pre-trained matrices (resp. LoRA matrices). For any* $R > 0$ *there exists a constant* $C_1(R) > 0$ *such that for any* $\mu, \nu \in \mathcal{P}_c(\mathbb{R}^d)$ *with support in* $B(0, R)$, *then, we have the following bound*

$$\|\mathcal{X}[\mu] - \tilde{\mathcal{X}}[\nu]\|_{L^\infty(\mathbb{R}^d, \mathbb{R}^d)} \leq 4R^3 \|V\|_{\mathrm{op}} \|A - \tilde{A}\|_{\mathrm{op}} + C_2(R) W_2(\mu, \nu). \tag{45}$$

*Proof.* First, we bound the difference between the two normalization constants $Z_A$ and $Z_{\tilde{A}}$.

$$
\begin{aligned}
|Z_A(x) - Z_{\tilde{A}}(x)| &= \left| \int_{B(0,R)} e^{\langle Ax, y \rangle} - e^{\langle \tilde{A}x, y \rangle} d\mu(y) \right| \\
&\leq \int_{B(0,R)} \left| e^{\langle Ax, y \rangle} - e^{\langle \tilde{A}x, y \rangle} \right| d\mu(y) \\
&\leq \int_{B(0,R) \cap \langle (\tilde{A}-A)x, y \rangle < 0} e^{\langle Ax, y \rangle} \left| 1 - e^{\langle (\tilde{A}-A)x, y \rangle} \right| d\mu(y) \\
&\quad + \int_{B(0,R) \cap \langle (\tilde{A}-A)x, y \rangle > 0} e^{\langle \tilde{A}x, y \rangle} \left| 1 - e^{\langle (A-\tilde{A})x, y \rangle} \right| d\mu(y) \\
&\leq \int_{B(0,R)} e^{\langle Ax, y \rangle} \left| \langle (\tilde{A} - A)x, y \rangle \right| d\mu(y) + \int_{B(0,R) \cap \langle (\tilde{A}-A)x, y \rangle > 0} e^{\langle \tilde{A}x, y \rangle} \left| \langle (\tilde{A} - A)x, y \rangle \right| d\mu(y) \\
&\leq \|\tilde{A} - A\|_{\mathrm{op}} \int_{B(0,R)} \left( e^{\langle Ax, y \rangle} + e^{\langle \tilde{A}x, y \rangle} \right) \|x\| \|y\| d\mu(y) \\
&\leq \|\tilde{A} - A\|_{\mathrm{op}} R^2 \left( Z_A(x) + Z_{\tilde{A}}(x) \right)
\end{aligned}
$$

where we used the mean value theorem at the third line to state that

$$1 - e^{\langle (\tilde{A}-A)x, y \rangle} = e^c \langle (\tilde{A} - A)x, y \rangle, \qquad c \in [\langle (\tilde{A} - A)x, y \rangle, 0], \tag{46}$$

and using the fact that $\langle (\tilde{A} - A)x, y \rangle$ is negative, then it yields that

$$1 - e^{\langle (\tilde{A}-A)x, y \rangle} \leq \left\langle (\tilde{A} - A)x, y \right\rangle. \tag{47}$$

So, we have proven

$$|Z_A(x) - Z_{\tilde{A}}(x)| \leq \|\tilde{A} - A\|_{\mathrm{op}} R^2 \left( Z_A(x) + Z_{\tilde{A}}(x) \right). \tag{48}$$

Now, we prove the bound between the two attention kernels. Let $x \in B(0, R)$, we have

$$
\begin{aligned}
\|\mathcal{X}[\mu](x) - \tilde{\mathcal{X}}[\mu](x)\| &= \left\| \int_{B(0,R)} \left( \frac{e^{\langle Ax,y \rangle}}{Z_A(x)} - \frac{e^{\langle \tilde{A}x,y \rangle}}{Z_{\tilde{A}}(x)} \right) Vy d\mu(y) \right\| \\
&\leq \|V\|_{\mathrm{op}} \int_{B(0,R)} \left| \frac{e^{\langle Ax,y \rangle}}{Z_A(x)} - \frac{e^{\langle \tilde{A}x,y \rangle}}{Z_{\tilde{A}}(x)} \right| \|y\| d\mu(y) \\
&\leq R\|V\|_{\mathrm{op}} \int_{B(0,R)} \left| \frac{e^{\langle Ax,y \rangle} Z_{\tilde{A}}(x) - e^{\langle \tilde{A}x,y \rangle} Z_A(x)}{Z_A(x) Z_{\tilde{A}}(x)} \right| d\mu(y) \\
&\leq \frac{R\|V\|_{\mathrm{op}}}{Z_A(x) Z_{\tilde{A}}(x)} \int_{B(0,R)} \left| e^{\langle Ax,y \rangle} \left( Z_{\tilde{A}}(x) - Z_A(x) \right) \right| \\
&\quad + \left| \left( e^{\langle Ax,y \rangle} - e^{\langle \tilde{A}x,y \rangle} \right) Z_A(x) \right| d\mu(y) \\
&\leq 2 \frac{R\|V\|_{\mathrm{op}}}{Z_A(x) Z_{\tilde{A}}(x)} \|\tilde{A} - A\|_{\mathrm{op}} R^2 \left( Z_A(x) + Z_{\tilde{A}}(x) \right) Z_A(x) \\
&\leq 2R^3 \|V\|_{\mathrm{op}} \|A - \tilde{A}\|_{\mathrm{op}} \frac{Z_A(x) + Z_{\tilde{A}}(x)}{Z_{\tilde{A}}(x)} \\
&\leq 2R^3 \|V\|_{\mathrm{op}} \|A - \tilde{A}\|_{\mathrm{op}} \left( 1 + \frac{\min\{Z_A(x), Z_{\tilde{A}}(x)\}}{\max\{Z_A(x), Z_{\tilde{A}}(x)\}} \right) \\
&\leq 4R^3 \|V\|_{\mathrm{op}} \|A - \tilde{A}\|_{\mathrm{op}}.
\end{aligned}
$$

We used the bound given by (48) for the right term in the line 4, and then we use symmetry concerning $A, \tilde{A}$ to deduce the bound at last line. To get the desired statement, we apply the triangular inequality and (42) $\qquad \square$

**Lemma C.4.** *Let $(Q, K, V)$ and $(\tilde{Q}, \tilde{K}, \tilde{V})$ two triple of attention matrices, and let $V$ a Value matrix. We denote $\mathcal{X} (\text{resp.} \tilde{\mathcal{X}})$ the* attention kernel *associated to the pre-trained matrices (resp. LoRA matrices). For any $R > 0$ there exists a constant $C_2(R) > 0$ such that for any $\mu, \nu \in \mathcal{P}_c(\mathbb{R}^d)$ with support in $B(0, R)$, then, we have the following bound*

$$\|\mathcal{X}[\mu] - \tilde{\mathcal{X}}[\nu]\|_{L^\infty(\mathbb{R}^d, \mathbb{R}^d)} \leq 2R^3 \min\{\|V\|_{\mathrm{op}}, \|\tilde{V}\|_{\mathrm{op}}\} \|A - \tilde{A}\|_{\mathrm{op}} + \|V - \tilde{V}\|_{\mathrm{op}} R + C_2(R) W_2(\mu, \nu). \tag{49}$$

*Proof.* We bound the quantity $\|\mathcal{X}[\mu](x) - \tilde{\mathcal{X}}[\nu](x)\|$ for every $x \in \mathbb{R}^d$ in the following way,

$$
\begin{aligned}
\|\mathcal{X}[\mu](x) - \tilde{\mathcal{X}}[\nu](x)\| &\leq \|\mathcal{X}[\nu](x) - \tilde{\mathcal{X}}[\nu(x)]\| + \|\mathcal{X}[\mu(x)] - \mathcal{X}[\nu(x)]\| \\
&\leq \left\| \int_{B(0,R)} \frac{e^{\langle Ax,y \rangle}}{Z_A(x)} Vy - \frac{e^{\langle \tilde{A}x,y \rangle}}{Z_{\tilde{A}}(x)} \tilde{V}y d\mu(y) \right\| + \|V\|_{\mathrm{op}} C_2(R) W_1(\mu, \nu) \\
&\leq \left\| \int_{\mathbb{R}^d} \frac{e^{\langle Ax,y \rangle}}{Z_A(x)} (V - \tilde{V}) y d\nu(y) \right\| + \left\| \int_{B(0,R)} \left( \frac{e^{\langle Ax,y \rangle}}{Z_A(x)} - \frac{e^{\langle \tilde{A}x,y \rangle}}{Z_{\tilde{A}}(x)} \right) \tilde{V}y d\mu(y) \right\| + C_2(R) W_2(\mu, \nu) \\
&\leq \|V - \tilde{V}\|_{\mathrm{op}} R + 4R^3 \|\tilde{V}\|_{\mathrm{op}} \|A - \tilde{A}\|_{\mathrm{op}} + C_2(R) W_2(\mu, \nu)
\end{aligned}
$$

Where in the third line, we used the lemmas (C.3) and (C.2). We remark that this inequality is symmetric to get the fact that we can take the minimum of the two bounds. $\qquad \square$

### C.2. Bound on the divergence between two dynamics with general perturbation

We now seek to establish a bound between two different dynamics initialized with same tokens, or let say a measure which is $\mu_0$ but with different attention matrices. We denote $V, Q$ and $K$ (resp.$\tilde{V}, \tilde{Q}$ and $\tilde{K}$) the three attention matrices of the pre-trained matrices (resp. LoRA matrices). Especially, we are interested in the setting in which $\tilde{V}, \tilde{Q}$ and $\tilde{K}$ are low rank

perturbations of the pre-trained parameters. Motivated by the LoRA paradigm, we want to enlight a bound between the two dynamics, and seek to find how many layers are necessary to distinguish between the tokens representation of the pre-trained dynamics and one of the LoRA dynamics.

One thing to notice first, is that we know because of the result of Geshkovski et al. (2023) that in the $t \to +\infty$ limit, the behavior between the clustering of tokens are radically different (see for instance the figures (7) or (5)). However, the rate of convergence towards the clusters is not clear. This work is focusing on these questions.

We first prove a more general result which only need a uniform bound on the two different attention kernels with respect with two different measures.

**Proposition C.5.** *Let $R > 0$, $\mathcal{X}, \tilde{\mathcal{X}}$ two attention kernels. Suppose $\exists C_1(R), C_2(R) > 0$ such that $\forall \mu, \nu \in \mathcal{P}_c(\mathbb{R}^d)$ supported in $B(0, R)$,*

$$\left\| \mathcal{X}[\mu(t)] - \tilde{\mathcal{X}}[\nu(t)] \right\|_{L^\infty(\mathbb{R}^d, \mathbb{R}^d)} \le C_1(R) + C_2(R) W_2(\mu, \nu). \tag{50}$$

*Then, $\forall (\mu_t)_{t \ge 0}, (\nu_t)_{t \ge 0}$ solutions of*

$$\partial_t \mu + \operatorname{div}(\mathcal{X}[\mu]\mu) = 0 \qquad \mu(0) = \mu_0, \tag{51}$$

$$\partial_t \nu + \operatorname{div}(\tilde{\mathcal{X}}[\nu]\nu) = 0 \qquad \nu(0) = \mu_0, \tag{52}$$

*we have*

$$\sup_{s \le t} W_2(\mu_s, \nu_s)^2 \le 2 C_1(R_t)^2 \times e^{2C(t)e^{3K_t}}, \tag{53}$$

*forall $t \ge 0$ where we define the following quantities*

$$C(t) = \int_0^t C_2(R_s)^2 ds, \tag{54}$$

$$K_t = 2\|Q^\top K\|_{\mathrm{op}} \|V\|_{\mathrm{op}} M_0^2 e^{2\|V\|_{\mathrm{op}}t} t, \tag{55}$$

$$R_t = M_0 e^{\max\{\|V\|_{\mathrm{op}}, \|\tilde{V}\|_{\mathrm{op}}\}t}. \tag{56}$$

*Proof.* The two measures $\mu_t$ and $\nu_t$ are solutions of the continuity equations defined by

$$\partial_t \mu + \operatorname{div}(\mathcal{X}[\mu]\mu) = 0 \quad \forall (t, x) \in \mathbb{R}_+ \times \mathbb{R}^d \qquad \mu(0) = \mu_0, \tag{57}$$

$$\partial_t \nu + \operatorname{div}(\tilde{\mathcal{X}}[\nu]\nu) = 0 \quad \forall (t, x) \in \mathbb{R}_+ \times \mathbb{R}^d \qquad \nu(0) = \mu_0. \tag{58}$$

Besides, by definition we have the existence of the two ODEs $(X_t)$ and $(Y_t)$ such that we have

$$\mu_t = (X_t)_\sharp \mu_0 \quad \text{and} \quad \nu_t = (Y_t)_\sharp \mu_0, \tag{59}$$

defined for every $x \in \mathbb{R}^d$, and $s \in [0, T]$ by

$$X_s(x, s) = x, \quad \frac{\mathrm{d}}{\mathrm{d}t} X_t(x, s) = \mathcal{X}[\mu_t](X_t(x, s)), \tag{60}$$

and,

$$Y_s(x, s) = x, \quad \frac{\mathrm{d}}{\mathrm{d}t} Y_t(x, s) = \tilde{\mathcal{X}}[\nu_t](Y_t(x, s)). \tag{61}$$

We decompose the proof into two steps

- First, we will prove a priori bound on the trajectory of the tokens, to establish a bound on the diameter of the support of the empirical measures depending on time.

- Then, we will use a Grönwall argument, to bound the Wasserstein distance between $(\mu_t)_{t \in [0,T]}$ and $(\nu_t)_{t \in [0,T]}$.

Suppose that $(x_1(t), \ldots, x_n(t))_{t \in [0,T]}$ is a system of tokens driven by the system of ODEs defined by

$$\frac{\mathrm{d}x_i(t)}{\mathrm{d}t} = \sum_{j=1}^{n} \frac{e^{\langle Qx_i(t), Kx_j(t) \rangle}}{\sum_{k=1}^{n} e^{\langle Qx_i(t), kx_k(t) \rangle}} V x_j(t). \tag{62}$$

So, we have that

$$
\begin{aligned}
\frac{1}{2} \frac{\mathrm{d}}{\mathrm{d}t} \max_{i \in [\![1,n]\!]} \|x_i(t)\|^2 &= \left\langle \frac{\mathrm{d}}{\mathrm{d}t} \max_{i \in [\![1,n]\!]} x_i(t), \max_{i \in [\![1,n]\!]} x_i(t) \right\rangle \\
&= \left\langle \sum_{j=1}^{n} \frac{e^{\langle Qx_i(t), Kx_j(t) \rangle}}{\sum_{k=1}^{n} e^{\langle Qx_i(t), kx_k(t) \rangle}} V x_j(t), \max_{i \in [\![1,n]\!]} x_i(t) \right\rangle \\
&\leq \sum_{j=1}^{n} \frac{e^{\langle Qx_i(t), Kx_j(t) \rangle}}{\sum_{k=1}^{n} e^{\langle Qx_i(t), kx_k(t) \rangle}} \langle V x_j, x_i \rangle \\
&\leq \|V\|_{\mathrm{op}} \max_{i \in [\![1,n]\!]} \|x_i(t)\|^2.
\end{aligned}
$$

Using Grönwall lemma, we deduce that for all $t > 0$, we have

$$\max_{i \in [\![1,n]\!]} \|x_i(t)\| \leq \max_{i \in [\![1,n]\!]} \|x_i(0)\| e^{\|V\|_{\mathrm{op}} t}. \tag{63}$$

So, by considering $V$ such that $V \in \arg\max\{\|V\|_{\mathrm{op}}, \|\tilde{V}\|_{\mathrm{op}}\}$, and by considering the initialization $M_0$ which have the maximal norm, we have that $\mu(t)$ and $\nu(t)$ is supported in the ball of radius $R_t = M_0 e^{\|V\|_{\mathrm{op}} t}$.

We have the bound on the Wasserstein distance of $\mu_t$ and $\nu_t$ by using the lemma (A.4)

$$W_2^2 \left( \Phi_{\mathcal{X}[\mu_t]}^t \sharp \mu_0, \Phi_{\tilde{\mathcal{X}}[\nu_t]}^t \sharp \mu_0 \right) \leq \int_{\mathbb{R}^n} \left\| \Phi_{\mathcal{X}[\mu_t]}^t(x) - \Phi_{\tilde{\mathcal{X}}[\nu_t]}^t(x) \right\|^2 d\mu_0(x). \tag{64}$$

Using the inequality (42) on the Lipschitz constant of $\mathcal{X}[\mu]$, and the bound on $R_t$, we get

$$\|\nabla x \mathcal{X}[\mu]\| \leq 2 \|Q^\top K\|_{\mathrm{op}} \|V\|_{\mathrm{op}} M_0^2 e^{2\|V\|_{\mathrm{op}} t} \overset{\bullet}{=} K_t, \tag{65}$$

which allows us to give a bound on the Lipschitz constant of $\mathcal{X}[\mu_s]$ up to time $s = t$, by $K_t$ as it is (65). We now want to bound the right term of (64). Define $r_t(x) = \Phi_{\mathcal{X}[\mu]}^t(x) - \Phi_{\tilde{\mathcal{X}}[\nu]}^t(x)$, we suppose that $r_t(x)$ is different of 0. We are willing to bound the norm of $r_t(x)$

$$
\begin{aligned}
\frac{\mathrm{d}}{\mathrm{d}t} \|r_t(x)\| &\leq \|\dot{r}_t(x)\| \\
&\leq \| \mathcal{X}[\mu_t] \left( \Phi_{\mathcal{X}[\mu_t]}^t(x) \right) - \tilde{\mathcal{X}}[\nu_t] \left( \Phi_{\tilde{\mathcal{X}}[\nu_t]}^t(x) \right) \| \\
&\leq \| \underbrace{\mathcal{X}[\mu_t] \left( \Phi_{\mathcal{X}[\mu_t]}^t(x) \right) - \tilde{\mathcal{X}}[\nu_t] \left( \Phi_{\mathcal{X}[\mu_t]}^t(x) \right)}_{(a)} \| \\
&\quad + \| \underbrace{\tilde{\mathcal{X}}[\nu_t] \left( \Phi_{\mathcal{X}[\mu_t]}^t(x) \right) - \tilde{\mathcal{X}}[\nu_t] \left( \Phi_{\tilde{\mathcal{X}}[\nu_t]}^t(x) \right)}_{(b)} \| \\
&\leq C_1(R_t) + C_2(R_t) W_2(\mu_t, \nu_t) + K_t \| \Phi_t^{\tilde{\mathcal{X}}[\nu_t]} - \Phi_t^{\mathcal{X}[\mu_t]} \|
\end{aligned}
$$

where we used reversed triangle inequality at first line, and the inequality (50) for $(a)$ on the different between two attention kernel and we used the inequality (65) on to bound $(b)$. So, we get

$$\frac{\mathrm{d}}{\mathrm{d}t} \|r_t(x)\| \leq K_t \|r_t(x)\| + C_1(R_t) + C_2(R_t) W_2(\mu, \nu). \tag{66}$$

Now, by using the Grönwall lemma in its differential form, we get the following inequality using the growth of $K_t$,

$$\|r_t(x)\| \leq \int_0^t (C_1(R_s) + C_2(R_s)W_2(\mu_s, \nu_s))e^{K_t(t-s)}ds. \tag{67}$$

We can re-inject the bound given by (67) in the expression (64), so we have for all $u \leq t$

$$
\begin{aligned}
W_2^2(\mu_u, \nu_u) = W_2^2\left(\Phi_{\mathcal{X}[\mu_u]}^u \sharp \mu_0, \Phi_{\tilde{\mathcal{X}}[\nu_u]}^u \sharp \mu_0\right) &\leq \int_{\mathbb{R}^n} \|r_u(x)\|^2 d\mu_0(x) \\
&\leq \int_{\mathbb{R}^n} \left(\int_0^u (C_1(R_s) + C_2(R_s)W_2(\mu_s, \nu_s)) e^{K_u(u-s)}ds\right)^2 d\mu_0(x) \\
&\leq t \int_{\mathbb{R}^n} \int_0^u (C_1(R_s)^2 + C_2(R_s)W_2(\mu_s, \nu_s))^2 e^{2K_u(u-s)}ds d\mu_0(x) \\
&\leq 2te^{2K_t t} \int_0^u C_1(R_s)^2 + (C_2(R_s)W_2(\mu_s, \nu_s))^2 \, ds \\
&\leq 2te^{2K_t t} \int_0^u C_1(R_t)^2 + C_2(R_t)^2 W_2(\mu_s, \nu_s)^2 ds
\end{aligned}
$$

We have used Cauchy-Schwarz inequality at the second line, at the third line we used the inequality $a + b \geq 2\sqrt{ab}$ for $(a, b) \in \mathbb{R}_+$, and at the last line, we have made use of $e^{K_t} \geq t$. Now, if we consider the function $f(t) = W_2(\mu_t, \nu_t)^2$, and make use of Grönwall lemma in its integral form, we get the inequality

$$W_2(\mu_t, \nu_t)^2 \leq 2C_1(R_t)^2 \times e^{2C(t)e^{3K_t}}, \tag{68}$$

where $C(t) = \int_0^t C_2(R_s)^2 ds$, we have used the fact that $C(t)e^{3K_t t} \geq e^{3K_t}$. It yields the desired inequality. □

*Remark* C.6. We notice that this proof ensures a double exponential bound which is quite relaxed. This inequality ensures that the two trajectories are close until a time $t \ll \log\log(\varepsilon)$.

*Remark* C.7. The bound (68) can also be used to bound the difference between the two trajectories of the normalized trajectories $(z_i(t))_{i \in [\![1,n]\!]}$ and $(\tilde{z}_i(t))_{i \in [\![1,n]\!]}$. We consider the two measures $\tilde{\mu}_t = \frac{1}{n}\sum_{i=1}^n \delta_{z_i(t)}$, and $\tilde{\nu}_t = \frac{1}{n}\sum_{i=1}^n \delta_{\tilde{z}_i(t)}$. We denote $|\lambda_1| \geq \cdots \geq |\lambda_n|$ the eigenvalues of $V$. We use (A.4) so we have the following inequality concerning the Wasserstein distance between $\tilde{\nu}_t$ and $\tilde{\mu}_t$

$$W_2(\tilde{\mu}_t, \tilde{\nu}_t) \leq \text{Lip}\left(e^{-tV}\right) W_2(\mu_t, \nu_t) \leq e^{-t|\lambda_n|} W_2(\mu_t, \nu_t). \tag{69}$$

*Remark* C.8. Another idea is to use the continuity equation directly for the rescaled token dynamics. The resulting vector field is no longer autonomous and is given by

$$\mathcal{X}[\mu](x) = \frac{\int_{\mathbb{R}^d} e^{\langle Qx, Ky\rangle} V(y-x)d\mu(y)}{\int_{\mathbb{R}^d} e^{\langle Qx, Ky\rangle} d\mu(y)}. \tag{70}$$

In the following, we will present the application of this proposition for different perturbations of attention matrices. Especially, we are interested in the low rank perturbations of the Attention matrices which is at the core of the LoRA paradigm. The propostion has two interesting corollaries, which are the following

**Corollary C.9.** *Perturbation with respect to parameters $V$*

*Let $R > 0$, and $\mu_0 \in \mathcal{P}_c(\mathbb{R}^d)$ with support in $B(0, R)$. We suppose that $Q, K \in M_d(\mathbb{R})$. We define the attention kernels in the following way*

$$\mathcal{X}[\mu](x) = \frac{\int_{\mathbb{R}^d} e^{\langle Qx, Ky\rangle} V y d\mu(y)}{\int_{\mathbb{R}^d} e^{\langle Qx, Ky\rangle} d\mu(y)}, \quad \text{and} \quad \tilde{\mathcal{X}}[\mu](x) = \frac{\int_{\mathbb{R}^d} e^{\langle Qx, Ky\rangle} \tilde{V} y d\mu(y)}{\int_{\mathbb{R}^d} e^{\langle Qx, Ky\rangle} d\mu(y)}. \tag{71}$$

*We define the two different dynamics on measures $(\mu_t)_{t\geq 0}$ and $(\nu_t)_{t\geq 0}$*

$$\partial_t \mu + \operatorname{div}(\mathcal{X}[\mu]\mu) = 0 \quad \forall (t,x) \in \mathbb{R}_+ \times \mathbb{R}^d \qquad \mu(0) = \mu_0, \tag{72}$$

$$\partial_t \nu + \operatorname{div}(\tilde{\mathcal{X}}[\nu]\nu) = 0 \quad \forall (t,x) \in \mathbb{R}_+ \times \mathbb{R}^d \qquad \nu(0) = \mu_0. \tag{73}$$

*Then, for all $t \geq 0$, we have the following bound*

$$W_2(\mu_t, \nu_t)^2 \leq 2\|V - \tilde{V}\|_{op}^2 R_t^2 \times e^{2C(t)e^{3K_t t}}. \tag{74}$$

$C_2(R)$ *is a constant depending on $R$ where we defined $C(t), K_t$ and $R_t$ as respectively* (54),(55) *and* (56).

*Proof.* Using the result of the lemma (49) for the low-rank perturbations of the Values attention matrices, we have for all $\mu, \nu$ which are supported in $B(0, R)$

$$\|\mathcal{X}[\mu] - \tilde{\mathcal{X}}[\nu]\|_{L^\infty(\mathbb{R}^d, \mathbb{R}^d)} \leq \|V - \tilde{V}\|_{op} R + C_1(R) W_2(\mu, \nu).$$

Applying the proposition (C.5), and noticing that we can bound $R_s$ by $R_t$ for all $t \geq s$, then we obtain the desired bound. $\quad\square$

**Proposition C.10.** *Perturbation with respect to parameters $Q, K$*

*Let $R > 0$, and $\mu_0 \in \mathcal{P}_c(\mathbb{R}^d)$ with support in $B(0, R)$. We suppose that*

- *Let $V \in M_d(\mathbb{R})$*

- *Let $Q(\text{resp. } \tilde{Q}), K(\text{resp. } \tilde{K}) \in M_d(\mathbb{R})$ we consider $A = K^T Q$ and $\tilde{A} = \tilde{Q}\tilde{K}^T$.*

*We define the attention kernels in the following way*

$$\mathcal{X}[\mu](x) = \frac{\displaystyle\int_{\mathbb{R}^d} e^{\langle Qx, Ky\rangle} V y \mathrm{d}\mu(y)}{\displaystyle\int_{\mathbb{R}^d} e^{\langle Qx, Ky\rangle} \mathrm{d}\mu(y)}, \quad \text{and} \quad \tilde{\mathcal{X}}[\mu](x) = \frac{\displaystyle\int_{\mathbb{R}^d} e^{\langle \tilde{Q}x, \tilde{K}y\rangle} V y \mathrm{d}\mu(y)}{\displaystyle\int_{\mathbb{R}^d} e^{\langle \tilde{Q}x, \tilde{K}y\rangle} \mathrm{d}\mu(y)}. \tag{75}$$

*We define the two different dynamics on measures $(\mu_t)_{t\geq 0}$ and $(\nu_t)_{t\geq 0}$*

$$\partial_t \mu + \operatorname{div}(\mathcal{X}[\mu]\mu) = 0 \quad \forall (t,x) \in \mathbb{R}_+ \times \mathbb{R}^d \qquad \mu(0) = \mu_0, \tag{76}$$

$$\partial_t \nu + \operatorname{div}(\tilde{\mathcal{X}}[\nu]\nu) = 0 \quad \forall (t,x) \in \mathbb{R}_+ \times \mathbb{R}^d \qquad \nu(0) = \mu_0. \tag{77}$$

*Then, for all $t \geq 0$, we have the following bound*

$$W_2(\mu_t, \nu_t)^2 \leq 4R_t^3 \|V\|_{op} \|A - \tilde{A}\|_{op} \times e^{2C(t)e^{3K_t t}}. \tag{78}$$

$C_2(R)$ *is a constant depending on $R$.*

*Proof.* Let $R > 0$, let $\mu, \nu \in \mathcal{P}_c(\mathbb{R}^d)$ such that $\mu, \nu$ are supported in $B(0, R)$. We have (45) which gives that

$$\|\mathcal{X}[\mu] - \tilde{\mathcal{X}}[\mu]\|_\infty \leq 2R^2 \|V\|_{op} \|A - \tilde{A}\|_{op}. \tag{79}$$

Combining it with (42), we obtain

$$\|\mathcal{X}[\mu] - \tilde{\mathcal{X}}[\nu]\| \leq 2R^2 \|V\|_{op} \|A - \tilde{A}\|_{op} + 2R^2 \|V\|_{op} \|A\|_{op} \left(1 + e^{2R^2\|A\|_{op}}\right) W_2(\mu, \nu). \tag{80}$$

Now, we can apply the result of (C.5) to obtain the desired result. $\quad\square$

**Proposition C.11.** *General perturbation*

*Let $R > 0$, and $\mu_0 \in \mathcal{P}_c(\mathbb{R}^d)$ with support in $B(0, R)$. We suppose that $V$ (resp. $\tilde{V} \in M_d(\mathbb{R})$), $Q$ (resp. $\tilde{Q}$), $K$(resp. $\tilde{K}) \in M_d(\mathbb{R})$ we consider $A = K^T Q$ and $\tilde{A} = \tilde{Q}\tilde{K}^T$. We define the attention kernels in the following way*

$$\mathcal{X}[\mu](x) = \frac{\int_{\mathbb{R}^d} e^{\langle Qx, Ky \rangle} V y \mathrm{d}\mu(y)}{\int_{\mathbb{R}^d} e^{\langle Qx, Ky \rangle} \mathrm{d}\mu(y)}, \quad and \quad \tilde{\mathcal{X}}[\mu](x) = \frac{\int_{\mathbb{R}^d} e^{\langle \tilde{Q}x, \tilde{K}y \rangle} V y \mathrm{d}\mu(y)}{\int_{\mathbb{R}^d} e^{\langle \tilde{Q}x, \tilde{K}y \rangle} \mathrm{d}\mu(y)}. \tag{81}$$

*We define the two different dynamics on measures $(\mu_t)_{t \geq 0}$ and $(\nu_t)_{t \geq 0}$*

$$\partial_t \mu + \mathrm{div}(\mathcal{X}[\mu]\mu) = 0 \quad \forall (t, x) \in \mathbb{R}_+ \times \mathbb{R}^d \qquad \mu(0) = \mu_0, \tag{82}$$

$$\partial_t \nu + \mathrm{div}(\tilde{\mathcal{X}}[\nu]\nu) = 0 \quad \forall (t, x) \in \mathbb{R}_+ \times \mathbb{R}^d \qquad \nu(0) = \mu_0. \tag{83}$$

*Then, for all $t \geq 0$, we have the following bound*

$$W_2(\mu_t, \nu_t)^2 \leq \left( 2\|V - \tilde{V}\|_{op}^2 R_t^2 + 4R_t^3 \|V\|_{\mathrm{op}} \|A - \tilde{A}\|_{\mathrm{op}} \right) \times e^{2C(t)e^{3K_t t}} \tag{84}$$

*$C_2(R)$ is a constant depending on $R$. This can be further simplified as*

$$W_2(\mu_t, \nu_t)^2 \leq 6R_t^3 \left( \|V - \tilde{V}\|_{op}^2 \vee \|V\|_{\mathrm{op}} \|A - \tilde{A}\|_{\mathrm{op}} \right) \times e^{2C(t)e^{3K_t t}} \tag{85}$$

*Proof.* We can make the same reasoning as before by using the result (C.4), and use the Proposition (C.5) to conclude. □

### C.3. Bound on the divergence between two dynamics with different initializations

In this subsection, we recall a result from Piccoli & Rossi (2013) and we go into more detail about the proof, as we need an increase in the constant $C(R, t)$, which is missing in the original article. We consider the continuity equation arising from the dynamic of the empirical measure of the rescaled tokens i.e

$$\partial_t \mu_t + \mathrm{div}(\mathcal{X}[\mu_t]\mu_t) = 0, \tag{86}$$

where we defined the rescaled vector field

$$\mathcal{X}[\mu_t](x) = \int_{\mathbb{R}^d} \frac{e^{\langle Qe^{tV}x, Ke^{tV}y \rangle}}{\int_{\mathbb{R}^d} e^{\langle Qe^{tV}x, Ke^{tV}y \rangle} d\mu_t(y)} V(y - x) d\mu_t(y). \tag{87}$$

**Lemma C.12.** *Let $(Q, K, V)$ be a triple of attention matrices. We consider two probability measures $\mu_0, \nu_0 \in \mathcal{P}_c(\mathbb{R}^d)$, then*

$$W_2(\mu_t, \nu_t) \leq e^{2Lt} W_2(\mu_0, \nu_0), \tag{88}$$

*where $L$ is given by the Lipschitz constant of the function $x \mapsto \Phi_{\mathcal{X}[\mu]}^t(x)$ restricted to $x \in B(0, R)$.*

*Proof.* We just apply the lemma (A.4) to the function $x \mapsto \Phi_{\mathcal{X}[\mu]}^t(x)$, we need to compute the Lipschitz constant of this function. This is a classical argument that we detail to be self-contained.

$$\frac{\mathrm{d}}{\mathrm{d}t} \|\Phi_{\mathcal{X}[\mu]}^t(x) - \Phi_{\mathcal{X}[\mu]}^t(y)\| \leq \left\| \frac{\mathrm{d}}{\mathrm{d}t} \left( \Phi_{\mathcal{X}[\mu]}^t(x) - \Phi_{\mathcal{X}[\mu]}^t(y) \right) \right\|$$
$$\leq \|\mathcal{X}[\mu_t](\Phi^t(x)) - \mathcal{X}[\mu_t](\Phi^t(y))\|$$
$$\leq \mathrm{Lip}(\mathcal{X}[\mu_t])\|\Phi_{\mathcal{X}[\mu]}^t(x) - \Phi_{\mathcal{X}[\mu]}^t(y)\|$$

Then, by the Grönwall lemma, we have the following inequality

$$\|\Phi_{\mathcal{X}[\mu]}^t(x) - \Phi_{\mathcal{X}[\mu]}^t(y)\| \leq \|x - y\| e^{L(t)}, \tag{89}$$

where $L(t) = \int_0^t \mathrm{Lip}(\mathcal{X}[\mu_s]) ds$, so if we denote $L(t) = \sup_{s \in [0,t]} \mathrm{Lip}(\mathcal{X}[\mu_s])$, we have the inequality desired (88). □

### C.4. Proof of Proposition 3.3

*Proof.* The proof is divided into two lemmas.

**Lemma C.13** (Cone collapse). *Let $V$ satisfy Assumption 3.2. Let $u_1$ be the top eigenvector of $V$, and assume that*

$$\min_{i \in [\![1,n]\!]} \langle x_i(0), u_1 \rangle \geq \gamma > 0.$$

*Let $(x_1(t), \ldots, x_n(t))_{t \geq 0}$ be the solution of Eq. (8). Then, for every $i \in [\![1, n]\!]$,*

$$\|x_i(t) - u_1\| \xrightarrow[t \to +\infty]{} 0.$$

*Proof.* The proof follows the cone-collapse argument of (Geshkovski et al., 2025). For notational simplicity, write $u = u_1$ and define

$$\varphi(t) := \min_{i \in [\![1,n]\!]} \langle u, x_i(t) \rangle.$$

By assumption, $\varphi(0) \geq \gamma > 0$. At every point of differentiability of $\varphi$, choose an index $i_*(t)$ attaining the minimum and set

$$x_*(t) := x_{i_*(t)}(t).$$

Thus

$$\langle u, x_*(t) \rangle = \varphi(t), \qquad \langle u, x_j(t) \rangle \geq \varphi(t) \quad \text{for every } j.$$

Differentiating $\varphi$ along the flow gives

$$\frac{\mathrm{d}}{\mathrm{d}t} \varphi(t) = \sum_{j=1}^{n} \alpha_j(t) \langle u, P_{x_*(t)} V x_j(t) \rangle,$$

where

$$\alpha_j(t) := \frac{\exp(\langle V x_*(t), x_j(t) \rangle)}{\sum_{k=1}^{n} \exp(\langle V x_*(t), x_k(t) \rangle)}.$$

Since $P_{x_*} u = u - \langle x_*, u \rangle x_*$, we obtain

$$\frac{\mathrm{d}}{\mathrm{d}t} \varphi(t) = \sum_{j=1}^{n} \alpha_j(t) \left( \langle u, V x_j(t) \rangle - \langle x_*(t), u \rangle \langle x_*(t), V x_j(t) \rangle \right)$$

$$= \sum_{j=1}^{n} \alpha_j(t) \left( \lambda_1 \langle u, x_j(t) \rangle - \varphi(t) \langle x_*(t), V x_j(t) \rangle \right).$$

For any unit vectors $y, z$, expanding in an eigenbasis of $V$ and using Cauchy–Schwarz on the orthogonal component gives

$$\langle y, V z \rangle \leq \lambda_1 \langle y, u \rangle \langle z, u \rangle + \lambda_2 \|P_{u^\perp} y\| \, \|P_{u^\perp} z\|.$$

Applying this inequality with $y = x_*(t)$ and $z = x_j(t)$ yields

$$\langle x_*(t), V x_j(t) \rangle \leq \lambda_1 \varphi(t) \langle u, x_j(t) \rangle + \lambda_2 (1 - \varphi(t)^2),$$

because

$$\|P_{u^\perp} x_*(t)\| = \sqrt{1 - \varphi(t)^2}, \qquad \|P_{u^\perp} x_j(t)\| \leq \sqrt{1 - \varphi(t)^2}.$$

Plugging this bound into the derivative estimate, we get

$$\frac{\mathrm{d}}{\mathrm{d}t} \varphi(t) \geq \sum_{j=1}^{n} \alpha_j(t) \left[ \lambda_1 \langle u, x_j(t) \rangle - \lambda_1 \varphi(t)^2 \langle u, x_j(t) \rangle - \lambda_2 \varphi(t)(1 - \varphi(t)^2) \right]$$

$$= (1 - \varphi(t)^2) \left[ \lambda_1 \sum_{j=1}^{n} \alpha_j(t) \langle u, x_j(t) \rangle - \lambda_2 \varphi(t) \right].$$

Since $\langle u, x_j(t) \rangle \geq \varphi(t)$ for every $j$ and $\sum_j \alpha_j(t) = 1$, we obtain

$$\sum_{j=1}^{n} \alpha_j(t)\langle u, x_j(t) \rangle \geq \varphi(t).$$

Therefore

$$\frac{\mathrm{d}}{\mathrm{d}t}\varphi(t) \geq (\lambda_1 - \lambda_2)\varphi(t)(1 - \varphi(t)^2).$$

Writing gap $:= \lambda_1 - \lambda_2 > 0$, this gives

$$\frac{\mathrm{d}}{\mathrm{d}t}\varphi(t) \geq \mathrm{gap}\, \varphi(t)(1 - \varphi(t)^2)$$

at every point of differentiability of $\varphi$. By comparison with the solution of

$$\dot{y} = \mathrm{gap}\, y(1 - y^2), \qquad y(0) = \varphi(0),$$

we get

$$\varphi(t) \geq \frac{\varphi(0)}{\sqrt{\varphi(0)^2 + (1 - \varphi(0)^2)e^{-2\mathrm{gap}t}}} \xrightarrow[t \to +\infty]{} 1.$$

Hence, for every $i$,

$$\langle u, x_i(t) \rangle \geq \varphi(t) \to 1.$$

Since $\|x_i(t)\| = \|u\| = 1$, we conclude that

$$\|x_i(t) - u\|^2 = 2(1 - \langle u, x_i(t) \rangle) \xrightarrow[t \to +\infty]{} 0.$$

This proves the lemma. $\qquad\qquad\qquad\qquad\qquad\qquad\qquad\qquad\qquad\qquad\qquad\qquad\qquad\qquad\qquad\quad$ $\square$

We now bound the distance between the leading eigenvectors of the original and perturbed matrices.

**Lemma C.14** (Perturbation of the top eigenvector). *Let $V$ be symmetric, with simple top eigenvalue $\lambda_1$ and eigengap*

$$\mathrm{gap} := \lambda_1 - \lambda_2 > 0.$$

*Let $u_1$ be a corresponding unit eigenvector. Let*

$$\widetilde{V} = V + \Delta V,$$

*where $\Delta V$ is symmetric. Define*

$$P_\perp := I - u_1 u_1^\top, \qquad a := \langle \Delta V u_1, u_1 \rangle, \qquad b := P_\perp \Delta V u_1, \qquad E := P_\perp \Delta V P_\perp.$$

*Let $\widetilde{u}_1$ be the leading eigenvector of $\widetilde{V}$, chosen so that $\langle u_1, \widetilde{u}_1 \rangle \geq 0$. Then*

$$\|u_1 - \widetilde{u}_1\| \lesssim \frac{2\|b\| + \|E\|_{\mathrm{op}}}{\mathrm{gap} + a}.$$

*Proof.* Since $\Delta V$ is symmetric, we have the orthogonal decomposition

$$\Delta V = a u_1 u_1^\top + u_1 b^\top + b u_1^\top + E.$$

Indeed, this follows by decomposing both variables into their components along $\mathrm{span}(u_1)$ and $u_1^\perp$. Introduce the intermediate matrix

$$A := V + a u_1 u_1^\top.$$

The top eigenvector of $A$ is still $u_1$, and the corresponding eigengap is

$$\delta_A = (\lambda_1 + a) - \lambda_2 = \mathrm{gap} + a.$$

Moreover,

$$\widetilde{V} = A + R, \qquad R := u_1 b^\top + b u_1^\top + E.$$

We have the operator norm bound

$$\|R\|_{\mathrm{op}} \le 2\|b\| + \|E\|_{\mathrm{op}}.$$

By the Davis–Kahan theorem applied to $A$ and $\widetilde{V} = A + R$, if $\theta$ denotes the angle between $u_1$ and $\widetilde{u}_1$, then

$$\sin\theta \lesssim \frac{\|R\|_{\mathrm{op}}}{\delta_A} \le \frac{2\|b\| + \|E\|_{\mathrm{op}}}{\mathrm{gap} + a}.$$

Since $\widetilde{u}_1$ is chosen so that $\langle u_1, \widetilde{u}_1 \rangle \ge 0$, we have $\theta \in [0, \pi/2]$ and

$$\|u_1 - \widetilde{u}_1\| = \sqrt{2(1 - \cos\theta)} \le \sqrt{2}\sin\theta.$$

Therefore

$$\|u_1 - \widetilde{u}_1\| \lesssim \frac{2\|b\| + \|E\|_{\mathrm{op}}}{\mathrm{gap} + a}.$$

$\square$

We can now conclude the proof. By Lemma C.13, the dynamics associated with $V$ converges to the leading eigenvector $u_1$. Applying the same lemma to the perturbed matrix $\widetilde{V}$ shows that the perturbed dynamics converges to $\widetilde{u}_1$, provided the initial data remains in the same cone, namely

$$\min_i \langle x_i(0), \widetilde{u}_1 \rangle > 0.$$

This condition follows from the assumed cone condition with respect to $u_1$ and the perturbative estimate of Lemma C.14, as soon as $\Delta V$ is sufficiently small. Finally, Lemma C.14 gives the desired control of the distance between the limiting clusters:

$$\|u_1 - \widetilde{u}_1\| \lesssim \frac{2\|b\| + \|E\|_{\mathrm{op}}}{\mathrm{gap} + a}.$$

This proves the proposition. $\square$

# D. Proofs of results in Section 4

## D.1. Homogenization techniques

Here, we briefly explain how to adapt the Neural ODEs techniques to the case with i.i.d. weights. One can take advantage of the stochastic approximation formula to use the classical martingale method for proving convergence of stochastic approximation towards diffusion processes (Stroock & Varadhan, 2007; Ethier & Kurtz, 1986). One might hope to obtain an SDE of the form

$$\mathrm{d}X_i(t) = \sigma[\mu(t)](X_i(t))\mathrm{d}B_t, \tag{90}$$

where $\sigma[\mu](x)$ is a volatility function encoding the fluctuations, and $B_t$ is a Brownian motion. This is the content of the following homogenization result. We work under the following assumption

**Assumption D.1.** Whenever $(A, V) \sim \rho$, we assume that $A - \mathbb{E}A = WW'^\top$ is independent from $V$. Moreover, there exists $C > 0$ such that $\{(V - \mathbb{E}V)_{ij}\}_{ij}$ and $\{(W)_{ij}, (W')_{ij}\}_{ij}$ are independent subGaussian with variance proxy $C\sigma_V^2$ and $C\sigma_A^2$ respectively.

We use the notations of Koubbi et al. (2026). Let define the following Itô SDE $(X^\eta(t))_{t \ge 0} \in (\mathbb{S}^{d-1})^n$ defined as

$$\mathrm{d}x_i(t) = \mathsf{P}_{x_i(t)}\left(b_\rho[\mu(t)](x_i(t))\,\mathrm{d}t + \int_\Theta \xi_\theta[\mu(t)](x_i(t))\,W(\mathrm{d}\theta, \mathrm{d}t)\right)$$
$$- \frac{1}{2}x_i(t)\int_\Theta \left\|\mathsf{P}_{x_i(t)}\xi_\theta[\mu(t)](x_i(t))\right\|^2 \rho(\mathrm{d}\theta)\,\mathrm{d}t \tag{91}$$

for all $i \in [n]$, where we have introduced the attention-induced velocity field

$$B_\theta[\mu](x) = \frac{1}{Z_{\boldsymbol{A}}[\mu](x)} \int e^{\beta\langle \boldsymbol{A}x, y\rangle} \boldsymbol{V} y \, \mu(\mathrm{d}y), \tag{92}$$

with normalizing constant

$$Z_{\boldsymbol{A}}[\mu](x) = \int e^{\beta\langle \boldsymbol{A}x, y\rangle} \, \mu(\mathrm{d}y).$$

We then decompose the velocity field into its expectation and variance term i.e.

$$B_{\theta_h^\ell}[\mu](x) = b_\rho[\mu](x) + \xi_{\theta_h^\ell}[\mu](x), \qquad b_\rho[\mu](x) := \mathbb{E}_{\theta \sim \rho} B_\theta[\mu](x).$$

**Theorem D.2** (Theorem 1 (Koubbi et al., 2026)). *Under Assumption D.1, for any $\varphi \in C^4((\mathbb{S}^{d-1})^n)$ the following approximation holds uniformly until macroscopic time $t_L = \eta L$:*

$$\sup_{t \in [0, t_L]} \left| \mathbb{E}\varphi(X(t)) - \mathbb{E}\varphi(X^\eta(t)) \right| \le C e^{C t_L} \eta(t_L + 1) \max(1, \alpha), \tag{93}$$

*where $C \ge 1$ depends on $\|\varphi\|_{C^4}$ but not on $\eta, \alpha, L$.*

From now on, since the bound (93) ensures that the error with the SDE is vanishing as the the number of layer is large, we study the SDE since it will ensure that the desired result. For $R > 0$, define the exit time

$$\tau(R) := \inf\{\ell \ge 0 : \max_{i=\{1,\dots,n\}} \|x_i^\ell - u_1\| > R\}. \tag{94}$$

Consider the iterates stopped at the hitting time $\tau(R)$ i.e $(x^{\ell \wedge \tau(R)})_{\ell \in \{1,\dots,L\}}$ where $a \wedge b = \min\{a, b\}$.

### D.2. Proof of Theorem 4.2

**Assumption D.3.** Let $(\Delta V^\ell)_{\ell \in \mathbb{N}}$ be i.i.d random variables drawn from a common distribution $\rho \in \mathcal{P}(\mathbb{R}^{d \times d})$ such that for all $(i, j) \in \{1, \dots, d\} \times \{1, \dots, d\}$, we have

$$(\Delta V^\ell) \overset{i.i.d}{\sim} \sqrt{L} \sum_{i=1}^r s_i u_i v_i^\top, \tag{95}$$

where $(u_i, v_i) \overset{i.i.d}{\sim} \mathcal{N}(0, \frac{I_d}{d})$ and $s_i > 0$.

We state here a formal version of the theorem

**Theorem D.4.** *We have*

$$\left| \mathbb{E}\left[ \sum_{i=1}^n (1 - \langle x_i^\ell, x\rangle^2) - n \sum_{i=1}^d \frac{\kappa}{2(\lambda_1 + \frac{d\kappa}{2} - \lambda_i)} \right] \right|$$
$$\lesssim C e^{C t_L} \frac{1}{L}(t_L + 1) + (\kappa + \|\delta\| + \varepsilon^3) n \exp\left( -t\left( \text{gap} + \frac{d\kappa}{2} \right) \right) + \exp(-C_S(t))\|\delta\|,$$

*where*

$$\kappa := \frac{c(s)}{d^2}, \qquad c(s) := \sum_{a=1}^r s_a^2.$$

The proof is divided in few lemmas on the behavior of the Itô SDE, and the conclusion arises from using the above result on the homogenization of the self-attention process.

*Proof of Theorem 4.2.* Let introduce few notations. For $\varepsilon > 0$, define the local neighbourhood

$$\mathcal{U}_\varepsilon := \left\{ (\xi_1, \dots, \xi_n) \in (T_x \mathbb{S}^{d-1})^n : \max_{1 \le i \le n} \|\xi_i\| \le \varepsilon \right\}.$$

For $\xi \in \mathcal{U}_\varepsilon$, write

$$X_i = \alpha_i x + \xi_i, \qquad \alpha_i := \sqrt{1 - \|\xi_i\|^2},$$

it is equivalent to define $\xi_i = \mathsf{P}_x X_i$. In the following, we denote

$$Q(t) = \sum_{i=1}^n \|\xi_i(t)\|^2, \ S(t) = \sum_{i=1}^n \|\xi_i(t) - \bar{\xi}(t)\|^2, \text{ and } B(t) = \|\bar{\xi}(t)\|^2, \tag{96}$$

where $\bar{\xi}(t) = \frac{1}{n} \sum_{i=1}^n \xi_i(t)$.

**Lemma D.5.** *We have*

$$\mathbb{E}[S_{t \wedge \tau_\varepsilon}] \le e^{-C_S t} S(0) + n\varepsilon^2 \mathbb{P}(\tau_\varepsilon \le t). \tag{97}$$

*where $C_S := 2\lambda_1 + (d-2)\kappa - C(1+\kappa)\varepsilon$.*

*Proof.* According to derivations and results from Koubbi et al. (2026), the Itô SDE for the LoRA updates should be of the following form

$$\mathrm{d}x_i(t) = \mathsf{P}_{x_i(t)} b_\rho[\mu(t)](x_i(t))\mathrm{d}t + \mathsf{P}_{x_i(t)} \int_\Theta \Delta V m_{\boldsymbol{A}}[\mu(t)](x_i(t))W(\mathrm{d}\theta, \mathrm{d}t)$$
$$- \frac{1}{2}\left(\int_\Theta \|P_{x_i(t)} \Delta V m_{\boldsymbol{A}}[\mu(t)](x_i(t))\|^2 \rho(\mathrm{d}\theta)\right) x_i(t),$$

where we have introduced

$$m_{\boldsymbol{A}}[\mu](x) := \int_{\mathbb{S}^{d-1}} \frac{e^{\langle \boldsymbol{A}x, y\rangle}}{Z_{\boldsymbol{A}}[\mu](x)} y \mathrm{d}\mu(y), \quad Z_{\boldsymbol{A}}[\mu](x) = \int_{\mathbb{S}^{d-1}} e^{\langle \boldsymbol{A}x, y\rangle} \mathrm{d}\mu(y). \tag{98}$$

An application of the Itô formula gives that the evolution of $\xi_i(t)$ is given by

$$\mathrm{d}\xi_i(t) = \mathsf{P}_x \mathsf{P}_{x_i(t)} b_\rho[\mu(t)](x_i(t))\mathrm{d}t + \mathsf{P}_x \mathsf{P}_{x_i(t)} \int_\Theta \Delta V m_{\boldsymbol{A}}[\mu(t)](x_i(t))W(\mathrm{d}\theta, \mathrm{d}t)$$
$$- \frac{1}{2}\left(\int_\Theta \|P_{x_i(t)} \Delta V m_{\boldsymbol{A}}[\mu(t)](x_i(t))\|^2 \rho(\mathrm{d}\theta)\right) \xi_i(t)\mathrm{d}t.$$

By Itô formula once again, we have the following Itô decomposition

$$\mathrm{d}\langle \xi_i(t), \xi_j(t)\rangle = \langle \xi_i(t), \mathrm{d}\xi_j(t)\rangle + \langle \mathrm{d}\xi_i(t), \xi_j(t)\rangle + \int_\Theta \langle \mathsf{P}_x \mathsf{P}_{x_i(t)} \Delta V m_{\boldsymbol{A}}[\mu(t)](x_i(t)), \mathsf{P}_x \mathsf{P}_{x_j(t)} \Delta V m_{\boldsymbol{A}}[\mu(t)](x_j(t))\rangle \rho(\mathrm{d}\theta)$$

$$= \left( \langle \mathsf{P}_{x_i(t)} b_\rho[\mu(t)](x_i(t)), \xi_j(t)\rangle + \langle \mathsf{P}_{x_j(t)} b_\rho[\mu(t)](x_j(t)), \xi_i(t)\rangle \right.$$

$$- \frac{1}{2}\langle \xi_i(t), \xi_j(t)\rangle \int_\Theta \left( \|\mathsf{P}_x \mathsf{P}_{x_i(t)} \Delta V m_{\boldsymbol{A}}[\mu(t)](x_i(t))\|^2 + \int_\Theta \|\mathsf{P}_x \mathsf{P}_{x_j(t)} \Delta V m_{\boldsymbol{A}}[\mu(t)](x_j(t))\|^2 \right) \rho(\mathrm{d}\theta)$$

$$+ \int_\Theta \langle \mathsf{P}_x \mathsf{P}_{x_i(t)} \Delta V m_{\boldsymbol{A}}[\mu(t)](x_i(t)), \mathsf{P}_x \mathsf{P}_{x_j(t)} \Delta V m_{\boldsymbol{A}}[\mu(t)](x_j(t))\rangle \rho(\mathrm{d}\theta) \Big) \mathrm{d}t$$

$$+ \sqrt{\alpha} \int_\Theta \left( \langle \mathsf{P}_x \mathsf{P}_{x_i(t)} \Delta V m_i(t), \xi_j(t)\rangle + \langle \xi_i(t), \mathsf{P}_x \mathsf{P}_{x_j(t)} \Delta V m_j(t)\rangle \right) W(\mathrm{d}\theta, \mathrm{d}t).$$

Then, for deterministic vectors $m_i, m_j$ and deterministic projectors,

$$\int_\Theta \langle \mathsf{P}_x \mathsf{P}_{x_i} \Delta V m_i, \mathsf{P}_x \mathsf{P}_{x_j} \Delta V m_j\rangle \rho(\mathrm{d}\theta) = \frac{c(s)}{d^2}\langle m_i, m_j\rangle \operatorname{Tr}\left(\mathsf{P}_{x_i} \mathsf{P}_x \mathsf{P}_{x_j}\right).$$

Moreover, since $\mathsf{P}_x x_i = \xi_i$, one has the exact identity

$$\operatorname{Tr}\left(\mathsf{P}_{x_i} \mathsf{P}_x \mathsf{P}_{x_j}\right) = d - 1 - \|\xi_i\|^2 - \|\xi_j\|^2 + \langle x_i, x_j\rangle\langle \xi_i, \xi_j\rangle.$$

Therefore

$$\int_{\Theta} \langle \mathsf{P}_x \mathsf{P}_{x_i} \Delta V m_i, \mathsf{P}_x \mathsf{P}_{x_j} \Delta V m_j \rangle \rho(\mathrm{d}\theta) = \frac{c(s)}{d^2} \langle m_i, m_j \rangle \left( d - 1 - \|\xi_i\|^2 - \|\xi_j\|^2 + \langle x_i, x_j \rangle \langle \xi_i, \xi_j \rangle \right).$$

Similarly,

$$a_i(t) = \int_{\Theta} \|\mathsf{P}_{x_i(t)} \Delta V m_i(t)\|^2 \rho(\mathrm{d}\theta) = \frac{c(s)}{d^2}(d-1)\|m_i(t)\|^2.$$

Plugging these identities into the previous lemma gives the closed form

$$\begin{aligned}
\mathrm{d}\langle \xi_i(t), \xi_j(t) \rangle = {} & \langle \mathsf{P}_x \mathsf{P}_{x_i(t)} b_\rho[\mu(t)](x_i(t)), \xi_j(t) \rangle \mathrm{d}t + \langle \xi_i(t), \mathsf{P}_x \mathsf{P}_{x_j(t)} b_\rho[\mu(t)](x_j(t)) \rangle \mathrm{d}t \\
& - \frac{c(s)}{2d^2}(d-1)\left(\|m_i(t)\|^2 + \|m_j(t)\|^2\right)\langle \xi_i(t), \xi_j(t) \rangle \mathrm{d}t \\
& + \frac{c(s)}{d^2}\langle m_i(t), m_j(t) \rangle \Big( d - 1 - \|\xi_i(t)\|^2 - \|\xi_j(t)\|^2 + \langle x_i(t), x_j(t) \rangle \langle \xi_i(t), \xi_j(t) \rangle \Big) \mathrm{d}t \\
& + \int_{\Theta} \Big( \langle \mathsf{P}_x \mathsf{P}_{x_i(t)} \Delta V m_i(t), \xi_j(t) \rangle + \langle \xi_i(t), \mathsf{P}_x \mathsf{P}_{x_j(t)} \Delta V m_j(t) \rangle \Big) W(\mathrm{d}\theta, \mathrm{d}t).
\end{aligned}$$

We use the two following claims in the following

**Lemma D.6.** *For $\varepsilon > 0$ small enough and every $\xi \in \mathcal{U}_\varepsilon$, one has, for every $(i, k) \in \{1, \ldots, n\}^2$,*

$$\langle \mathsf{P}_{x_i} \xi_k, b_\rho[\mu_x](x_i) \rangle = \langle \xi_k, V\bar{\xi} \rangle - \lambda_1 \langle \xi_i, \xi_k \rangle + R_i^V(\xi),$$

*where the remainders satisfy $\left| \sum_{i=1}^n R_i^V(\xi) \right| \leq C_V \varepsilon Q$. Besides, we have*

$$\langle m_A[\mu(t)](x_i(t)), m_A[\mu(t)](x_j(t)) \rangle = 1 + \|\bar{\xi}\|^2 - \frac{1}{n}\sum_{k=1}^n \|\xi_k\|^2 + R_{ij}^m(\xi) = 1 + B(t) - \frac{Q(t)}{n} + R_{ij}^m(\xi),$$

*where $|R_{ij}^m(\xi)| \leq C\|\xi\|^4$.*

*Proof.* We write

$$b_\rho[\mu_x](x_i) = \sum_{j=1}^n \frac{\exp(\langle A x_i, x_j \rangle)}{Z_A[\mu_x](x_i)} V x_j.$$

Since

$$x_j = \alpha_j x + \xi_j, \qquad \alpha_j = 1 - \frac{1}{2}\|\xi_j\|^2 + O(\|\xi_j\|^4),$$

and $V x = \lambda_1 x$, we have

$$V x_j = \lambda_1 \alpha_j x + V \xi_j.$$

Moreover,

$$\mathsf{P}_{x_i} \xi_k = \xi_k - \langle \xi_k, x_i \rangle x_i = \xi_k - \langle \xi_i, \xi_k \rangle x_i.$$

Therefore

$$\langle \mathsf{P}_{x_i} \xi_k, V x_j \rangle = \langle \xi_k, V\xi_j \rangle - \lambda_1 \alpha_j \alpha_i \langle \xi_i, \xi_k \rangle - \langle \xi_k, \xi_i \rangle \langle x_i, V\xi_j \rangle.$$

The last term is cubic on $\mathcal{U}_\varepsilon$, and $\alpha_j^2 = 1 + O(\varepsilon^2)$. Hence

$$\langle \mathsf{P}_{x_i} \xi_k, V x_j \rangle = \langle \xi_k, V\xi_j \rangle - \lambda_1 \langle \xi_i, \xi_k \rangle + O(\varepsilon \langle \xi_i, \xi_j \rangle).$$

The attention weights satisfy

$$\frac{\exp(\langle A x_i, x_j \rangle)}{Z_A[\mu_x](x_i)} = \frac{1}{n} + O(\varepsilon^2)$$

uniformly on $\mathcal{U}_\varepsilon$, because the first-order variation vanishes in the chart around the diagonal cluster. Thus

$$\langle \mathsf{P}_{x_i}\xi_k, b_\rho[\mu_x](x_i)\rangle = \frac{1}{n}\sum_{j=1}^n \langle \xi_k, \boldsymbol{V}\xi_j\rangle - \lambda_1\langle \xi_i, \xi_k\rangle + R_i^V(\xi)$$
$$= \langle \xi_k, \boldsymbol{V}\bar{\xi}\rangle - \lambda_1\langle \xi_i, \xi_k\rangle + R_i^V(\xi).$$

and the remainder satisfies

$$|R^V(\xi)| \le C_V \varepsilon Q.$$

For the second part, we have

$$\|m_{\boldsymbol{A}}[\mu(t)](x_i(t))\|^2 = \sum_{j,j'=1}^n (\alpha_j\alpha_{j'} + \langle \xi_j, \xi_{j'}\rangle)\frac{e^{\lambda_1\alpha_i(\alpha_j+\alpha_{j'})+\langle \boldsymbol{V}\xi_i, \xi_j+\xi_{j'}\rangle}}{\sum_{j,j'=1}^n e^{\lambda_1\alpha_i(\alpha_j+\alpha_{j'})+\langle \boldsymbol{V}\xi_i, \xi_j+\xi_{j'}\rangle}}$$
$$= 1 + \frac{1}{n^2}\sum_{j,j'=1}^n \langle \xi_j, \xi_{j'}\rangle - \frac{1}{n}\sum_{j=1}^n \|\xi_j\|^2 + R_i^m(\xi),$$

where $\alpha_j = \sqrt{1 - \|\xi_j\|^2}$, and, locally around the clustered configuration, $|R_i^m(\xi)| \le C\|\xi\|^4$. For the third part, we have

$$\langle m_{\boldsymbol{A}}[\mu(t)](x_i(t)), m_{\boldsymbol{A}}[\mu(t)](x_j(t))\rangle = \sum_{k,\ell=1}^n \langle x_k(t), x_\ell(t)\rangle\frac{e^{\langle \boldsymbol{V}x_i(t), x_k(t)\rangle}e^{\langle \boldsymbol{V}x_j(t), x_\ell(t)\rangle}}{Z_{\boldsymbol{A}}[\mu(t)](x_i(t))Z_{\boldsymbol{A}}[\mu(t)](x_j(t))}.$$

Since $x_k = \alpha_k x + \xi_k$, $\qquad \alpha_k = \sqrt{1 - \|\xi_k\|^2}$, and $\boldsymbol{V}x = \lambda_1 x$, this can be written as

$$\langle m_{\boldsymbol{A}}[\mu(t)](x_i(t)), m_{\boldsymbol{A}}[\mu(t)](x_j(t))\rangle = \sum_{k,\ell=1}^n (\alpha_k\alpha_\ell + \langle \xi_k, \xi_\ell\rangle)\frac{e^{\lambda_1\alpha_i\alpha_k+\langle \boldsymbol{V}\xi_i, \xi_k\rangle+\lambda_1\alpha_j\alpha_\ell+\langle \boldsymbol{V}\xi_j, \xi_\ell\rangle}}{\sum_{k,\ell=1}^n e^{\lambda_1\alpha_i\alpha_k+\langle \boldsymbol{V}\xi_i, \xi_k\rangle+\lambda_1\alpha_j\alpha_\ell+\langle \boldsymbol{V}\xi_j, \xi_\ell\rangle}}.$$

Now

$$\alpha_k\alpha_\ell = 1 - \frac{1}{2}\|\xi_k\|^2 - \frac{1}{2}\|\xi_\ell\|^2 + O(\|\xi\|^4),$$

and therefore

$$\langle x_k, x_\ell\rangle = 1 - \frac{1}{2}\|\xi_k\|^2 - \frac{1}{2}\|\xi_\ell\|^2 + \langle \xi_k, \xi_\ell\rangle + O(\|\xi\|^4).$$

Moreover, the exponential weights are equal to $1/n^2$ up to an error of order $O(\|\xi\|^2)$. Since the weights sum to one, their order-two perturbation does not contribute against the constant term 1, and its product with $\langle x_k, x_\ell\rangle - 1$ is of order $O(\|\xi\|^4)$. Hence

$$\langle m_{\boldsymbol{A}}[\mu(t)](x_i(t)), m_{\boldsymbol{A}}[\mu(t)](x_j(t))\rangle = 1 + \frac{1}{n^2}\sum_{k,\ell=1}^n \left(-\frac{1}{2}\|\xi_k\|^2 - \frac{1}{2}\|\xi_\ell\|^2 + \langle \xi_k, \xi_\ell\rangle\right) + R_{ij}^m(\xi)$$
$$= 1 - \frac{1}{n}\sum_{k=1}^n \|\xi_k\|^2 + \frac{1}{n^2}\sum_{k,\ell=1}^n \langle \xi_k, \xi_\ell\rangle + R_{ij}^m(\xi).$$

Thus

$$\langle m_{\boldsymbol{A}}[\mu(t)](x_i(t)), m_{\boldsymbol{A}}[\mu(t)](x_j(t))\rangle = 1 + \|\bar{\xi}\|^2 - \frac{1}{n}\sum_{k=1}^n \|\xi_k\|^2 + R_{ij}^m(\xi),$$

with $|R_{ij}^m(\xi)| \le C\|\xi\|^4$. $\qquad\square$

First, by Lemma D.6, for every $i, j \in \{1, \ldots, n\}$,

$$\langle \mathsf{P}_x\mathsf{P}_{x_i(t)}b_\rho[\mu(t)](x_i(t)), \xi_j(t)\rangle = \langle \mathsf{P}_{x_i(t)}\xi_j(t), b_\rho[\mu(t)](x_i(t))\rangle$$
$$= \langle \xi_j(t), \boldsymbol{V}\bar{\xi}(t)\rangle - \lambda_1\langle \xi_i(t), \xi_j(t)\rangle + R_{ij}^V(t),$$

and similarly
$$\langle \xi_i(t), \mathsf{P}_x \mathsf{P}_{x_j(t)} b_\rho[\mu(t)](x_j(t)) \rangle = \langle \xi_i(t), \mathbf{V}\bar{\xi}(t) \rangle - \lambda_1 \langle \xi_i(t), \xi_j(t) \rangle + R^V_{ji}(t).$$

Therefore the drift contribution of the value term is

$$\langle \mathsf{P}_x \mathsf{P}_{x_i(t)} b_\rho[\mu(t)](x_i(t)), \xi_j(t) \rangle + \langle \xi_i(t), \mathsf{P}_x \mathsf{P}_{x_j(t)} b_\rho[\mu(t)](x_j(t)) \rangle = \langle \xi_i(t) + \xi_j(t), \mathbf{V}\bar{\xi}(t) \rangle - 2\lambda_1 C_{ij}(t) + R^{V,\text{pair}}_{ij}(t),$$

where $R^{V,\text{pair}}_{ij}(t) := R^V_{ij}(t) + R^V_{ji}(t)$.

By the second part of Lemma D.6,

$$-\frac{c(s)}{2d^2}(d-1)\left( \|m_i(t)\|^2 + \|m_j(t)\|^2 \right) C_{ij}(t) + \frac{c(s)}{d^2}\langle m_i(t), m_j(t) \rangle \Big( d - 1 - q_i(t) - q_j(t) + \langle x_i(t), x_j(t) \rangle C_{ij}(t) \Big)$$
$$= \kappa\left( 1 + B(t) - \frac{Q(t)}{n} \right)\Big( d - 1 - q_i(t) - q_j(t) - (d-2)C_{ij}(t) \Big) + R^{\text{LoRA}}_{ij}(t).$$

Equivalently, expanding only up to second order,

$$-\frac{\alpha c(s)}{2d^2}(d-1)\left( \|m_i(t)\|^2 + \|m_j(t)\|^2 \right) C_{ij}(t) + \alpha \frac{c(s)}{d^2}\langle m_i(t), m_j(t) \rangle \Big( d - 1 - q_i(t) - q_j(t) + \langle x_i(t), x_j(t) \rangle C_{ij}(t) \Big)$$
$$= \kappa\Big( d - 1 + (d-1)(B(t) - \frac{Q(t)}{n}) - q_i(t) - q_j(t) - (d-2)C_{ij}(t) \Big) + R^{\text{LoRA}}_{ij}(t),$$

where, on the local set $\mathcal{U}_\varepsilon$,
$$|R^{\text{LoRA}}_{ij}(t)| \le C\kappa \|\xi(t)\|^4.$$

Hence the pair process satisfies the locally closed SDE

$$\begin{aligned}
\mathrm{d}C_{ij}(t) = {}& \Big[ \langle \xi_i(t) + \xi_j(t), \mathbf{V}\bar{\xi}(t) \rangle - 2\lambda_1 C_{ij}(t) \Big] \mathrm{d}t \\
& + \kappa\Big( d - 1 + (d-1)(B(t) - \frac{Q(t)}{n}) - q_i(t) - q_j(t) - (d-2)C_{ij}(t) \Big) \mathrm{d}t \\
& + R_{ij}(t)\mathrm{d}t + \mathrm{d}M_{ij}(t),
\end{aligned}$$

where $R_{ij}(t) := R^{V,\text{pair}}_{ij}(t) + R^{\text{LoRA}}_{ij}(t)$, and

$$\mathrm{d}M_{ij}(t) := \sqrt{\alpha} \int_\Theta \Big( \langle \mathsf{P}_x \mathsf{P}_{x_i(t)} \Delta V m_i(t), \xi_j(t) \rangle + \langle \xi_i(t), \mathsf{P}_x \mathsf{P}_{x_j(t)} \Delta V m_j(t) \rangle \Big) W(\mathrm{d}\theta, \mathrm{d}t).$$

In other words,

$$\begin{aligned}
\mathrm{d}\langle \xi_i(t), \xi_j(t) \rangle = {}& \Big[ \langle \xi_i(t) + \xi_j(t), \mathbf{V}\bar{\xi}(t) \rangle - 2\lambda_1 \langle \xi_i(t), \xi_j(t) \rangle \Big] \mathrm{d}t \\
& + \kappa\Big( d - 1 + (d-1)(B(t) - \frac{Q(t)}{n}) - \|\xi_i(t)\|^2 - \|\xi_j(t)\|^2 - (d-2)\langle \xi_i(t), \xi_j(t) \rangle \Big) \mathrm{d}t \\
& + R_{ij}(t)\mathrm{d}t + \mathrm{d}M_{ij}(t).
\end{aligned}$$

**Bounding $S$.** Applying the above decomposition, we obtain a closed equation for the pairwise distance

$$D_{ij}(t) := \|\xi_i(t) - \xi_j(t)\|^2 = \|\xi_i(t)\|^2 + \|\xi_j(t)\|^2 - 2\langle \xi_i(t), \xi_j(t) \rangle.$$

Indeed,
$$\mathrm{d}D_{ij}(t) = \mathrm{d}\|\xi_i(t)\|^2 + \mathrm{d}\|\xi_j(t)\|^2 - 2\mathrm{d}\langle \xi_i(t), \xi_j(t) \rangle.$$

Using the previous formula with $(i,i)$ and $(j,j)$ gives

$$\begin{aligned}
\mathrm{d}\|\xi_i(t)\|^2 = {}& \Big[ 2\langle \xi_i(t), \mathbf{V}\bar{\xi}(t) \rangle - 2\lambda_1 \|\xi_i(t)\|^2 \Big] \mathrm{d}t \\
& + \kappa\Big( d - 1 + (d-1)(B(t) - \frac{Q(t)}{n}) - d\|\xi_i(t)\|^2 \Big) \mathrm{d}t \\
& + R_{ii}(t)\mathrm{d}t + \mathrm{d}M_{ii}(t),
\end{aligned}$$

and similarly for $\|\xi_j(t)\|^2$. Therefore,

$$
\begin{aligned}
\mathrm{d}D_{ij}(t) = &\Big[2\langle\xi_i(t), \boldsymbol{V}\bar{\xi}(t)\rangle + 2\langle\xi_j(t), \boldsymbol{V}\bar{\xi}(t)\rangle - 2\lambda_1\|\xi_i(t)\|^2 - 2\lambda_1\|\xi_j(t)\|^2\Big]\mathrm{d}t \\
&- 2\Big[\langle\xi_i(t) + \xi_j(t), \boldsymbol{V}\bar{\xi}(t)\rangle - 2\lambda_1\langle\xi_i(t), \xi_j(t)\rangle\Big]\mathrm{d}t \\
&+ \kappa\Big[2\big(d - 1 + (d-1)(B(t) - \frac{Q(t)}{n})\big) - d\|\xi_i(t)\|^2 - d\|\xi_j(t)\|^2\Big]\mathrm{d}t \\
&- 2\kappa\Big[d - 1 + (d-1)(B(t) - \frac{Q(t)}{n}) - \|\xi_i(t)\|^2 - \|\xi_j(t)\|^2 - (d-2)\langle\xi_i(t), \xi_j(t)\rangle\Big]\mathrm{d}t \\
&+ \big(R_{ii}(t) + R_{jj}(t) - 2R_{ij}(t)\big)\mathrm{d}t + \mathrm{d}M_{ii}(t) + \mathrm{d}M_{jj}(t) - 2\mathrm{d}M_{ij}(t).
\end{aligned}
$$

The barycentric term cancels exactly:

$$
2\langle\xi_i, \boldsymbol{V}\bar{\xi}\rangle + 2\langle\xi_j, \boldsymbol{V}\bar{\xi}\rangle - 2\langle\xi_i + \xi_j, \boldsymbol{V}\bar{\xi}\rangle = 0.
$$

Moreover, the additive LoRA contribution also cancels:

$$
2\big(d - 1 + (d-1)(B - \frac{Q}{n})\big) - 2\big(d - 1 + (d-1)(B - \frac{Q}{n})\big) = 0.
$$

Thus

$$
\begin{aligned}
\mathrm{d}D_{ij}(t) = &-2\lambda_1\Big(\|\xi_i(t)\|^2 + \|\xi_j(t)\|^2 - 2\langle\xi_i(t), \xi_j(t)\rangle\Big)\mathrm{d}t - \kappa(d-2)\Big(\|\xi_i(t)\|^2 + \|\xi_j(t)\|^2 - 2\langle\xi_i(t), \xi_j(t)\rangle\Big)\mathrm{d}t \\
&+ \widetilde{R}_{ij}(t)\mathrm{d}t + \mathrm{d}\widetilde{M}_{ij}(t),
\end{aligned}
$$

where $\widetilde{R}_{ij}(t) := R_{ii}(t) + R_{jj}(t) - 2R_{ij}(t)$, and $\mathrm{d}\widetilde{M}_{ij}(t) := \mathrm{d}M_{ii}(t) + \mathrm{d}M_{jj}(t) - 2\mathrm{d}M_{ij}(t)$. Equivalently,

$$
\mathrm{d}D_{ij}(t) = -\big(2\lambda_1 + (d-2)\kappa\big)D_{ij}(t)\mathrm{d}t + \widetilde{R}_{ij}(t)\mathrm{d}t + \mathrm{d}\widetilde{M}_{ij}(t). \tag{99}
$$

Summing over all pairs, we obtain

$$
\mathrm{d}S(t) = -\big(2\lambda_1 + (d-2)\kappa\big)S(t)\mathrm{d}t + R_S(t)\mathrm{d}t + \mathrm{d}M_S(t),
$$

where

$$
R_S(t) := \frac{1}{2n}\sum_{i,j=1}^{n}\widetilde{R}_{ij}(t), \qquad \mathrm{d}M_S(t) := \frac{1}{2n}\sum_{i,j=1}^{n}\mathrm{d}\widetilde{M}_{ij}(t).
$$

On the stopped interval

$$
\tau_\varepsilon := \inf\left\{t \geq 0 : \max_{1 \leq i \leq n}\|\xi_i(t)\| \geq \varepsilon\right\},
$$

the local Taylor remainders satisfy the centered estimate

$$
|R_S(t)| \leq C(1 + \kappa)\varepsilon S(t), \qquad t \leq \tau_\varepsilon.
$$

Hence, on $[0, \tau_\varepsilon]$,

$$
\mathrm{d}S(t) \leq -\big(2\lambda_1 + (d-2)\kappa - C(1+\kappa)\varepsilon\big)S(t)\mathrm{d}t + \mathrm{d}M_S(t).
$$

Set

$$
\gamma_\varepsilon := 2\lambda_1 + (d-2)\kappa - C(1+\kappa)\varepsilon.
$$

We assume that $\varepsilon > 0$ is chosen small enough so that $\gamma_\varepsilon > 0$. Up to the exit time

$$
\tau_\varepsilon := \inf\left\{t \geq 0 : \max_{1 \leq i \leq n}\|\xi_i(t)\| \geq \varepsilon\right\},
$$

the previous computations give

$$
\mathrm{d}S(t) \leq -\gamma_\varepsilon S(t)\,\mathrm{d}t + \mathrm{d}M_S(t), \qquad t < \tau_\varepsilon,
$$

where $M_S$ is a local martingale. Equivalently, for every $t \geq 0$,

$$S(t \wedge \tau_\varepsilon) \leq S(0) - \gamma_\varepsilon \int_0^t \mathbf{1}_{\{s < \tau_\varepsilon\}} S(s) \, \mathrm{d}s + M_S(t \wedge \tau_\varepsilon).$$

Multiplying by the integrating factor and using Itô's formula gives

$$\mathrm{d}\left(e^{\gamma_\varepsilon t} S(t)\right) \leq e^{\gamma_\varepsilon t} \, \mathrm{d}M_S(t), \qquad t < \tau_\varepsilon.$$

Hence

$$e^{\gamma_\varepsilon (t \wedge \tau_\varepsilon)} S(t \wedge \tau_\varepsilon) \leq S(0) + \int_0^{t \wedge \tau_\varepsilon} e^{\gamma_\varepsilon s} \, \mathrm{d}M_S(s).$$

By localization and optional stopping, the martingale term has zero expectation. Therefore

$$\mathbb{E}\left[e^{\gamma_\varepsilon (t \wedge \tau_\varepsilon)} S(t \wedge \tau_\varepsilon)\right] \leq S(0).$$

In particular,

$$\mathbb{E}\left[S(t) \mathbf{1}_{\{t < \tau_\varepsilon\}}\right] \leq e^{-\gamma_\varepsilon t} S(0).$$

Since the paths are continuous and $S(\tau_\varepsilon) \leq Q(\tau_\varepsilon) \leq n\varepsilon^2$, we also have

$$\mathbb{E}S(t \wedge \tau_\varepsilon) \leq e^{-\gamma_\varepsilon t} S(0) + n\varepsilon^2 \mathbb{P}(\tau_\varepsilon \leq t).$$

$\square$

**Lemma D.7.** *We have the following inequality*

$$|\mathbb{E}[Q(t)] - q_\infty| \lesssim (\kappa + \|\delta\| + \varepsilon^3) n \exp\left(-t\left(\text{gap} + \frac{d\kappa}{2}\right)\right) + \exp(-C_S(t))\|\delta\| + (n + q_\infty)\mathbb{P}(\tau_\varepsilon \leq t), \qquad (100)$$

*where* $q_\infty := n \sum_{a=2}^d \frac{\kappa}{2\left(\lambda_1 + \frac{d\kappa}{2} - \lambda_a\right)}$.

*Proof.* We first derive the Itô sde for the barycenter of $\xi_i$. On the time interval $[0, \tau_\varepsilon)$, by previous computations, the local barycenter dynamics has the form

$$\mathrm{d}\bar{\xi}(t) = \mathsf{P}_x \frac{1}{n} \sum_{i=1}^n \mathsf{P}_{x_i(t)} b_\rho[\mu(t)](x_i(t)) \mathrm{d}t + \mathsf{P}_x \frac{1}{n} \sum_{i=1}^n \mathsf{P}_{x_i(t)} \int_\Theta \Delta V m_A[\mu(t)](x_i(t)) W(\mathrm{d}\theta, \mathrm{d}t)$$

$$- \frac{1}{2n} \sum_{i=1}^n \left(\int_\Theta \|\mathsf{P}_{x_i(t)} \Delta V m_A[\mu(t)](x_i(t))\|^2 \rho(\mathrm{d}\theta)\right) \xi_i(t) \mathrm{d}t.$$

Using the same proof as in Theorem D.6, we obtain that the local barycenter dynamics has the following form:

$$\mathrm{d}\bar{\xi}(t) = M\bar{\xi}(t)\mathrm{d}t + \sqrt{\kappa} \, \mathrm{d}\beta_t + \mathcal{R}_{\bar{\xi}}(t)\mathrm{d}t + \mathrm{d}\mathcal{N}_{\bar{\xi}}(t),$$

where we introduce the tangent operator $V_T := \mathsf{P}_x V \mathsf{P}_x\big|_{T_x \mathbb{S}^{d-1}}$, and the effective linearized barycenter drift $M := V_T - \lambda_1 I_{T_x \mathbb{S}^{d-1}} - \frac{d\kappa}{2} I_{T_x \mathbb{S}^{d-1}}$, and $\beta_t$ is a standard Brownian motion on $T_x \mathbb{S}^{d-1}$, and the terms $\mathcal{R}_{\bar{\xi}}$ and $\mathcal{N}_{\bar{\xi}}$ collect the local Taylor remainders. All the following computations are performed on the event $\{t < \tau_\varepsilon\}$. The contribution of the complementary event $\{\tau_\varepsilon \leq t\}$ will be estimated at the end. Denote $\Gamma(t) = \bar{\xi}(t)\bar{\xi}(t)^\top$, we have

$$\mathrm{d}\Gamma(s) = \left(M\Gamma(s) + \Gamma(s)M^\top + \kappa I_{x^\perp}\right) \mathrm{d}s +$$
$$+ \sqrt{\kappa}\mathrm{d}\beta_s \bar{\xi}(s)^\top + \bar{\xi}(s)\mathrm{d}\beta_s^\top + \mathcal{R}_{\bar{\xi}(s)}\bar{\xi}(s)^\top + \bar{\xi}(s)\mathcal{R}_{\bar{\xi}(s)}^\top \mathrm{d}s$$
$$+ \mathcal{E}_{\bar{\xi}}(t)\mathrm{d}t + \mathrm{d}\mathcal{M}_{\Gamma(t)},$$

where the two last terms collect i) the quadratic variations erors coming from $\mathcal{N}_{\bar{\xi}}$ and the second is a martingale term.

**Solution of the linearized equation** The linearized equation for the covariance is given by $\Gamma_{lin}(t)$ following the Lyapunov equation

$$\frac{\mathrm{d}}{\mathrm{d}t}\Gamma_{lin}(t) = M\Gamma_{lin}(t) + \Gamma_{lin}(t)M^\top + \kappa I_{T_x\mathbb{S}^{d-1}}, \quad \Gamma_{lin}(0) = \bar{\xi} \otimes \bar{\xi}. \tag{101}$$

The solution of this linear SDE is given by

$$\Gamma_{lin}(t) = e^{tM}\Gamma_{lin}(0)e^{tM^\top} + \kappa \int_0^t e^{sM}I_{x^\perp}e^{sM^\top}\mathrm{d}s. \tag{102}$$

**Comparison between the linearized equation and the original SDE** We differentiate $\Gamma - \Gamma_{lin}$, and we obtain

$$\mathrm{d}\Gamma(s) - \mathrm{d}\Gamma_{lin}(s) = M(\Gamma(s) - \Gamma_{lin}(s)) + (\Gamma(s) - \Gamma_{lin}(s))M^\top$$
$$+ \sqrt{\kappa}\mathrm{d}\beta_s\bar{\xi}(s)^\top\mathrm{d}s + \bar{\xi}(s)\mathrm{d}\beta_s^\top + \mathcal{R}_{\bar{\xi}(s)}\bar{\xi}(s)^\top + \bar{\xi}(s)\mathcal{R}_{\bar{\xi}(s)}^\top\mathrm{d}s$$
$$+ \mathcal{E}_{\bar{\xi}}(t)\mathrm{d}t + \mathrm{d}\mathcal{M}_{\Gamma(t)},$$

We now use Itô formula to $s \mapsto e^{(t-s)M}(\Gamma(s) - \Gamma_{lin}(s))e^{(t-s)M^\top}$, and we obtain

$$\mathrm{d}e^{(t-s)M}(\Gamma(s) - \Gamma_{lin}(s))e^{(t-s)M^\top}$$
$$= -e^{(t-s)M}(M\Gamma(s) - M\Gamma_{lin}(s))e^{(t-s)M^\top}\mathrm{d}s - e^{(t-s)M}(\Gamma(s)M^\top - \Gamma_{lin}(s)M^\top)e^{(t-s)M^\top}\mathrm{d}s$$
$$+ e^{(t-s)M}(M(\Gamma(s) - \Gamma_{lin}(s)) + (\Gamma(s) - \Gamma_{lin}(s))M^\top)e^{(t-s)M^\top}\mathrm{d}s$$
$$+ e^{(t-s)M}\left(\sqrt{\kappa}\mathrm{d}\beta_s\bar{\xi}(s)^\top + \bar{\xi}(s)\mathrm{d}\beta_s^\top + \mathcal{R}_{\bar{\xi}(s)}\bar{\xi}(s)^\top + \bar{\xi}(s)\mathcal{R}_{\bar{\xi}(s)}^\top\mathrm{d}s\right)e^{(t-s)M^\top}$$
$$+ e^{(t-s)M}\left(\mathcal{E}_{\bar{\xi}}(t)\mathrm{d}t + \mathrm{d}\mathcal{M}_{\Gamma(t)}\right)e^{(t-s)M^\top}$$
$$= e^{(t-s)M}\left(\sqrt{\kappa}\mathrm{d}\beta_s\bar{\xi}(s)^\top + \bar{\xi}(s)\mathrm{d}\beta_s^\top + \mathcal{R}_{\bar{\xi}(s)}\bar{\xi}(s)^\top + \bar{\xi}(s)\mathcal{R}_{\bar{\xi}(s)}^\top\mathrm{d}s\right)e^{(t-s)M^\top}$$
$$+ e^{(t-s)M}\left(\mathcal{E}_{\bar{\xi}}(t)\mathrm{d}t + \mathrm{d}\mathcal{M}_{\Gamma(t)}\right)e^{(t-s)M^\top}$$

Integrating this, we obtain

$$\Gamma(t) - \Gamma_{lin}(t) = \int_0^t e^{(t-s)M}\left(\sqrt{\kappa}\mathrm{d}\beta_s\bar{\xi}(s)^\top + \bar{\xi}(s)\mathrm{d}\beta_s^\top\right)e^{(t-s)M^\top}$$
$$+ \int_0^t e^{(t-s)M}\left(\mathcal{R}_{\bar{\xi}(s)}\bar{\xi}(s)^\top + \bar{\xi}(s)\mathcal{R}_{\bar{\xi}(s)}^\top + \mathcal{N}_{\bar{\xi}(s)}\bar{\xi}(s)^\top + \bar{\xi}(s)\mathcal{N}_{\bar{\xi}(s)}^\top\right)e^{(t-s)M^\top}\mathrm{d}s$$

Recall that we have the following bounds

$$\|\mathcal{R}_{\bar{\xi}(s)}\| \lesssim (1+\kappa)\varepsilon\left(\|\bar{\xi}\| + \frac{S^{1/2}}{\sqrt{n}}\right), \quad |\mathcal{R}_{\bar{\xi}(s)}\bar{\xi}(s)^\top + \bar{\xi}(s)\mathcal{R}_{\bar{\xi}(s)}^\top| \lesssim (1+\kappa)\varepsilon(\|\bar{\xi}\|^2 + \frac{S}{n}).$$

Taking expectation, we obtain

$$|\mathbb{E}[\mathrm{Tr}(\Gamma(t))] - \mathrm{Tr}(\Gamma_{lin}(t))| \lesssim (1+\kappa)\varepsilon \int_0^t \|\bar{\xi}(s)\|^2 \sum_{i=1}^d e^{2\lambda_i(M)(t-s)}\mathrm{d}s. \tag{103}$$

Besides, taking the trace in (102), we obtain

$$\mathbb{E}\left[\mathrm{Tr}\left(\Gamma_{lin}(t)\right)\right] = \mathrm{Tr}\left(e^{tM}\Gamma_{lin}(0)e^{tM^\top}\right) + \kappa \int_0^t \mathrm{Tr}\left(e^{sM}I_{x^\perp}e^{sM^\top}\right)\mathrm{d}s$$

$$= \langle e^{tM}\bar{\xi}(0), e^{tM}\bar{\xi}(0)\rangle + \kappa \int_0^t \sum_{i=1}^d \langle e^{sM}e_i, e^{sM}e_i\rangle\mathrm{d}s$$

$$= \langle e^{tM}\bar{\xi}(0), e^{tM}\bar{\xi}(0)\rangle + \kappa \int_0^t \sum_{i=2}^d e^{2s(\lambda_i - \lambda_1 - \frac{d\kappa}{2})}\mathrm{d}s$$

$$= \sum_{i=2}^d \frac{\kappa}{2(-\lambda_i + \lambda_1 + \frac{d\kappa}{2})} - e^{2t(\lambda_i - \lambda_1 - \frac{d\kappa}{2})}\sum_{i=2}^d \frac{\kappa}{2(-\lambda_i + \lambda_1 + \frac{d\kappa}{2})} + \langle e^{tM}\bar{\xi}(0), e^{tM}\bar{\xi}(0)\rangle.$$

Using the comparison with the linearized covariance, we obtain on the local event $\{t < \tau_\varepsilon\}$

$$\left| \mathbb{E}\left[ B(t)\mathbf{1}_{\{t < \tau_\varepsilon\}} \right] - \operatorname{Tr}\Gamma_{\mathrm{lin}}(t)\,\mathbb{P}(t < \tau_\varepsilon) \right| \lesssim (\kappa + \|\delta\| + \varepsilon^3)e^{-2t(\mathrm{gap}+d\kappa/2)} + \frac{1}{n}e^{-C_S t}S_0.$$

Since

$$Q(t) = S(t) + nB(t),$$

and since

$$\mathbb{E}\left[ S(t)\mathbf{1}_{\{t < \tau_\varepsilon\}} \right] \le e^{-C_S t}S_0,$$

we deduce

$$\left| \mathbb{E}\left[ (Q(t) - q_\infty)\mathbf{1}_{\{t < \tau_\varepsilon\}} \right] \right| \lesssim (\kappa + \|\delta\| + \varepsilon^3)n e^{-2t(\mathrm{gap}+d\kappa/2)} + e^{-C_S t}S_0.$$

Finally,

$$|\mathbb{E}[Q(t)] - q_\infty| \le \left| \mathbb{E}\left[ (Q(t) - q_\infty)\mathbf{1}_{\{t < \tau_\varepsilon\}} \right] \right|$$
$$+ \mathbb{E}\left[ Q(t)\mathbf{1}_{\{\tau_\varepsilon \le t\}} \right] + q_\infty \mathbb{P}(\tau_\varepsilon \le t).$$

Since $0 \le Q(t) \le n$, this gives

$$|\mathbb{E}[Q(t)] - q_\infty| \lesssim (\kappa + \|\delta\| + \varepsilon^3)n e^{-2t(\mathrm{gap}+d\kappa/2)} + e^{-C_S t}S_0 + (n + q_\infty)\mathbb{P}(\tau_\varepsilon \le t).$$

$\square$

We bound the probability of the event $\{\tau_\varepsilon \le t\}$. Recall that

$$\tau_\varepsilon := \inf\left\{ s \ge 0 : \max_{i=1,\dots,n}\|\xi_i(s)\| \ge \varepsilon \right\}.$$

Using the decomposition

$$\xi_i = \bar{\xi} + (\xi_i - \bar{\xi}),$$

we have, for every $s \ge 0$,

$$\max_{i=1,\dots,n}\|\xi_i(s)\| \le \|\bar{\xi}(s)\| + \max_{i=1,\dots,n}\|\xi_i(s) - \bar{\xi}(s)\| \le \|\bar{\xi}(s)\| + \sqrt{S(s)}.$$

Therefore, if $\tau_\varepsilon \le t$, then either

$$\sup_{0 \le s \le t \wedge \tau_\varepsilon}\|\bar{\xi}(s)\| \ge \frac{\varepsilon}{2},$$

or

$$\sup_{0 \le s \le t \wedge \tau_\varepsilon}S(s) \ge \frac{\varepsilon^2}{4}.$$

Consequently,

$$\mathbb{P}(\tau_\varepsilon \le t) \le \mathbb{P}\left( \sup_{0 \le s \le t \wedge \tau_\varepsilon}\|\bar{\xi}(s)\| \ge \frac{\varepsilon}{2} \right) + \mathbb{P}\left( \sup_{0 \le s \le t \wedge \tau_\varepsilon}S(s) \ge \frac{\varepsilon^2}{4} \right).$$

The second term is controlled by (97). Indeed,

$$e^{C_S(s \wedge \tau_\varepsilon)}S(s \wedge \tau_\varepsilon)$$

is a nonnegative supermartingale. Hence, by Doob's maximal inequality,

$$\mathbb{P}\left( \sup_{0 \le s \le t \wedge \tau_\varepsilon}S(s) \ge \frac{\varepsilon^2}{4} \right) \le \mathbb{P}\left( \sup_{0 \le s \le t}e^{C_S(s \wedge \tau_\varepsilon)}S(s \wedge \tau_\varepsilon) \ge \frac{\varepsilon^2}{4} \right)$$
$$\le \frac{4S_0}{\varepsilon^2}.$$

It remains to control the barycenter. Introduce $D(t) = \mathcal{M}(t) + \mathcal{A}(t)$, where $\mathcal{M}$ is a martingale term defined as

$$\mathcal{M}(t) = \sqrt{\kappa}\int_0^t e^{(t-s)M}\left( \mathrm{d}\beta_s\bar{\xi}(s)^\top + \bar{\xi}(s)\mathrm{d}\beta_s^\top \right)e^{(t-s)M^\top},$$

and $\mathcal{A}$ is a Taylor expansion term i.e.

$$\mathcal{A}(t) = \int_0^t e^{(t-s)M} \left( \mathcal{R}_{\bar{\xi}(s)} \bar{\xi}(s)^\top + \bar{\xi}(s) \mathcal{R}_{\bar{\xi}(s)}^\top + \mathcal{N}_{\bar{\xi}(s)} \bar{\xi}(s)^\top + \bar{\xi}(s) \mathcal{N}_{\bar{\xi}(s)}^\top \right) e^{(t-s)M^\top} \mathrm{d}s.$$

Using computations made in the above lemma, we have

$$\Gamma(t) - \Gamma_{lin}(t) = D(t).$$

We now bound the probability of deviations of the supremum of $D(t)$. Using BDG inequality, we have

$$\mathbb{P}(\sup_{s \leq t \wedge \tau_\varepsilon} \mathrm{Tr}\mathcal{M}(s) \geq u) \leq C \frac{[\mathrm{Tr}\mathcal{M}]_t}{u^2}$$

The quadratic variation of this terms is bounded by

$$[\mathrm{Tr}\mathcal{M}]_t \lesssim \kappa \sum_{i=2}^d \int_0^t e^{4(t-s)\lambda(M)_i} \langle e_i, \bar{\xi}(s) \rangle^2 \mathrm{d}s \lesssim \frac{\kappa}{4\lambda(M)_2} \varepsilon^2.$$

For the second term, we use the local estimate

$$\|\mathcal{R}_{\bar{\xi}(s)}\| \lesssim (1+\kappa)\varepsilon \left( \|\bar{\xi}\| + \frac{S^{1/2}}{\sqrt{n}} \right), \quad |\mathcal{R}_{\bar{\xi}(s)} \bar{\xi}(s)^\top + \bar{\xi}(s) \mathcal{R}_{\bar{\xi}(s)}^\top| \lesssim (1+\kappa)\varepsilon(\|\bar{\xi}\|^2 + \frac{S}{n}),$$

and we obtain

$$\sup_{s \leq t \wedge \tau_\varepsilon} \mathrm{Tr}\mathcal{A}(s) \lesssim \int_0^{t \wedge \tau_\varepsilon} e^{-2a_*(t-s)} \left( (1+\kappa)\varepsilon(B(s) + \frac{S(s)}{n}) + \kappa \left( \varepsilon^2 B(s) + \frac{S(s)}{n} + \varepsilon^4 \right) \right) \mathrm{d}s$$

Taking expectations, we obtain

$$\mathbb{E}[\sup_{s \leq t \wedge \tau_\varepsilon} \mathrm{Tr}\mathcal{A}(s)] \lesssim \frac{(1+\kappa)\varepsilon}{a_*} (B_0 + b_\infty + \frac{S_0}{nC_S}) + \frac{\kappa}{a_*} \left( \varepsilon^2(B_0 + b_\infty) + \frac{S_0}{nC_S} + \varepsilon^4 \right).$$

Then, by Markov inequality, we obtain

$$\mathbb{P}\left( \sup_{0 \leq s \leq t \wedge \tau_\varepsilon} \left| \|\bar{\xi}(s)\|^2 - \Gamma_{lin}(s) \right| \geq r \right) \lesssim \frac{\kappa\varepsilon^2}{4\lambda(M)_2 r^2} + \frac{(1+\kappa)\varepsilon}{a_* r} (B_0 + b_\infty + \frac{S_0}{nC_S}) + \frac{\kappa}{a_* r} \left( \varepsilon^2(B_0 + b_\infty) + \frac{S_0}{nC_S} + \varepsilon^4 \right)$$

$$\lesssim \frac{\kappa\varepsilon^2}{4\lambda(M)_2 r^2} + \frac{(1+\kappa)\varepsilon}{a_* r} (\delta + b_\infty) + \frac{\kappa}{a_* r} \left( \varepsilon^2 b_\infty + \delta \right).$$

Combining both estimates, we have that

$$\mathbb{P}(\tau_\varepsilon \leq t) \leq \frac{\kappa}{4\lambda(M)_2} + \frac{(1+\kappa)}{a_*} (\delta + b_\infty) + \frac{\kappa}{a_*} \left( \varepsilon b_\infty + \frac{\delta}{\varepsilon} \right) \frac{4\delta^2}{\varepsilon^2}.$$

Denote $(x_i^\ell)_{i \in \{1,\ldots,n\}}$ the iterates of the Post-Layer norm transformers as described in the main paper. By Theorem D.2, we have under Assumption D.1, for any $\varphi \in C^4((\mathbb{S}^{d-1})^n)$ the following approximation holds uniformly until macroscopic time $t_L = \eta L$:

$$\sup_{t \in [0, t_L]} \left| \mathbb{E}\varphi(x(t)) - \mathbb{E}\varphi(x^\eta(t)) \right| \leq Ce^{Ct_L} \eta(t_L + 1) \max(1, \alpha), \tag{104}$$

Applying it to $\varphi : x_i \mapsto 1 - \langle x_i, x \rangle^2$, we obtain

$$\left| \mathbb{E}\left[ \sum_{i=1}^n \left( 1 - \langle x_i^\ell, x \rangle^2 \right) - n \sum_{i=1}^d \frac{\kappa}{2(\lambda_1 + \frac{d\kappa}{2} - \lambda_i)} \right] \right|$$

$$\lesssim Ce^{Ct_L} \frac{1}{L} (t_L + 1) + (\kappa + \|\delta\| + \varepsilon^3)n \exp\left( -2t \left( \mathrm{gap} + \frac{d\kappa}{2} \right) \right) + \exp(-C_S(t))\|\delta\| + C(\varepsilon).$$

Besides, it is clear that in the scaling $\eta_L \ll 1/\sqrt{L}$ that the dynamic is given by

$$\frac{\mathrm{d}}{\mathrm{d}t}x_i(t) = \mathsf{P}_{x_i(t)}\mathbb{E}[\sum_{j=1}^{n} \frac{\exp(\langle \boldsymbol{A}x_i(t), x_j(t)\rangle)}{Z_{\boldsymbol{A}}[\mu_{X(t)}(x_i)]}(\boldsymbol{V} + \Delta V)x_j(t)] = \mathsf{P}_{x_i(t)}\sum_{j=1}^{n} \frac{\exp(\langle \boldsymbol{A}x_i(t), x_j(t)\rangle)}{Z_{\boldsymbol{A}}[\mu_{X(t)}(x_i)]}\boldsymbol{V}x_j(t),$$

which is the original dynamics and then the behavior is given by the Cone collapse lemma, all tokens converge to the biggest eigenvector in the same hemisphere. $\qquad\square$

### D.3. Proof of Theorem 4.6

*Proof.* We prove the two statements separately.

**Stability for small depths.** We use the finite-time Wasserstein stability estimate from Proposition C.5. Namely, for every finite time horizon $T > 0$, there exists $C_T > 0$ such that

$$\sup_{0 \leq t \leq T} W_2(\mu_t, \nu_t) \leq C_T\|\widetilde{V} - V\|_{\mathrm{op}},$$

where

$$\mu_t := \frac{1}{n}\sum_{i=1}^{n}\delta_{z_i(t)}, \qquad \nu_t := \frac{1}{n}\sum_{i=1}^{n}\delta_{\widetilde{z}_i(t)}.$$

Taking $T = 10T_\delta$ and choosing

$$\eta(\delta) := \frac{\delta}{2\sqrt{n}\,C_{10T_\delta}},$$

we obtain

$$\sup_{0 \leq t \leq 10T_\delta} W_2(\mu_t, \nu_t) \leq \frac{\delta}{2\sqrt{n}}.$$

Fix $t \in [T_\delta, 10T_\delta]$. By definition of $T_\delta$,

$$\mathrm{d}(z_i(t), \mathcal{C}) \leq \delta \qquad \text{for every } i \in \{1, \ldots, n\}.$$

Let

$$E := \left\{y \in \mathbb{R}^d : \mathrm{d}(y, \mathcal{C}) > 2\delta\right\}.$$

Let $\pi$ be an optimal coupling between $\mu_t$ and $\nu_t$. If $x$ belongs to the support of $\mu_t$ and $y \in E$, then the triangle inequality gives

$$\|x - y\| \geq \mathrm{d}(y, \mathcal{C}) - \mathrm{d}(x, \mathcal{C}) > \delta.$$

Hence

$$W_2(\mu_t, \nu_t)^2 = \int \|x - y\|^2\,\mathrm{d}\pi(x, y) \geq \delta^2\pi(\mathbb{R}^d \times E).$$

Since the second marginal of $\pi$ is $\nu_t$,

$$\pi(\mathbb{R}^d \times E) = \nu_t(E) = \frac{1}{n}\#\left\{i : \mathrm{d}(\widetilde{z}_i(t), \mathcal{C}) > 2\delta\right\}.$$

Therefore

$$\#\left\{i : \mathrm{d}(\widetilde{z}_i(t), \mathcal{C}) > 2\delta\right\} \leq \frac{nW_2(\mu_t, \nu_t)^2}{\delta^2} \leq \frac{1}{4}.$$

The left-hand side is an integer, hence it is zero. Thus

$$\widetilde{S}_{2\delta}(t) = \{1, \ldots, n\}.$$

Since $t \in [T_\delta, 10T_\delta]$ was arbitrary, this proves the stability statement.

**Bifurcation at larger depths.** We now prove the bifurcation statement. We first establish the spectral dominance estimate.

For notational simplicity, in this paragraph we omit tildes and write $z_i(t)$ for the modified tokens. Set $m_i(t) := \ell(z_i(t))$. The attention logit can be written as

$$w_{ij}(t) := \left\langle Qe^{t\widetilde{V}}z_i(t), Ke^{t\widetilde{V}}z_j(t) \right\rangle.$$

Using the spectral expansion

$$e^{t\widetilde{V}}z_i(t) = \sum_{p=1}^{d} e^{\lambda_p t}\varphi_p^*(z_i(t))\varphi_p,$$

we obtain

$$w_{ij}(t) = c_{11}e^{2\lambda_1 t}m_i(t)m_j(t) + R_{ij}(t),$$

where

$$R_{ij}(t) := \sum_{(p,q)\neq(1,1)} c_{pq}e^{(\lambda_p+\lambda_q)t}\varphi_p^*(z_i(t))\varphi_q^*(z_j(t)).$$

Since $\lambda_1 > |\lambda_2|$ and $\mathrm{gap} = \lambda_1 - |\lambda_2|$, for every $(p,q) \neq (1,1)$ we have

$$\lambda_p + \lambda_q \leq 2\lambda_1 - \mathrm{gap}.$$

Therefore, as long as $\widetilde{S}_{2\delta}(t) = \{1,\ldots,n\}$, the uniform bound in Assumption 4.5 gives

$$|R_{ij}(t)| \leq C_{\mathrm{sub}}e^{(2\lambda_1-\mathrm{gap})t}.$$

Let $i_-$ be the token associated with the non-maximal cluster $c_-$, and let $j_+$ be a token associated with the maximal cluster $c_+$. If the exit has not occurred before time $t$, then, by continuity and by the separation of the cluster tubes induced by the inequalities in Assumption 4.5, these tokens remain in their respective cluster tubes i.e.

$$\|z_{i_-}(t) - c_-\| \leq 2\delta, \qquad \|z_{j_+}(t) - c_+\| \leq 2\delta.$$

In particular,

$$m_{i_-}(t) \geq m_* - 2\|\ell\|\delta \geq \frac{m_*}{2}.$$

Moreover, for any token $r$ not belonging to the maximal cluster tube,

$$m_{j_+}(t) - m_r(t) \geq D_+ - 4\|\ell\|\delta \geq \frac{D_+}{2}.$$

Hence

$$w_{i_-j_+}(t) - w_{i_-r}(t) = c_{11}e^{2\lambda_1 t}m_{i_-}(t)\big(m_{j_+}(t) - m_r(t)\big) + R_{i_-j_+}(t) - R_{i_-r}(t)$$
$$\geq c_{11}e^{2\lambda_1 t}\frac{m_*}{2}\frac{D_+}{2} - 2C_{\mathrm{sub}}e^{(2\lambda_1-\mathrm{gap})t}$$
$$= a_*e^{2\lambda_1 t} - 2C_{\mathrm{sub}}e^{(2\lambda_1-\mathrm{gap})t}.$$

By the definition of $T_{\mathrm{dom}}(\delta)$, for all $t \geq T_{\mathrm{dom}}(\delta)$,

$$2C_{\mathrm{sub}}e^{(2\lambda_1-\mathrm{gap})t} \leq \frac{a_*}{2}e^{2\lambda_1 t}.$$

Thus we have $w_{i_-j_+}(t) - w_{i_-r}(t) \geq \frac{a_*}{2}e^{2\lambda_1 t}$. Equivalently,

$$e^{w_{i_-r}(t)} \leq e^{w_{i_-j_+}(t)}\exp\left(-\frac{a_*}{2}e^{2\lambda_1 t}\right).$$

Summing over all non-maximal tokens gives

$$\sum_{r\notin I_+} P_{i_-r}(t) \leq n\exp\left(-\frac{a_*}{2}e^{2\lambda_1 t}\right).$$

The second term in the definition of $T_{\mathrm{dom}}(\delta)$ ensures that

$$\sum_{r \notin I_+} P_{i_- r}(t) \leq p_*.$$

We now use the evolution of the leading coordinate. Since $\ell = \varphi_1^*$, the modified dynamics gives

$$\frac{\mathrm{d}}{\mathrm{d}t} m_{i_-}(t) = \lambda_1 \sum_{j=1}^{n} P_{i_- j}(t)\big(m_j(t) - m_{i_-}(t)\big).$$

For $j$ in the maximal cluster tube, we have

$$m_j(t) - m_{i_-}(t) \geq D_- - 4\|\ell\|\delta \geq \frac{D_-}{2}.$$

For arbitrary $j$, the uniform bound gives

$$|m_j(t) - m_{i_-}(t)| \leq 2M.$$

Therefore

$$\frac{\mathrm{d}}{\mathrm{d}t} m_{i_-}(t) \geq \lambda_1 \left[ \frac{D_-}{2} \sum_{j \in I_+} P_{i_- j}(t) - 2M \sum_{j \notin I_+} P_{i_- j}(t) \right]$$

$$= \lambda_1 \left[ \frac{D_-}{2} - \left( \frac{D_-}{2} + 2M \right) \sum_{j \notin I_+} P_{i_- j}(t) \right]$$

$$\geq \lambda_1 \left[ \frac{D_-}{2} - \left( \frac{D_-}{2} + 2M \right) p_* \right].$$

By the definition of $p_*$, the last quantity is bounded below by $\frac{\lambda_1 D_-}{4}$. Thus, for every $t \geq T_{\mathrm{dom}}(\delta)$, as long as the modified dynamics has not exited the original $2\delta$-clustered regime,

$$\frac{\mathrm{d}}{\mathrm{d}t} \ell(\widetilde{z}_{i_-}(t)) \geq \frac{\lambda_1 D_-}{4}.$$

We now conclude. Set $t_0 := \max\{T_\delta, T_{\mathrm{dom}}(\delta)\}$. If $\widetilde{S}_{2\delta}(t_0) \neq \{1, \ldots, n\}$, then $T^*(\delta) \leq t_0$ and there is nothing to prove. Otherwise, the token $i_-$ lies in the $2\delta$-tube around $c_-$. Hence

$$|\ell(\widetilde{z}_{i_-}(t_0)) - \ell(c_-)| \leq 2\|\ell\|\delta.$$

As long as the token remains inside this same tube, one must have

$$|\ell(\widetilde{z}_{i_-}(t)) - \ell(c_-)| \leq 2\|\ell\|\delta.$$

But the drift lower bound implies that this cannot remain true for longer than

$$\frac{4\|\ell\|\delta}{\lambda_1 D_-/4} = \frac{16\|\ell\|\delta}{\lambda_1 D_-}.$$

Consequently, the modified dynamics exits the original $2\delta$-clustered regime before time

$$t_0 + \frac{16\|\ell\|\delta}{\lambda_1 D_-}.$$

Thus

$$T^*(\delta) \leq \max\{T_\delta, T_{\mathrm{dom}}(\delta)\} + \frac{16\|\ell\|\delta}{\lambda_1 D_-}.$$

This proves the bifurcation estimate. The final asymptotic bound follows from the definition of $T_{\mathrm{dom}}(\delta)$. $\qquad\square$

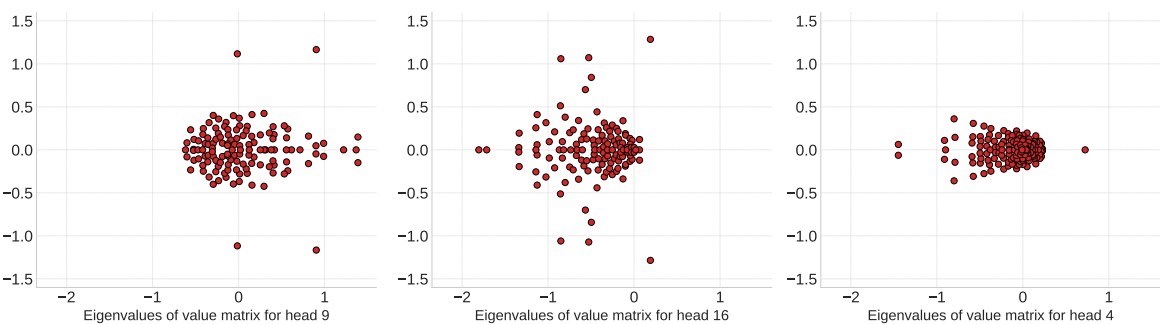

*Figure 7.* The eigenvalues of $V$ in the pre-trained ALBERT model (Head 9,16 and 4). Notice the spectral gap between the largest and second-largest eigenvalues.

# E. Numerical analysis

In this section, we will detail the numerical experiments we have carried out to illustrate the results we have obtained and then compare them with real cases.

## E.1. Simulation of self-attention dynamic

Unless indicated otherwise, all figures presented in this paper were generated by discretizing the underlying dynamics (either (2) or (7) using a fourth-order Runge-Kutta scheme with a step size of 0.1 inspired from the code of the paper Geshkovski et al. (2023). Depending on the figure, we implement different types of tokens initialization, and also different attention parameters.

### E.1.1. FIGURE (5) ILLUSTRATING THEOREM 4.6

For the Figure 5, we have taken the same setting as the phase diagram i.e $\tilde{Q} = Q = \tilde{K} = K = I_2$, and $V = I_2$, $\tilde{V} = \begin{pmatrix} 1 & 0 \\ 0 & 1 - \varepsilon \end{pmatrix}$ with $\varepsilon = 0.01$.

## E.2. Illustration and numerical analysis of Theorem 4.2

We also empirically test the result

## E.3. Methodology for Geometric Alignment

In Section 4.1.2, we presented the geometric alignment between LoRA updates and base model features. For a given layer, let $W_0 \in \mathbb{R}^{d_{out} \times d_{in}}$ be the frozen base weight. We compute its Singular Value Decomposition (SVD) and extract the top-$k$ singular vectors (with $k = 1$ for the stable direction analysis).

$$W_0 \approx U_k \Sigma_k V_k^T \tag{105}$$

The LoRA update is defined as $\Delta W = BA$. We extract the orthonormal basis of the row space of $A$ (Input subspace) and the column space of $B$ (Output subspace) using SVD.

We then calculate the principal angles between these LoRA subspaces and the base model's top-$k$ singular vectors. The raw alignment score is the mean singular value of the projection of the LoRA basis onto the base model basis. To make this metric comparable across layers of different dimensions and adapter ranks, we normalize it against a random baseline:

$$y = \frac{\text{Measured Cosine Similarity}}{\sqrt{r/d}} \tag{106}$$

A score of $y = 1.0$ implies the LoRA update aligns with the base model no better than a random subspace. The results in Figure 4 show values exceeding 1.0, confirming non-random interference.

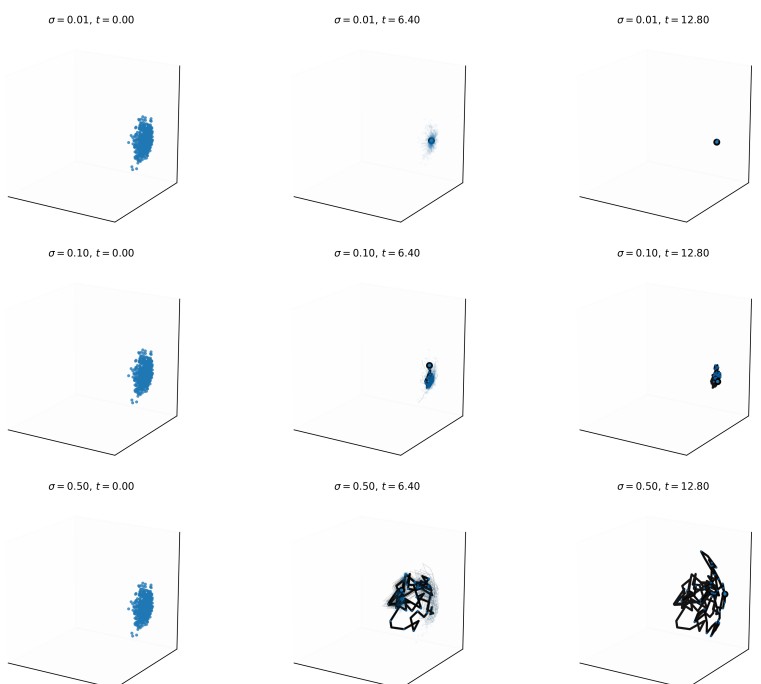

*Figure 8.* Particles stages for small, intermediate and large noise i.e ($\sigma \in \{0.01, 0.10, 0.5.\}$. The blurred lines shows the evolution of the particles over time. We observe that the particles converge to a single point i.e synchronize, and the magnitude of deviation around this point is dependent on the noise magnitude $\sigma$.

### E.3.1. FIGURE (5) ILLUSTRATING (4.6)

For this figure, we initialize tokens with uniform random variables on the hypercube $[-5, 5]^2$, we consider $\tilde{V} = V = I_2$, and $\tilde{Q} = \tilde{V} = \mathbf{e}_1 \mathbf{e}_1^T$ and $Q = K = I_2$.

### E.4. Spectrum of Value, Query and Key matrices in real-world Transformers

#### E.4.1. EIGENVALUES OF ALBERT'S VALUE MATRICES.

In Figure 7 we illustrate the eigenvalues of the value matrices $V_h$ for a couple of heads h in a pre-trained ALBERT model. We focus on ALBERT-xlarge-v2 available online at https://huggingface.co/albert-xlarge-v2. This version uses 16 heads, with sequences of length $n = 256$ and tokens of dimension $d = 128$.

#### E.4.2. EIGENVALUES OF ALBERT'S QUERY, KEY MATRICES.

We download the ALBERT model, compute the rank of the matrix parameters $A = Q^T K$, and plot this histogram. Notice that in ALBERT, all the attention heads share this property of being low rank. it is suggested by the following figure

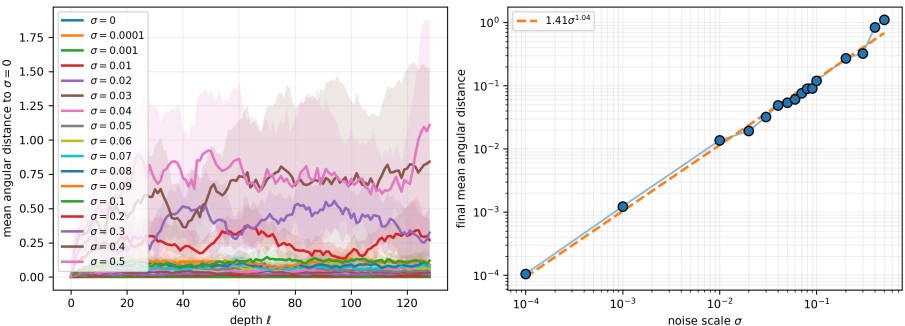

*Figure 9.* Evolution of the mean distance to the stationnary vector for the original dynamics and scaling with the noise scale.

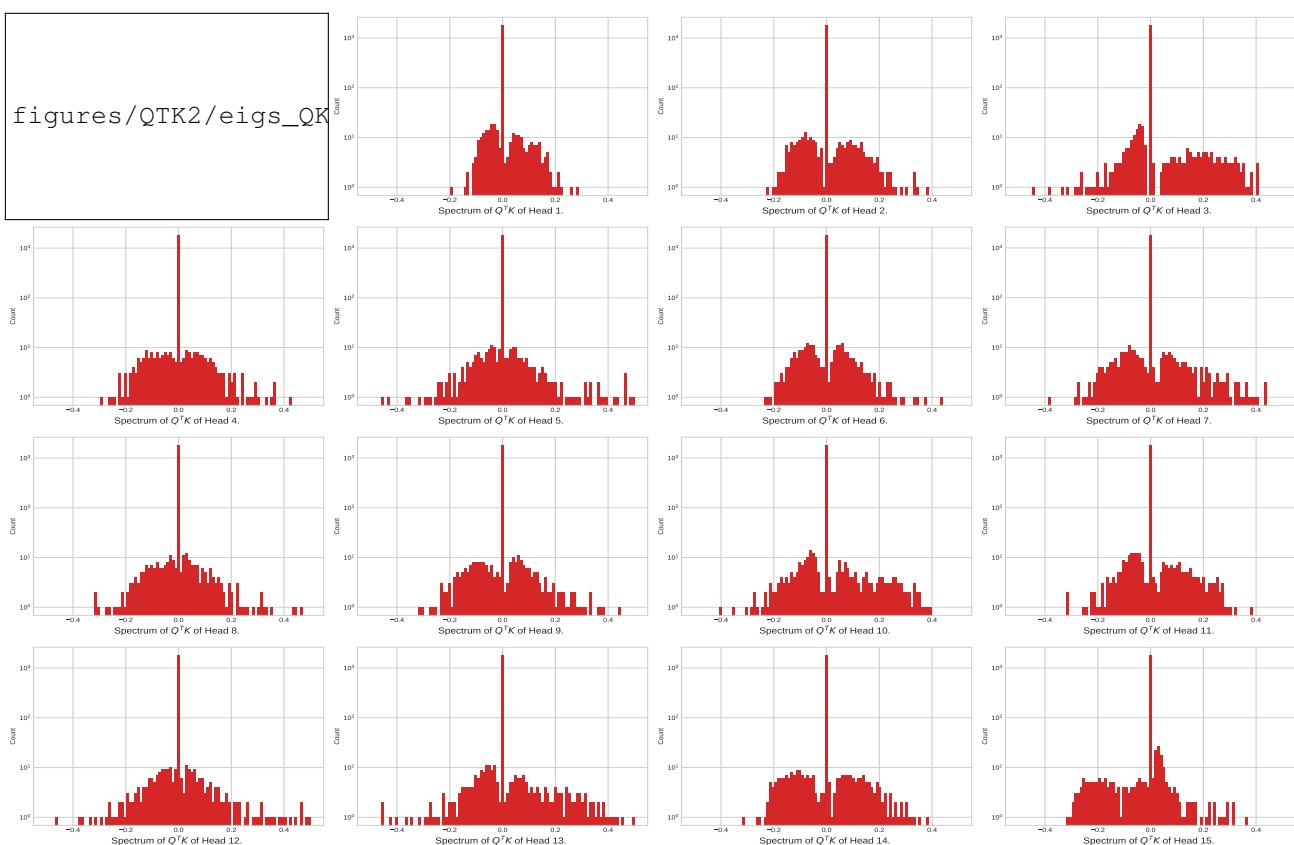

*Figure 10.* Histograms of the eigenvalues of matrix $A_h = Q_h^T K_h$ where $h$ denotes the number of attention head in a Albert XL model. The y-axis is in log-scale. The number of eigenvalues associated with the 0 eigenvalue is very large (roughly 200 ) which implies that the matrix is low-rank.

### E.4.3. EIGENVALUES OF LLAMA-2 7B'S VALUE MATRICES.

We investigated whether the assumptions of our theorems were valid for LLM models such as LlaMa 2 7B. The result is that very few attention layers seem to satisfy the assumptions of our theorems. This leaves many questions open from a theoretical point of view. We focus on Llama-2 7B avalaible online at https://huggingface.co/meta-llama/Llama-2-7b-chat-hf. This model is composed of 32 layers with dimension $d = 4096$. In addition, Llama 2 has 32 attention heads in each layer. Here we have represented the spectrum of the matrix obtained by concatenating the 32 Values matrices in each layer.

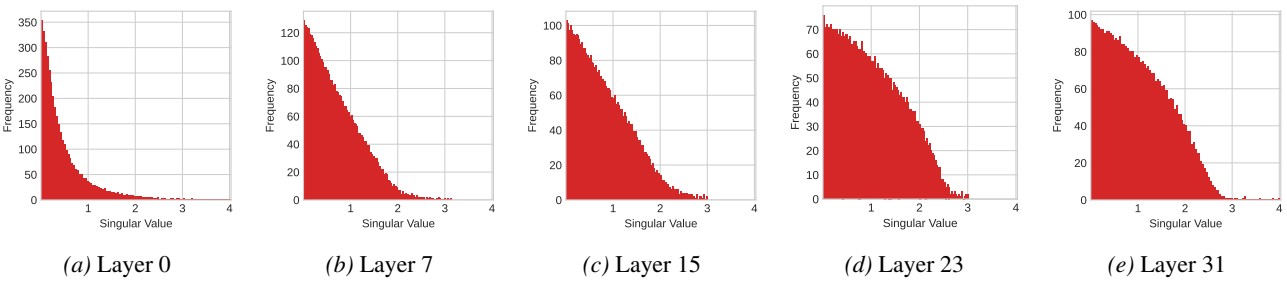

*(a)* Layer 0    *(b)* Layer 7    *(c)* Layer 15    *(d)* Layer 23    *(e)* Layer 31

*Figure 11.* The SVD values show an interesting behavior. We can see for all blocks that there are outliers, and the distribution is becoming more and more concave on the latter blocks.

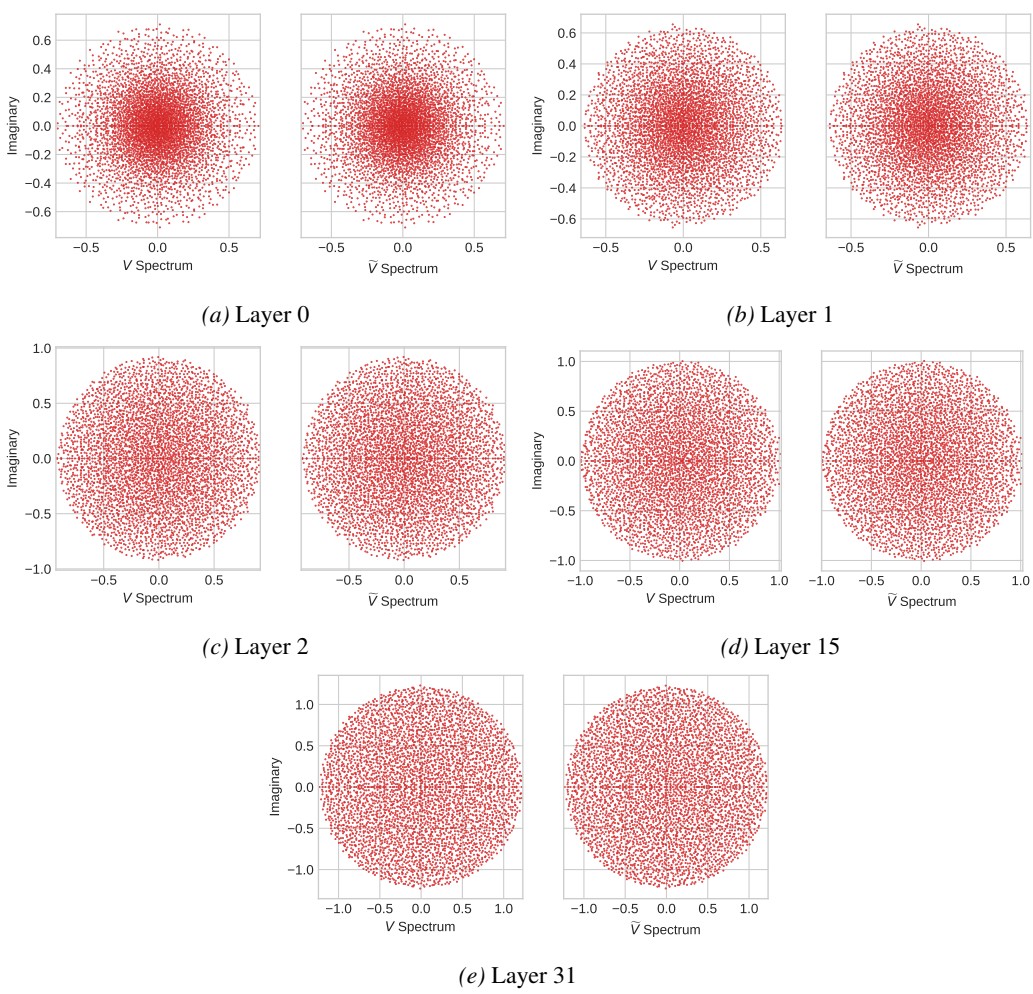

*(e)* Layer 31

*Figure 12.* Spectrum of different Values $V_h$ and $\tilde{V}_h$ matrix parameters for different layers $h$. The spectrum of both $V$ and $\tilde{V}$ seems similar to a random matrix of the Complex Ginibre Ensemble (random matrix with i.i.d Gaussian variables at each index). The spectrum of the early layers is less localized than the layers of the late layers in the sense that it seems that there are more outliers in the first layers.

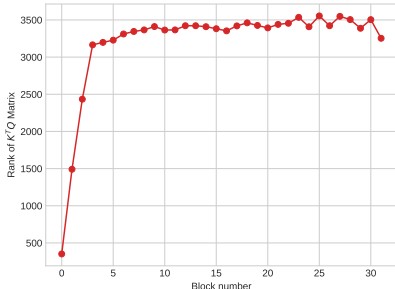

*Figure 13.* Rank of $K^T Q$ matrices for each of the 32 attention blocks of LLama-2-7b

### E.4.4. RANK OF LLAMA-2 7B'S QUERY, KEY MATRICES.

We investigated whether the assumptions of our theorems were valid for LLM models such as LlaMa 2 7B. The result is that very few attention layers seem to satisfy the assumptions of our theorems, however, from layer 3, it can be observed that the rank of the matrix $K^T Q$ is approximately 3500 in a 4096-dimensional space, which is not insignificant.. This leaves many questions open from a theoretical point of view. One surprising fact is that at the first layer, the matrix $A = K^T Q$ is low rank around 500 in a space of dimension 4096, then the rank of the matrices $A_h = K_h^T Q_h$ increases with the number of layers up to around 3600 which is not anymore small compare to the ambient space.

### E.5. Llama 2 clustering

To achieve this, we computed the correlations between the outputs of the tokens at layer $L$ in Llama 2. Surprisingly, we did not observe clear clustering when the input was text from Wikipedia, but we did observe clustering in the case of random text inputs

In this research, the Llama 2 7b model was subjected to further analysis. The model was modified by doubling the number of hidden layers from the original 32 to 64, with the latter 32 layers mirroring the weights of the first. For the input, English sentences were generated randomly using https://www.dummytextgenerator.com/, as well as from a random Wikipedia article (which happened to be about overdetermination). These sentences were then fed into the modified model, and the scalar product between each token was calculated at various stages within the model. The data, as illustrated in Figure 14, showed a consistent trend: the average scalar product between tokens gradually increased, approaching unity as the tokens progressed through the augmented layers. Interestingly, it is more visible with the dummy text. This pattern may be indicative of a systematic process of token representation refinement occurring within the model. Specifically, the rise in scalar product values suggests the possibility of clustering or convergence in the token vector space, which may imply that the model is effectively refining and aligning token representations as it processes the input through its increased depth.

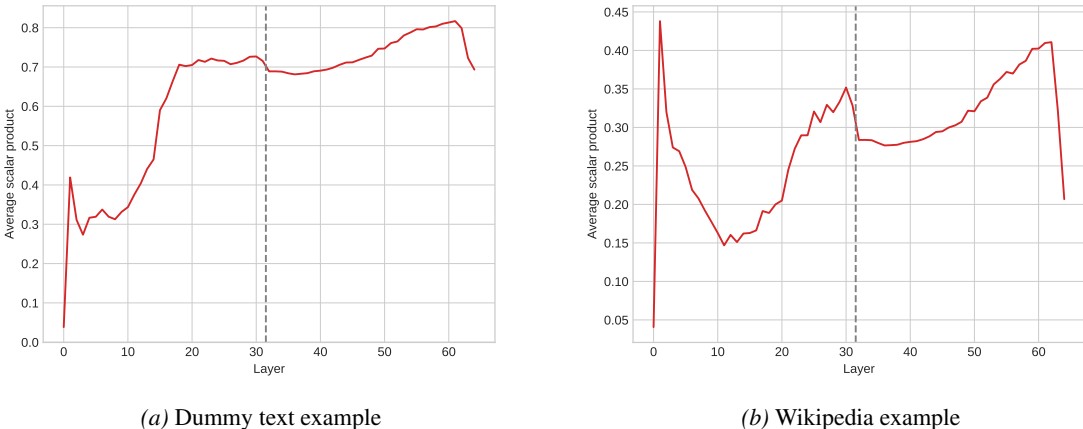

*(a)* Dummy text example                    *(b)* Wikipedia example

*Figure 14.* Average scalar product across tokens, for each layer in the augmented Llama 2 7b model, the vertical dashed line indicates where the regular Llama-2 model would have stopped

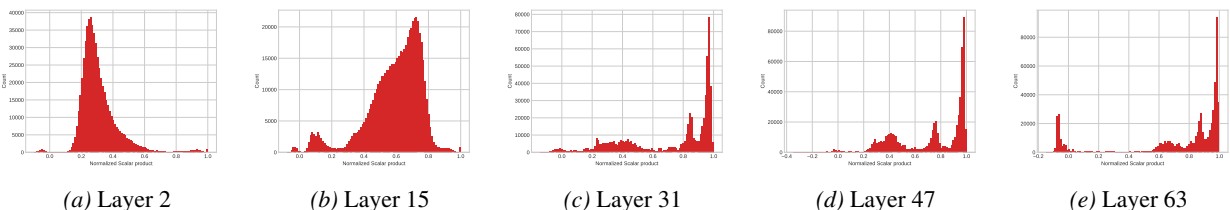

*(a)* Layer 2          *(b)* Layer 15          *(c)* Layer 31          *(d)* Layer 47          *(e)* Layer 63

*Figure 15.* Distribution of scalar products after passing through several layers of the model, for the dummy text example.

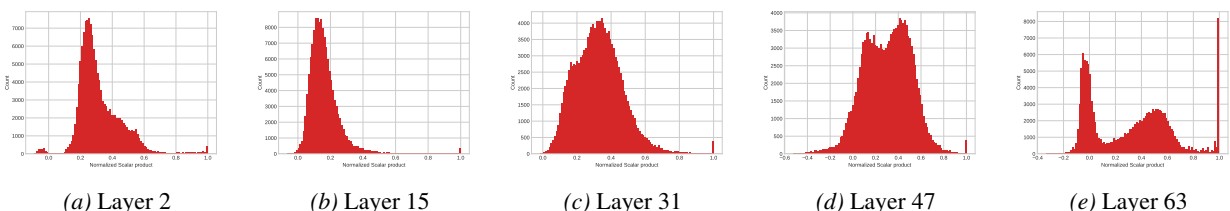

*(a)* Layer 2          *(b)* Layer 15          *(c)* Layer 31          *(d)* Layer 47          *(e)* Layer 63

*Figure 16.* Distribution of scalar products after passing through several layers of the model, for the Wikipedia example.

