# OpenReview forum: "Understanding Catastrophic Forgetting In LoRA via Mean-Field Attention Dynamics"
_ICML.cc/2026/Conference — ICML 2026 regular_

### Official Review · Reviewer_DBZH · 2026-03-08

**Soundness:** 3
**Presentation:** 3
**Significance:** 3
**Originality:** 3
**Overall Recommendation:** 4
**Confidence:** 3

**Summary:**

This paper investigates catastrophic forgetting in LoRA fine-tuning through the lens of mean-field theory. It models the Transformer forward pass as an interacting particle system and treats LoRA as a low-rank perturbation, quantifying "forgetting" as a geometric drift in token clustering. The authors derive stability bounds using Wasserstein distance and identify phase transitions related to perturbation norms and network depth. Theoretical findings suggest that alignment with the Value matrix's spectral gap drives instability, and experiments on synthetic tasks and Llama-2 offer empirical support for this geometric interpretation of forgetting.

**Compliance With Llm Reviewing Policy:**

Affirmed.

**Final Justification:**

The rebuttal usefully clarifies that the paper studies a simplified tied-weights toy model and is primarily about the inference-time effect of LoRA, which helps delimit the scope of the contribution. However, my main concerns remain only partially addressed: it still does not formally explain whether the depth-based phase transition survives layer heterogeneity, and the discrepancy between rank-insensitivity in the random-perturbation theory and the clear rank dependence observed during training is still explained mostly through intuition about optimization bias toward unstable directions. The paper itself acknowledges that the toy model does not fully capture the training-time rank effect and that its theoretical assumptions are only partially applicable to Llama 2, so I view these concerns as only partially resolved. Accordingly, I will maintain my original score.

**Key Questions For Authors:**

1. The conclusion regarding the "depth-based phase transition" relies heavily on the tied-weights assumption ($V^l = V$). In realistic settings where weights vary significantly per layer, does this "stability window" still exist, or does the heterogeneity of layers disrupt the predicted bifurcation dynamics?

2. Figure 4 demonstrates that Rank plays a significant role in forgetting during actual training, contradicting the "rank-independence" predicted for random perturbations. Can the authors provide a formal hypothesis or intuition on how the optimization process (gradient updates) specifically exploits higher ranks to align with the unstable eigen-directions ($u_1$)?

**Limitations:**

yes

**Strengths And Weaknesses:**

### Strengths

1. The application of mean-field theory and interacting particle systems to analyze LoRA-induced forgetting is highly novel, providing a rigorous geometric framework that complements existing optimization-based explanations.

2. The theoretical derivation linking the spectral gap of the Value matrix to cluster stability offers deep insights, effectively providing a theoretical justification for why methods like Orthogonal LoRA are effective in practice.


### Weaknesses

1. The theoretical model relies on strong simplifying assumptions, such as tied weights across layers and ideal clustering behavior. These assumptions create a significant gap between the theory and real-world LLMs (e.g., Llama-2), where the authors admit no clear clustering is observed on natural text.

2. There is a disconnect between theory and practice regarding Rank; the theory predicts rank-independence for random perturbations, but experiments show rank-dependence during training. The current model fails to account for the specific dynamics of gradient-based optimization.

3. The terminology "Oversmoothing" is borrowed from GNNs, and equating it directly with "catastrophic forgetting" in LLMs is not fully justified. While an empirical correlation is shown, the causal link remains somewhat loose.

---

> ### Author Rebuttal · Authors · 2026-03-30
>
> We thank the reviewer for the thoughtful comments and for highlighting these important limitations and directions for improvement.
> ___
> > The theoretical model relies on strong simplifying assumptions, such as tied weights across layers and ideal clustering behavior. These assumptions create a significant gap between the theory and real-world LLMs (e.g., Llama-2), where the authors admit no clear clustering is observed on natural text.
>
> We agree that the model is deliberately minimal. Its goal is not to provide a faithful full-scale model of Transformer training, but rather to isolate a mechanism that remains analytically tractable. In our view, this is the appropriate role of the toy model developed in the paper. Despite its simplicity, it still captures some qualitative empirical features of the phenomenon, in particular the clustering-type behavior reported in our experiments. More broadly, we believe that several components of the analysis can be extended beyond the tied-weight setting, although we do not pursue this here.
>
>  From a practical perspective, we do observe clustering-like behavior in Llama 2; see the corresponding experiments see Section E.6. Llama 2 clustering, in particular Figure 18, and 19.
>
>
>
> > There is a disconnect between theory and practice regarding Rank; the theory predicts rank-independence for random perturbations, but experiments show rank-dependence during training. The current model fails to account for the specific dynamics of gradient-based optimization.
>
> We agree with this point. Our current theory concerns random perturbations and therefore does not model the optimization dynamics of LoRA training itself. In particular, the rank-independence predicted by the isotropic random-perturbation model should not be expected to persist once the updates are shaped by gradient descent. We discuss this in more detail in our response to Question 1 from Reviewer 4dav, where we propose that the training dynamics induces anisotropic updates that are biased toward particularly sensitive or unstable eigendirections of the pretrained model.
>
> > The terminology "Oversmoothing" is borrowed from GNNs, and equating it directly with "catastrophic forgetting" in LLMs is not fully justified. While an empirical correlation is shown, the causal link remains somewhat loose.
>
> Thank you for this remark. We agree that the term oversmoothing may be misleading in this context. Our analysis is not directly about oversmoothing in the narrow sense usually used in the GNN literature. Rather, it concerns the deformation or destabilization of the asymptotic clustered geometry induced by mean-field attention dynamics. In a revised version, we will clarify this point and use more precise terminology, such as representation drift, representation collapse, or rank collapse, depending on context. We refer to our response to Reviewer EhGL for a more detailed discussion.
> ___
>
> >  The conclusion regarding the "depth-based phase transition" relies heavily on the tied-weights assumption. In realistic settings where weights vary significantly per layer, does this "stability window" still exist, or does the heterogeneity of layers disrupt the predicted bifurcation dynamics?
>
> This is a very natural and interesting question. We believe that the same qualitative mechanism should persist beyond the tied-weight setting, and that the analysis can likely be adapted to models with heterogeneous or i.i.d. layerwise attention weights, as in [1,2]. That said, this extension is nontrivial and falls outside the scope of the present paper. We therefore view it as an important direction for future work rather than a claim established here.
>
> [1 Lev Fedorov, et al. "Clustering in Deep Stochastic Transformers"
>
> [2] Borjan Geshkovski et.al. "Homogenized Transformers"
>
> ___
> > Figure 4 demonstrates that Rank plays a significant role in forgetting during actual training, contradicting the "rank-independence" predicted for random perturbations. Can the authors provide a formal hypothesis or intuition on how the optimization process (gradient updates) specifically exploits higher ranks to align with the unstable eigen-directions (
> )?
>
> Our main hypothesis is that the updates produced by training are not isotropic and, more importantly, are biased toward eigenspaces associated with dominant pretrained directions. These are precisely the directions to which the forward dynamics is most sensitive. As a result, the backward pass may naturally concentrate more energy on such subspaces during training, leading to a stronger interaction between LoRA updates and unstable directions as the rank increases. This offers a plausible explanation for the rank dependence observed in Figure 4. We do not analyze this mechanism formally in the present paper, since our focus here is on the forward dynamics, but we agree that it is an important and promising direction for future work.

---

> > ### Author Rebuttal · Reviewer_DBZH · 2026-04-03
> >
> > The rebuttal usefully clarifies that the paper studies a simplified tied-weights toy model and is primarily about the inference-time effect of LoRA, which helps delimit the scope of the contribution. However, my main concerns remain only partially addressed: it still does not formally explain whether the depth-based phase transition survives layer heterogeneity, and the discrepancy between rank-insensitivity in the random-perturbation theory and the clear rank dependence observed during training is still explained mostly through intuition about optimization bias toward unstable directions. The paper itself acknowledges that the toy model does not fully capture the training-time rank effect and that its theoretical assumptions are only partially applicable to Llama 2, so I view these concerns as only partially resolved. Accordingly, I will maintain my original score.

---

### Official Review · Reviewer_4dav · 2026-03-12

**Soundness:** 3
**Presentation:** 4
**Significance:** 3
**Originality:** 4
**Overall Recommendation:** 4
**Confidence:** 4

**Summary:**

The paper’s main goal is to provide a theoretical explanation for catastrophic forgetting under LoRA fine-tuning by modeling Transformer as a mean-field dynamical system perturbed by low-rank updates. The depth is used as a proxy for time in the dynamical system. It aims to derive conditions under which these perturbations remain stable and when eventually it causes representation drift that correlates with forgetting.

**Compliance With Llm Reviewing Policy:**

Affirmed.

**Final Justification:**

The authors have taken into consideration my concerns about the paper very well and added new theoretical and experimental results to solidify the paper. Hence, I am increasing my score.

**Key Questions For Authors:**

Question 1. For each layer, decompose the Value perturbation as
$\Delta V_\ell = a_\ell u_1 u_1^\top + C_\ell, u_1^\top C_\ell u_1 = 0.$
To model rank effects beyond isotropic second-moment control, assume the residual anisotropy $C_\ell$ increases with LoRA rank (r) through its operator norm. To explain the rank dependence in Figure 4, can you analyze this setup?

Question 2. For each depth, explicitly quantify representation drift between a large-scale pretrained model and its LoRA-adapted counterpart. Using the drift, define an empirical analogue of the bifurcation depth for a tolerance parameter. Can you test the claim of Theorem 5.6 by plotting  bifurcation depth as a function of LoRA fine-tuning step?

**Limitations:**

Yes

**Strengths And Weaknesses:**

**Strengths:**

* Theory shows norm of the perturbation controls a phase transition as found experimentally in Fig. 4.
* Authors show for a perturbed value matrix, there exists a bifurcation depth after which the perturbed dynamics can leave the pre-trained clustering neighborhood. Phase transition with respect to depth scaling: theory explains Fig. 7 qualitatively.

**Weakness:**

* The rank dependence of Fig. 4 is not explained by theoretical analysis.
* Enough large-scale experiments are not there to test the claim of Theorem 5.6.

---

> ### Author Rebuttal · Authors · 2026-03-30
>
> We warmly thank the reviewer for the stimulating questions. We particularly appreciate your improvement suggestions. We address your two questions below.
> _____
>
> >Question 2. For each depth, explicitly quantify representation drift between a large-scale pretrained model and its LoRA-adapted counterpart. Using the drift, define an empirical analogue of the bifurcation depth for a tolerance parameter. Can you test the claim of Theorem 5.6 by plotting bifurcation depth as a function of LoRA fine-tuning step?
>
> We ran the requested experiment using the Qwen 3 0.6B LoRA checkpoints from Figure 4. For each checkpoint step $s$ and depth $\ell$, we measured the relative Frobenius representation drift $\Delta_\ell(s)$ against the base model on 512 held-out $\texttt{wiki40b}$ samples, defined as:
>
> $$
> \Delta_{\ell}(s) = \frac{
>  |H_{\ell}^{LoRA}(s)-H_{\ell}^{base}|
> }{
> \lvert H_{\ell}^{base}\rvert
> }
> $$
>
> We define the empirical bifurcation depth $b_\tau(s)$ as the first layer where $\Delta_\ell(s) > \tau$, as suggested.
>
> The results ([link](https://ibb.co/TfNb7hS)) strongly align with Theorem 5.6. As fine-tuning progresses, the stability window shrinks. For $\tau=0.2$, the average bifurcation depth drops from layer 28 at the start of training down to layer 16 at the end. We also checked cosine distance for robustness and found the exact same qualitative behavior (dropping from layer 28 down to 17 at $\tau=0.02$).
>
> Additionally, we generated depth-by-step heatmaps showing the drift originating in deeper layers and propagating to shallower layers over time (Links by rank: [8](https://ibb.co/9kJ3NWyZ), [32](https://ibb.co/VYFQ2834), [128](https://ibb.co/bRC6tKwM), [512](https://ibb.co/1GhgfHpz) ).
>
> We will include the bifurcation curve in the main text and these heatmaps in the appendix. While the fine-tuning step $s$ acts as an empirical proxy for the accumulated LoRA perturbation, this serves as a strong practical validation of the dynamics predicted by Theorem 5.6.
>
> ___
> > **Question 1.** *For each layer, decompose the Value perturbation as $\Delta V^\ell = a^\ell u_1u_1^\top + C^\ell$. To model rank effects beyond isotropic second-moment control, assume the residual anisotropy increases with the LoRA rank $r$ through its operator norm. Can you analyze this setup and explain the rank dependence observed in Figure 4?*
>
> We thank the reviewer for this suggestion. The question is a bit ambiguous (random or deterministic LoRA), and here we focus here on the deterministic case, which already yields a clean rank-sensitive mechanism.
>
> Assume that, for each layer $\ell$, the LoRA update is deterministic and anisotropic:
> $$
> \Delta V^\ell = C^\ell,
> $$
> where $C^\ell \in \mathbb{R}^{d\times d}$ has rank $r$ and satisfies
> $$
> u_1^\top C^\ell u_1 = 0.
> $$
> Here $u_1$ denotes the dominant eigenvector of the pretrained matrix $V$.
>
> To simplify the discussion, consider first the tied case $C^\ell \equiv C$, and let
> $
> \widetilde V = V + C.
> $
> Let $\widetilde u_1$ be a leading eigenvector of $\widetilde V$, normalized by
> $
> u_1^\top \widetilde u_1 = 1.
> $
>
> Writing
> $
> \widetilde u_1 = u_1 + z, z \perp u_1,
> $
> and decomposing into $\mathrm{span}(u_1)$ and $u_{1}^\perp$ ,one obtains
> $$
> z = (\tilde{\lambda}_1 I - \Lambda - E)^{-1}\gamma,
> $$
> where $\widetilde\lambda_1$ is the leading eigenvalue of $\widetilde V$, $\Lambda$ is the restriction of $V$ to $u_1^\perp$, $E$ is the restriction of $C$ to $u_1^\perp$, and $\gamma = C u_1.$
>
> Assume for convenience that $E$ and $\Lambda$ commute, the expression becomes diagonal in the eigenbasis $(u_j)_{j\ge 2}$ of $V$. Writing
>
> $$
> \gamma = \sum_{j=2}^d \alpha_j u_j, \qquad \alpha_j := \langle C u_1, u_j\rangle,
> $$
>
> we obtain the approximation
>
> $$
> \|u_1-\widetilde u_1\|^2= \frac{\|z\|^2}{1+\|z\|^2 }\approx\sum_{j=2}^d(\frac{\alpha_j}{\lambda_1-\lambda_j-e_j(r)})^2,
> $$
>
> where $(e_j(r))_{j=2}^d$ are the eigenvalues of $E$.
>
> This expression makes the rank effect transparent, the perturbation becomes larger when the update places more mass on  directions $u_j$ for which the spectral gap $\lambda_1-\lambda_j$ is small. Since $\operatorname{rank}(C)=r$, only a limited number of $\alpha_{j}\neq 0$, and increasing $r$ allows the update to affect more such directions. In particular, under the commutation assumption, the deviation is given by
>
> $$ \|u_1-\widetilde u_1\|^2
> \approx
> \sum_{j:\, e_j(r)\neq 0}
> \left(
> \frac{\alpha_j}{\lambda_1-\lambda_j-e_j(r)}
> \right)^2.
> $$
> This provides a deterministic rank-sensitive analogue of Proposition 4.3, the anisotropic transverse component $C u_1$ controls the drift of the leading eigendirection. In particular, higher-rank LoRA updates can induce larger forgetting when they align with eigenspaces corresponding to smaller spectral gaps.
>
> We agree that this refined deterministic perspective helps explain the empirical behavior in Figure 4, and we will mention this deterministic extension in a revised version.

---

> > ### Author Rebuttal · Reviewer_4dav · 2026-04-03
> >
> > The authors have addressed all my concerns well in the rebuttal, and I highly appreciate their clarifications. Therefore, I would like to increase my original score to 4: weak accept.

---

### Official Review · Reviewer_EhGL · 2026-03-12

**Soundness:** 3
**Presentation:** 2
**Significance:** 3
**Originality:** 3
**Overall Recommendation:** 4
**Confidence:** 3

**Summary:**

The paper investigates the concept of catastrophic forgetting under LoRA using a mean-field, share weight single-head attention toy model, where token embeddings evolve as interacting particles and LoRA is modeled as a low-rank perturbation of attention. The authors consider the concept of forgetting as drift in token-clustering geometry, and they prove finite-time Wasserstein stability bound, stability result of asymptotic cluster in Post-LayerNorm dynamic, and norm- and depth-driven phase transitions, supported by synthetic, modular-arithmetic, and large-model experiments.

**Compliance With Llm Reviewing Policy:**

Affirmed.

**Final Justification:**

The authors have adequately addressed my concerns. Thus, I keep my positive score.

**Key Questions For Authors:**

Please refer to the Weaknesses.

**Limitations:**

Yes.

**Strengths And Weaknesses:**

**Strengths:**

- The paper addresses an important problem: understanding catastrophic forgetting in LoRA from a principled theoretical perspective.
- The theoretical contributions are substantial.

**Weaknesses:**

- The work does not appear directly related to oversmoothing. The authors likely mean the clustering phenomenon from the mean-field Transformer literature. Including "Oversmoothing" in the title is misleading, as the main text never mentions or connects the two concepts.
- The toy model makes strong simplifying assumptions (weight sharing, $V=A$, single head, no MLP) that limit the applicability of the results in practical scenarios. While these seem necessary for tractability, could the authors discuss how the conclusions would change if the assumptions are relaxed
- Proposition 4.3 requires $\Delta V \in \text{Sym}(d)$, but LoRA updates are generally not symmetric. Could the analysis extend to the non-symmetric case?
- In Assumption 5.1, the distribution of $\Sigma_V^\ell$ should be clarified in the main text (it only appears in Appendix D.1). The notation $\wedge$ in Eq. (25) is also undefined.
- Please provide detailed experimental settings for Figure 4 (dataset, learning rate, number of seeds). Without these, robustness is hard to assess.
- Minor: Line 307/308 has a typo ("Equations (26) and (26)" should be "(25) and (26)"). The last equation on page 13 (Line 712) appears to be missing an equal sign.

---

> ### Author Rebuttal · Authors · 2026-03-30
>
> We thank the reviewer for their time in reviewing our work and overall positive feedback! Below we address their concerns in the weaknesses.
>
> ---
> > Weakness:
> >  The work does not appear directly related to oversmoothing. The authors likely mean the clustering phenomenon from the mean-field Transformer literature.
> >   Including "Oversmoothing" in the title is misleading, as the main text never mentions or connects the two concepts.
>
> We thank the reviewer for this remark. We agree that the term oversmoothing is often understood as the convergence of token representations toward indistinguishable features. Our analysis is slightly different. In that sense, rank collapse and oversmoothing should be viewed as related motivating phenomena, whereas our main results are more directly about cluster formation, spectral stability, and representation drift. We will revise the terminology accordingly and clarify this distinction in the paper. Possible alternatives include rank collapse, representation collapse, token homogenization, or token uniformity, depending on the context. We refer to [1,2,3] for representative results in this direction.
>
> [1] Ruili Feng et al., Rank Diminishing in Deep Neural Networks
>
> [2] Lorenzo Noci et al., Signal Propagation in Transformers: Theoretical Perspectives and the Role of Rank Collapse
>
> [3] Shuangfei Zhai et al. Stabilizing transformer training by preventing attention
> entropy collapse
>
> ___
>
> > Weakness:
> >  The toy model makes strong simplifying assumptions (weight sharing,
> , single head, no MLP) that limit the applicability of the results in practical scenarios. While these seem necessary for tractability, could the authors discuss how the conclusions would change if the assumptions are relaxed
>
> We split our response into two parts.
> * Clustering in practical models.
> From a practical perspective, we do observe clustering-like behavior in Llama 2; see the corresponding experiments see Section E.6. Llama 2 clustering, in particular Figure 18, and 19.
>
> * Assumption needed:
> This is a very reasonable question. The assumptions used in the paper are sufficient for our analysis, but they are certainly not necessary. There are several mathematically tractable settings in which clustering-type behavior can be analyzed, for instance:
>    * General weight-sharing case [4]
>    * Transformers at initialization (with random heads) [1,2],
>    * Continuous-time varying weights [3],
>    * Causal masking transformers [5] explaining the emergence of Attention sinks [6].
>
> We restrict in the present paper to the repeated-head setting mainly for clarity of exposition and to keep the arguments as transparent as possible. We expect the qualitative mechanism to extend to a broader class of models, and we chose this setting only as a first step.
>
> [1] Lev Fedorov, et al. "Clustering in Deep Stochastic Transformers"
>
> [2] Borjan Geshkovski et.al. "Homogenized Transformers"
>
> [3] Álvaro Rodríguez Abella et.al. "Consensus Is All You Get: The Role of Attention in Transformers"
>
> [4] Giuseppe Bruno, et al. "Emergence of Meta-Stable Clustering in Meanfield Transformer Models"
>
> [5] Nikita Karagodin et.al "Clustering in Causal Attention Masking"
>
> [6] Guangxuan Xiao et.al. "Efficient Streaming Language Models with Attention Sinks"
>
> ___
> >  Please provide detailed experimental settings for Figure 4 (dataset, learning rate, number of seeds).
> >
> >  Without these, robustness is hard to assess.
>
> Thank you for pointing this out. We will add the experimental settings for Figure~4 explicitly in the revised version. For the LoRA training forgetting experiment, we fine-tune Qwen/Qwen3-0.6B on the $\texttt{flytech/python-codes-25k}$ training split, and we evaluate forgetting on a distinct base dataset, namely $\texttt{wiki40b}$ with French configuration ($\texttt{fr}$), test split, text field \texttt{text}, using $512$ evaluation samples. The LoRA runs use learning rate $5\times 10^{-4}$, batch size $32$ with gradient accumulation $2$, for $10$ epochs, with evaluation every $20$ steps. The seed is fixed at 42.
>
> ___
> > Proposition 4.3 requires $\Delta V \in \mathrm{Sym}$ but LoRA updates are generally not symmetric. Could the analysis .
>
> We thank the reviewer for this suggestion. The spectral part of the argument can indeed be extended to non-symmetric update matrices. The main reason we restricted to the symmetric case in the paper was ease of presentation, together with the fact that the symmetric setting is more directly compatible with the gradient-flow structure used in our dynamical analysis.

---

> > ### Author Rebuttal · Reviewer_EhGL · 2026-04-03
> >
> > Thank you for the response. I will maintain my original score.

---

### Decision · Program_Chairs · 2026-04-30

**Decision:**

Accept (regular)

**Comment:**

This paper studies theoretically catastrophic forgetting during LoRA fine-tuning. It models the Transformer forward pass as a mean-field dynamical system and treats LoRA as a low-rank perturbation within that setup. Using ideas from dynamical systems, the authors identify phase transitions driven by factors like perturbation size and network depth. Their analysis also provides bounds on when the model starts to drift and connects the spectral gap of the Value matrix to the stability of representation clusters. These predictions line up well with experiments on real large language models.

The reviewers found the work clear, novel, and theoretically strong, especially in how it uses mean-field ideas to shed light on LoRA-related forgetting, backed by solid empirical results. Some concerns came up during the discussion, including the use of the term “oversmoothing,” strong simplifying assumptions like tied weights, and questions about experimental details and an apparent mismatch between theory and observed rank effects. The authors responded thoroughly: they agreed to fix the terminology, clarified that the assumptions were mainly for analytical simplicity, and added an extension that explains the rank dependence seen in experiments. Given the strength of the theory, the supporting experiments, and the authors’ constructive responses, all the reviewers proposed acceptance and **I also recommend accepting the paper.**